# Near-Optimal Policy Identification in Robust Constrained Markov Decision Processes via Epigraph Form

**Toshinori Kitamura[1*], Tadashi Kozuno[2,4], Wataru Kumagai[2], Kenta Hoshino[3],**
**Yohei Hosoe[3], Kazumi Kasaura[2], Masashi Hamaya[2], Paavo Parmas[1], Yutaka Matsuo[1]**

[1] The University of Tokyo, [2] OMRON SINIC X Corporation, [3] Kyoto University, [4] Osaka University
[*]toshinori-k@weblab.t.u-tokyo.ac.jp

## Abstract

Designing a safe policy for uncertain environments is crucial in real-world control systems. However, this challenge remains inadequately addressed within the Markov decision process (MDP) framework. This paper presents the first algorithm guaranteed to identify a near-optimal policy in a robust constrained MDP (RCMDP), where an optimal policy minimizes cumulative cost while satisfying constraints in the worst-case scenario across a set of environments. We first prove that the conventional policy gradient approach to the Lagrangian max-min formulation can become trapped in suboptimal solutions. This occurs when its inner minimization encounters a sum of conflicting gradients from the objective and constraint functions. To address this, we leverage the epigraph form of the RCMDP problem, which resolves the conflict by selecting a single gradient from either the objective or the constraints. Building on the epigraph form, we propose a bisection search algorithm with a policy gradient subroutine and prove that it identifies an $\varepsilon$-optimal policy in an RCMDP with $\widetilde{\mathcal{O}}(\varepsilon^{-4})$ robust policy evaluations.

## 1 Introduction

In real-world decision-making, it is crucial to design policies that satisfy safety constraints even in uncertain environments. For example, self-driving cars must drive efficiently while maintaining a safe distance from obstacles, regardless of environmental uncertainties such as road conditions, weather, or the state of the vehicle's components. Traditionally, within the Markov decision process (MDP) framework, constraint satisfaction and environmental uncertainty have been addressed separately—through constrained MDP (CMDP; e.g., Altman, 1999), which aims to minimize cumulative cost while satisfying constraints, and robust MDP (RMDP; e.g., Iyengar, 2005), which aims to minimize the worst-case cumulative cost over an uncertainty set of possible environments. However, in practice, both robustness and constraint satisfaction are important. The recent robust constrained MDP (RCMDP) framework addresses this dual need by aiming to minimize the worst-case cost while robustly satisfying the constraints. Despite significant theoretical progress in CMDPs and RMDPs (see Appendix A), theoretical results on RCMDPs remain scarce. Even in the tabular setting, where the state and action spaces are finite, there exists no algorithm with guarantees to find a near-optimal policy in an RCMDP.

The difficulty of RCMDPs arises from the challenging optimization process, which simultaneously considers robustness and constraints. The dynamic programming (DP) approach, popular in unconstrained RMDPs, is unsuitable for constrained settings where Bellman's principle of optimality can be violated (Haviv, 1996; Bellman et al., 1957). Similarly, the linear programming (LP) approach, commonly used for CMDPs, is inadequate due to the nonconvexity of the robust formulation (Iyengar, 2005; Grand-Clément & Petrik, 2024). Consequently, the policy gradient method with the Lagrangian formulation has been studied as the primary remaining option (Russel et al., 2020; Wang et al., 2022). The Lagrangian formulation approximates the RCMDP problem $\min_\pi \{f(\pi) \mid h(\pi) \leq 0\}$ by $\max_{\lambda \geq 0} \min_\pi f(\pi) + \lambda h(\pi)$, where $f(\pi)$ and $h(\pi)$ represent the

| Approach | MDP | CMDP | RMDP | RCMDP |
|---|---|---|---|---|
| Dynamic Programming | ✓ 
 Bellman et al. (1957) | ✗ | ✓ 
 Iyengar (2005) | ✗ |
| Linear Programming | ✓ 
 Denardo (1970) | ✓ 
 Altman (1999) | ✗ | ✗ |
| Lagrangian + PG | ✓ 
 Agarwal et al. (2021) | ✓ 
 Ding et al. (2020) | ✓ 
 Wang et al. (2023) | ✗ |
| **Epigraph + PG (Ours)** | ✓ | ✓ | ✓ | ✓ |

Table 1: Summary of approaches and the problem settings. "PG" denotes Policy Gradient. Each cell displays a "✓" indicating the presence of an algorithm with this approach that guarantees yielding an $\varepsilon$-optimal policy. Representative works supporting each "✓" are listed below it. Conversely, "✗" denotes settings where the approach either isn't suitable or lacks performance guarantees.

worst-case cumulative cost—called the (cost) return[1]—and the worst-case constraint violation of a policy $\pi$, respectively. There have been a few attempts to provide theoretical guarantees for the Lagrangian approach (Wang et al., 2022; Zhang et al., 2024); however, no existing studies offer rigorous and satisfactory guarantees that the max-min problem yields the same solution as the original RCMDP problem. As a result, existing Lagrangian-based algorithms lack theoretical performance guarantees. This leaves us with a fundamental question:

*How can we identify a near-optimal policy in a tabular RCMDP?*

We address this question by presenting three key contributions, which are summarized as follows:

**Gradient conflict in the Lagrangian formulation (Section 3).** We first show that solving the Lagrangian formulation is inherently difficult, even when its max-min problem can yield an optimal policy. Given the limitations of DP and LP approaches as discussed, the policy gradient method might seem like a viable alternative to solve the max-min. However, our Theorem 1 reveals that policy gradient methods can get trapped in a local minimum during the inner minimization of the Lagrangian formulation. This occurs when the gradients, $\nabla f(\pi)$ and $\nabla h(\pi)$, conflict with each other, causing their sum $\nabla f(\pi) + \lambda \nabla h(\pi)$ to cancel out, even when the policy $\pi$ is not optimal. Consequently, the Lagrangian approach for RCMDPs may not reliably lead to a near-optimal policy.

**Epigraph form of RCMDP (Section 4).** We then demonstrate that the *epigraph form*, commonly used in constrained optimization literature (Boyd & Vandenberghe, 2004; Beyer & Sendhoff, 2007; Rahimian & Mehrotra, 2019), entirely circumvents the challenges associated with the Lagrangian formulation. The epigraph form transforms the RCMDP problem into $\min_y \{y \mid \min_\pi \max\{f(\pi) - y, h(\pi)\} \leq 0\}$, introducing an auxiliary minimization problem of $\min_\pi \max\{f(\pi) - y, h(\pi)\}$ and minimizing its threshold variable $y$. Unlike the Lagrangian approach, which necessitates **summing** $\nabla f(\pi)$ **and** $\nabla h(\pi)$, policy gradient methods for the auxiliary problem update the policy by **selecting either** $\nabla f(\pi)$ **or** $\nabla h(\pi)$, thanks to the maximum operator in the problem. As a result, the epigraph form avoids the problem of conflicting gradient sums, preventing policy gradient methods from getting stuck in suboptimal minima (Theorem 4).

**A new RCMDP algorithm (Section 5).** Finally, we propose a tabular RCMDP algorithm called **Epi**graph **R**obust **C**onstrained **P**olicy **G**radient **S**earch (**EpiRC-PGS**, pronounced as "Epic-P-G-S"). The algorithm employs a double-loop structure: the inner loop verifies the feasibility of the threshold variable $y$ by performing policy gradients on the auxiliary problem, while the outer loop employs a bisection search to determine the minimal feasible $y$. EpiRC-PGS is guaranteed to find an $\varepsilon$-optimal policy[2] with $\widetilde{\mathcal{O}}(\varepsilon^{-4})$ robust policy evaluations (Corollary 1), where $\widetilde{\mathcal{O}}(\cdot)$ represents the conventional big-O notation excluding polylogarithmic terms. Since RCMDP generalizes plain MDP, CMDP, and RMDP, our EpiRC-PGS is applicable to all these types of MDPs, ensuring near-optimal policy identification for each. Table 1 compares existing approaches in various MDP settings. Due to the page limitation, more related work is provided in Appendix A. We discuss limitations and potential future directions in Section 7.

---

[1]We often use the term *return* to refer specifically to the objective cost return. When discussing a return value in the context of RCMDP's constraints, we refer to it as the *constraint return*.

[2]The definition of an $\varepsilon$-optimal policy is provided in Definition 1

## 2 PRELIMINARY

We use the shorthand $\mathbb{R}_+ := [0, \infty)$. The set of probability distributions over $\mathcal{S}$ is denoted by $\mathscr{P}(\mathcal{S})$. For two integers $a \leq b$, we define $[\![a, b]\!] := \{a, \dots, b\}$. If $a > b$, $[\![a, b]\!] := \emptyset$. For a vector $x \in \mathbb{R}^N$, its $n$-th element is denoted by $x_n$ or $x(n)$, and we use the convention that $\|x\|_2 = \sqrt{\sum_i x_i^2}$ and $\|x\|_\infty = \max_i |x_i|$. For two vectors $x, y \in \mathbb{R}^N$, we denote $\langle x, y \rangle = \sum_i x_i y_i$. We let $\mathbf{0} := (0, \dots, 0)^\top$ and $\mathbf{1} := (1, \dots, 1)^\top$, with their dimensions being clear from the context. All scalar operations and inequalities should be understood point-wise when applied to vectors and functions. Given a finite set $\mathcal{S}$, we often treat a function $f : \mathcal{S} \to \mathbb{R}$ as a vector $f \in \mathbb{R}^{|\mathcal{S}|}$. Both notations, $f : \mathcal{S} \to \mathbb{R}$ and $f \in \mathbb{R}^{|\mathcal{S}|}$, are used depending on notational convenience. Finally, $\partial f(x)$ and $\nabla f(x)$ denote the set of (Fréchet) subgradients and gradient of $f : \mathcal{X} \to \mathbb{R}$ at a point $x$, respectively. Their formal definitions are deferred to Definition 2.

### 2.1 CONSTRAINED MARKOV DECISION PROCESS

Let $N \in \mathbb{Z}_{\geq 0}$ be the number of constraints. An infinite-horizon discounted constrained MDP (CMDP) is defined as a tuple $(\mathcal{S}, \mathcal{A}, \gamma, P, \mathcal{C}, b, \mu)$, where $\mathcal{S}$ denotes the finite state space with size $S$, $\mathcal{A}$ denotes the finite action space with size $A$, $\gamma \in (0, 1)$ denotes the discount factor, and $\mu \in \mathscr{P}(\mathcal{S})$ denotes the initial state distribution. For notational brevity, let $H := (1 - \gamma)^{-1}$ denotes the effective horizon. Further, $b := (b_1, \dots, b_N) \in [0, H]^N$ denotes the constraint threshold vector, where $b_n$ is the threshold scalar for the $n$-th constraint, $\mathcal{C} := \{c_n\}_{n \in [\![0, N]\!]}$ denotes the set of cost functions, where $c_n : \mathcal{S} \times \mathcal{A} \to [0, 1]$ denotes the $n$-th cost function and $c_n(s, a)$ denotes the $n$-th cost when taking an action $a$ at a state $s$. $c_0$ is for the objective to optimize and $\{c_1, \dots, c_N\}$ are for the constraints. $P : \mathcal{S} \times \mathcal{A} \to \mathscr{P}(\mathcal{S})$ denotes the transition probability kernel, which can be interpreted as the environment with which the agent interacts. $P(s' \mid s, a)$ denotes the state transition probability to a new state $s'$ from a state $s$ when taking an action $a$.

### 2.2 POLICY AND VALUE FUNCTIONS

A (Markovian stationary) policy is defined as $\pi \in \mathbb{R}^{SA}$ such that $\pi(s, \cdot) \in \mathscr{P}(\mathcal{A})$ for any $s \in \mathcal{S}$. $\pi(s, a)$ denotes the probability of taking an action $a$ at state $s$. The set of all the policies is denoted as $\Pi$, which corresponds to the *direct parameterization* policy class presented in Agarwal et al. (2021). Although non-Markovian policies can yield better performance in general RMDP problems (Wiesemann et al., 2013), for simplicity, we focus on Markovian stationary policies in this paper. With an abuse of notation, for two functions $\pi, g \in \mathbb{R}^{SA}$, we denote $\langle \pi, g \rangle = \sum_{s, a \in \mathcal{S} \times \mathcal{A}} \pi(s, a) g(s, a)$.

For a policy $\pi$ and transition kernel $P$, let $d_P^\pi : \mathcal{S} \to \mathbb{R}_+$ denote the occupancy measure of $\pi$ under $P$. $d_P^\pi(s)$ represents the expected discounted number of times $\pi$ visits state $s$ under $P$, such that $d_P^\pi(s) = (1 - \gamma)\mathbb{E}\left[\sum_{h=0}^\infty \gamma^h \mathbb{1}\{s_h = s\} \mid s_0 \sim \mu, \pi, P\right]$. Here, the notation means that the expectation is taken over all possible trajectories, where $a_h \sim \pi(s_h, \cdot)$ and $s_{h+1} \sim P(\cdot \mid s_h, a_h)$.

For a $\pi \in \mathbb{R}^{SA}$ and a cost $c \in \mathbb{R}^{SA}$, let $Q_{c,P}^\pi : \mathcal{S} \times \mathcal{A} \to \mathbb{R}$ be the action-value function such that [3]

$$Q_{c,P}^\pi(s, a) = c(s, a) + \gamma \sum_{s' \in \mathcal{S}} P(s' \mid s, a) \sum_{a' \in \mathcal{A}} \pi(s', a') Q_{c,P}^\pi(s', a') \quad \forall (s, a) \in \mathcal{S} \times \mathcal{A} .$$

Let $V_{c,P}^\pi : \mathcal{S} \to \mathbb{R}$ be the state-value function such that $V_{c,P}^\pi(s) = \sum_{a \in \mathcal{A}} \pi(s, a) Q_{c,P}^\pi(s, a)$ for any $s \in \mathcal{S}$. If $\pi \in \Pi$, $V_{c,P}^\pi(s)$ represents the expected cumulative cost of $\pi$ under $P$ with an initial state $s$. We denote the (cost) return function as $J_{c,P}(\pi) := \sum_{s \in \mathcal{S}} \mu(s) V_{c,P}^\pi(s)$.

**Policy gradient method.** For a problem $\min_{\pi \in \Pi} f(\pi)$ where $f : \Pi \to \mathbb{R}$ is differentiable at $\pi \in \Pi$, policy gradient methods with direct parameterization update $\pi$ to a new policy $\pi'$ as follows:

$$\pi' := \text{Proj}_\Pi(\pi - \alpha \nabla f(\pi)) \overset{(a)}{=} \arg\min_{\pi' \in \Pi} \langle \nabla f(\pi), \pi' - \pi \rangle + \frac{1}{2\alpha} \|\pi' - \pi\|_2^2 , \quad (1)$$

where $\alpha > 0$ is the learning rate and $\text{Proj}_\Pi$ denotes the Euclidean projection operator onto $\Pi$. The equality (a) is a standard result (see, e.g., Parikh et al. (2014)). The following lemma provides the gradient of $J_{c,P}(\pi)$ for the direct parameterization policy class $\Pi$ (e.g., Agarwal et al. (2021)).

---

[3] The domain of $Q_{c,P}^\pi$ is not restricted to $\Pi$ to ensure well-defined policy gradients over $\pi \in \Pi$.

**Lemma 1** (Policy gradient theorem). *For any $\pi \in \Pi$, transition kernel $P : \mathcal{S} \times \mathcal{A} \to \mathscr{P}(\mathcal{S})$, and cost $c \in \mathbb{R}^{SA}$, the gradient is given by $(\nabla J_{c,P}(\pi))(s,a) = H d_P^\pi(s) Q_{c,P}^\pi(s,a) \quad \forall (s,a) \in \mathcal{S} \times \mathcal{A}$.*

## 2.3 ROBUST CONSTRAINED MARKOV DECISION PROCESS

An infinite-horizon discounted robust constrained MDP (RCMDP) is defined as a tuple $(\mathcal{S}, \mathcal{A}, \gamma, \mathcal{U}, \mathcal{C}, b, \mu)$, where $\mathcal{U}$ is a compact set of transition kernels, called the uncertainty set, which can be either finite or infinite. The infinite uncertainty set typically requires some structural assumptions. A common structure is the $(s,a)$-rectangular set (Iyengar, 2005; Nilim & El Ghaoui, 2005), defined as $\mathcal{U} = \times_{s,a} \mathcal{U}_{s,a}$, where $\mathcal{U}_{s,a} \subseteq \mathscr{P}(\mathcal{S})$ and $\times_{s,a}$ denotes a Cartesian product over $\mathcal{S} \times \mathcal{A}$. We remark that our work is **not** limited to any specific structural assumption, but rather considers a general, tractable uncertainty set (see Assumptions 1, 2 and 3). When $N = 0$, an RCMDP reduces to an RMDP. When $\mathcal{U} = \{P\}$, an RCMDP becomes a CMDP.

For a cost $c \in \mathbb{R}^{SA}$, let $J_{c,\mathcal{U}}(\pi) := \max_{P \in \mathcal{U}} J_{c,P}(\pi)$ denote the worst-case (cost) return function, which represents the cost return of $\pi$ under the most adversarial environment within $\mathcal{U}$. The goal of an RCMDP is to find a solution to the following constrained optimization problem:

$$(\textbf{RCMDP}) \quad J^\star := \min_{\pi \in \Pi} J_{c_0,\mathcal{U}}(\pi) \text{ such that } J_{c_n,\mathcal{U}}(\pi) \le b_n \quad \forall n \in [\![1,N]\!] . \tag{2}$$

Let $\Pi_{\mathrm{F}} := \left\{ \pi \in \Pi \mid \max_{n \in [\![1,N]\!]} J_{c_n,\mathcal{U}}(\pi) - b_n \le 0 \right\}$ be the set of all the feasible policies. We assume that $\Pi_{\mathrm{F}}$ is non-empty. An optimal policy $\pi^\star \in \Pi_{\mathrm{F}}$ is a solution to Equation (2).

**Definition 1.** $\pi \in \Pi$ is $\varepsilon$-optimal[4] if $J_{c_0,\mathcal{U}}(\pi) - J^\star \le \varepsilon$ and $\max_{n \in [\![1,N]\!]} J_{c_n,\mathcal{U}}(\pi) - b_n \le \varepsilon$.

## 3 CHALLENGES OF LAGRANGIAN FORMULATION

To motivate our formulations and algorithms presented in subsequent sections, this section illustrates the limitations of using the conventional Lagrangian approach for RCMDPs. By introducing Lagrangian multipliers $\lambda := (\lambda_1, \ldots, \lambda_N) \in \mathbb{R}_+^N$, Equation (2) is equivalent to

$$J^\star = \min_{\pi \in \Pi} \max_{\lambda \in \mathbb{R}_+^N} J_{c_0,\mathcal{U}}(\pi) + \sum_{n=1}^{N} \lambda_n (J_{c_n,\mathcal{U}}(\pi) - b_n) .$$

As this $\min_\pi$ is hard to solve due to the inner maximization, Russel et al. (2020); Mankowitz et al. (2020); Wang et al. (2022) swap the min-max and consider the following Lagrangian formulation:

$$(\textbf{Lagrange}) \quad L^\star := \max_{\lambda \in \mathbb{R}_+^N} \min_{\pi \in \Pi} L_\lambda(\pi) \text{ where } L_\lambda(\pi) := J_{c_0,\mathcal{U}}(\pi) + \sum_{n=1}^{N} \lambda_n (J_{c_n,\mathcal{U}}(\pi) - b_n) . \tag{3}$$

Let $\lambda^\star$ be a solution to Equation (3). The Lagrangian approach aims to solve Equation (3) by expecting $\pi^\star \in \arg\min_{\pi \in \Pi} L_{\lambda^\star}(\pi)$. However, this expectation may not hold, as swapping the min-max is not necessarily equivalent to the original min-max problem (Boyd & Vandenberghe, 2004). Therefore, to guarantee the performance of the Lagrange approach, the two questions must be addressed:

(i) *Can we ensure $\pi^\star \in \arg\min_{\pi \in \Pi} L_{\lambda^\star}(\pi)$?*    (ii) *If so, is it tractable to solve $\min_{\pi \in \Pi} L_\lambda(\pi)$?*

However, answering these questions affirmatively is challenging due to the following issues:

(i) **$\pi^\star$ solution challenge.** To ensure that $\pi^\star \in \arg\min_{\pi \in \Pi} L_{\lambda^\star}(\pi)$, the standard approach is to establish the strong duality, i.e., $J^\star = L^\star$ (Boyd & Vandenberghe, 2004). In the CMDP setting, where $\mathcal{U} = \{P\}$, strong duality has been proven to hold (Altman, 1999; Paternain et al., 2019; 2022). However, proving strong duality for RCMDPs is highly non-trivial compared to CMDPs.

When $\mathcal{U} = \{P\}$, a typical proof strategy is to combine the sum of cost returns $J_{c_0,P}, \ldots, J_{c_N,P}$ in $L_\lambda(\pi)$ into a single return function. For example, when $N = 1$, we have $L_\lambda(\pi) = J_{c',P}(\pi)$, where

---

[4] Strict constraint satisfaction is straightforward by using a slightly stricter threshold $b' := b - \varepsilon$.

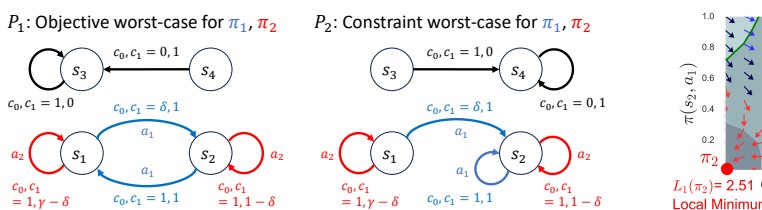

(a) RCMDP presented in Example 1, where $\delta \geq 0$.    (b) $L_1(\pi)$ and policy gradients.

Figure 1: **(a)**: An RCMDP example illustrating the **gradient conflict challenge**. Action labels are omitted when transitions are action-independent. **(b)**: Policy gradients in the example with $(\gamma, \delta, b_1) = (0.4, 0.09, 0)$. Arrows represent the gradient to decrease $L_1(\pi)$. $\pi_2$ attracts policy gradients but is a local minimum since $L_1(\pi_2) > L_1(\pi_1)$, where $\pi_1(\cdot, a_1) = 1$ and $\pi_2(\cdot, a_2) = 1$.

$c' := c_0 + \lambda(c_1 - b_1/H\mathbf{1})$. Since $J_{c',P}(\pi)$ is linearly represented as $J_{c',P}(\pi) = \langle c', d_P^\pi \rangle^5$, it is easy to show $\min_\pi \max_\lambda J_{c',P}(\pi) = \max_\lambda \min_\pi J_{c',P}(\pi)$ by Sion's minimax theorem (Sion, 1958).

However, applying this return-combining strategy to RCMDPs is difficult. For $n \in [\![0, N]\!]$, let $P_n \in \arg\max_{P \in \mathcal{U}} J_{c_n,P}(\pi)$. When $|\mathcal{U}| \neq 1$, $L_\lambda(\pi)$ has a sum of returns $J_{c_0,P_0}(\pi) \dots J_{c_N,P_N}(\pi)$, where $P_0 \neq \dots \neq P_N$ may differ. Thus, $L_\lambda(\pi)$ no longer takes the form of $\langle c', d_P^\pi \rangle$ for a single environment $P$, making well-established duality proof techniques in CMDP literature inapplicable.

(ii) **Gradient conflict challenge.** Unfortunately, even if strong duality holds and $\pi^\star$ can be found by $\pi^\star \in \arg\min_{\pi \in \Pi} L_{\lambda^\star}(\pi)$, solving this minimization remains challenging. Since the sum of robust cost returns in $L_{\lambda^\star}(\pi)$ excludes the use of DP and convex-optimization approaches (Iyengar, 2005; Altman, 1999; Grand-Clément & Petrik, 2024), the policy gradient method such as Equation (1) is the primary remaining option to solve $\min_{\pi \in \Pi} L_\lambda(\pi)$.

In CMDP setting when $|\mathcal{U}| = 1$, the problem $\min_{\pi \in \Pi} L_\lambda(\pi)$ reduces to solving a standard MDP and thus the policy gradient method is ensured to solve $\min_{\pi \in \Pi} L_\lambda(\pi)$ (Agarwal et al., 2021). However, the following Theorem 1 shows that even when $|\mathcal{U}| = 2$, RCMDP can trap the policy gradient in a local minimum that does not solve $\min_{\pi \in \Pi} L_\lambda(\pi)$:

**Theorem 1.** *For any $\gamma \in (0, 1)$, there exist a $\bar{\lambda} > 0$, a policy $\bar{\pi} \in \Pi$ and an RCMDP with $\mu > \mathbf{0}$ satisfying the following condition: There exists a positive constant $R > 0$ such that, for any $b_1 \in \mathbb{R}$,*

$$L_{\bar{\lambda}}(\bar{\pi}) < L_{\bar{\lambda}}(\pi) \ \forall \pi \in \{\pi \in \Pi \mid \|\pi - \bar{\pi}\|_2 \leq R, \ \pi \neq \bar{\pi}\} \ \text{ but } \ L_{\bar{\lambda}}(\bar{\pi}) \geq \min_{\pi \in \Pi} L_{\bar{\lambda}}(\pi) + \frac{3\gamma H}{16} \ . \quad (4)$$

*Moreover, there exists a $b_1 \in (0, H)$ where $\bar{\lambda}$ satisfies $\bar{\lambda} \in \arg\max_{\lambda \in \mathbb{R}_+^N} \min_{\pi \in \Pi} L_\lambda(\pi)$.*

The detailed proof is deferred to Appendix G. Essentially, the proof constructs a simple RCMDP where **the policy gradients for the objective and the constraint are in conflict**.

**Example 1.** Consider the RCMDP with $\mathcal{U} = \{P_1, P_2\}$ presented in Figure 1a with $\delta = 0$, $\mu = 0.25\mathbf{1}$, and set $\lambda = 1$ for simplicity. Let $\pi_1$ and $\pi_2$ be policies that select $a_1$ and $a_2$ for all states, respectively. For both policies, the objective worst-case is $P_1$ and the constraint worst-case is $P_2$ (see Appendix G). Hence, switching from policy $\pi_2$ to taking action $a_1$ decreases the objective return in $P_1$ but increases the constraint return in $P_2$. This conflict causes the gradients of $\pi_2$ for the objective ($\nabla J_{c_0,P_1}(\pi_2)$) and for the constraint ($\nabla J_{c_1,P_2}(\pi_2)$) to sum to a constant vector, i.e.,

$$(\nabla L_1(\pi_2))(s, \cdot) = (\nabla J_{c_0,P_1}(\pi_2) + \nabla J_{c_1,P_2}(\pi_2))(s, \cdot) = \text{constant} \cdot \mathbf{1} \quad \forall s \in \mathcal{S} \ , \quad (5)$$

showing that $\pi_2$ is a stationary point. However, $\pi_2$ cannot solve $\min_{\pi \in \Pi} L_1(\pi)$ because $\pi_1$ would clearly result in a smaller $L_1(\pi)$. This stationary point becomes a strict local minimum when $\delta > 0$, where $\pi_2$ slightly prefers $a_2$ over $a_1$ (see Appendix G for details).

Figure 1b computationally illustrates this negative result by plotting the landscape of $L_1(\pi)$ in the RCMDP example across all possible policies for $(\gamma, \delta) = (0.4, 0.09)$. We set $b_1 = 0$ as it does not influence the landscape of $L_1(\pi)$. In this example, $\pi_2$ becomes a local minimum that attracts the policy gradient but fails to solve $\min_{\pi \in \Pi} L_1(\pi)$, as $\pi_1$ achieves $L_\lambda(\pi_1) < L_\lambda(\pi_2)$.

---

[5]With an abuse of notation, here we denote $d_P^\pi(s, a) = \mathbb{E}\left[\sum_{h=0}^\infty \gamma^h \mathbb{1}\{s_h = s, a_h = a\} \mid s_0 \sim \mu, \pi, P\right]$.

## 4 EPIGRAPH FORM OF RCMDP

This section introduces the epigraph form of RCMDP, which overcomes the challenges discussed in Section 3. For a constrained optimization problem of the form $\min_x\{f(x)\mid h(x)\leq 0\}$ with $x\in\mathbb{R}^n$ and $f,h:\mathbb{R}^n\to\mathbb{R}$, its epigraph form is defined as:

$$\min_{x,y} y \quad \text{such that} \quad f(x)\leq y \quad \text{and} \quad h(x)\leq 0 \tag{6}$$

with variables $x\in\mathbb{R}^n$ and $y\in\mathbb{R}$. It is well-known that $(x,y)$ is optimal for Equation (6) if and only if $x$ is optimal for the original problem and $y=f(x)$ (see, e.g., Boyd & Vandenberghe, 2004).

Using Equation (6) and by introducing a new optimization variable $b_0\in[0,H]$, an RCMDP becomes

$$J^\star = \min_{b_0\in[0,H],\pi\in\Pi} b_0 \quad \text{such that} \quad \underbrace{J_{c_0,\mathcal{U}}(\pi)\leq b_0}_{\text{constraint to minimize objective}} \quad \text{and} \quad \underbrace{J_{c_n,\mathcal{U}}(\pi)\leq b_n\ \forall n\in[\![1,N]\!]}_{\text{constraints for }\pi\in\Pi_F} . \tag{7}$$

Intuitively, Equation (7) seeks the smallest objective threshold $b_0$ such that a feasible policy $\pi\in\Pi_F$ exists with its cumulative objective cost satisfying $J_{c_0,\mathcal{U}}(\pi)\leq b_0$.

We transform Equation (7) into a more convenient form. Define $\Delta_{b_0}:\mathbb{R}^{SA}\to\mathbb{R}$ such that

$$\Delta_{b_0}(\pi) = \max_{n\in[\![0,N]\!]} J_{c_n,\mathcal{U}}(\pi) - b_n \quad \forall\pi\in\Pi , \tag{8}$$

which denotes the maximum violation of the constraints $\max_{n\in[\![1,N]\!]} J_{c_n,\mathcal{U}}(\pi) - b_n \leq 0$ with the additional constraint $J_{c_0,\mathcal{U}}(\pi) - b_0 \leq 0$. By moving $\min_{\pi\in\Pi}$ in Equation (7) to its constraint and using $\Delta_{b_0}(\pi)$, Equation (7) can be transformed as follows:

**Theorem 2.** *Let* $\Delta_{b_0}^\star := \min_{\pi\in\Pi}\Delta_{b_0}(\pi)$, *where* $\Delta_{b_0}(\pi)$ *is defined in Equation* (8). *Then,*

$$(\textbf{Epigraph Form}) \quad J^\star = \min_{b_0\in[0,H]} b_0 \ \text{such that} \ \Delta_{b_0}^\star \leq 0 . \tag{9}$$

*Furthermore, if* $b_0 = J^\star$, *any* $\pi\in\arg\min_{\pi\in\Pi}\Delta_{b_0}(\pi)$ *is optimal.*

The proof is provided in Appendix H.2. Instead of Equation (7), we call Equation (9) the epigraph form of RCMDP. Since the epigraph form provides $J^\star$ and $\pi^\star$, it overcomes the $\pi^\star$ **solution challenge** discussed in Section 3. The remaining task is to develop an algorithm to solve Equation (9).

## 5 EPiRC-PGS ALGORITHM

According to Theorem 2, we can solve an RCMDP by first identifying the optimal return value $b_0 = J^\star$ and then solving $\min_{\pi\in\Pi}\Delta_{b_0}(\pi)$. Our **Epi**graph **R**obust **C**onstrained **P**olicy **G**radient **S**earch (`EpiRC-PGS`) algorithm implements these two steps using a double-loop structure: (i) an outer loop that determines $b_0 = J^\star$ through a bisection search over $b_0 \in [0,H]$ (Section 5.1), and (ii) an inner loop that solves $\min_{\pi\in\Pi}\Delta_{b_0}(\pi)$ using a policy gradient subroutine (Section 5.2).

Note that without any assumptions about the uncertainty set $\mathcal{U}$, solving an RCMDP is NP-hard (Wiesemann et al., 2013). However, imposing concrete structures on $\mathcal{U}$ can restrict the applicability of `EpiRC-PGS`. To enable `EpiRC-PGS` to handle a broader class of $\mathcal{U}$, we consider $\mathcal{U}$ where we can approximate the robust cost return value $J_{c_n,\mathcal{U}}(\pi)$ and its subgradient $\partial J_{c_n,\mathcal{U}}(\pi)$ as follows:

**Assumption 1** (Robust policy evaluator)**.** For each $n\in[\![0,N]\!]$, we have an algorithm $\widehat{J}_n:\Pi\to\mathbb{R}$ such that $|\widehat{J}_n(\pi) - J_{c_n,\mathcal{U}}(\pi)|\leq\varepsilon_{\text{est}}$ for any $\pi\in\Pi$, where $\varepsilon_{\text{est}}\geq 0$.

**Assumption 2.** $\mathcal{U}$ is either (i) a finite set or (ii) a compact set such that, for any $\pi\in\Pi$, $\nabla J_{c_n,P}(\pi)$ is continuous with respect to $P\in\mathcal{U}$.

**Assumption 3** (Subgradient evaluator)**.** For each $n\in[\![0,N]\!]$, we have an algorithm $\widehat{J}_n^\partial:\Pi\to\mathbb{R}^{SA}$ such that $\min_{g\in\partial J_{c_n,\mathcal{U}}(\pi)}\|\widehat{J}_n^\partial(\pi) - g\|_2 \leq \varepsilon_{\text{grd}}$ for any $\pi\in\Pi$, where $\varepsilon_{\text{grd}}\geq 0$.

Assumptions 1, 2 and 3 are satisfied for most tractable uncertainty sets, such as finite, ball (Kumar et al., 2024), $R$-contamination (Wang & Zou, 2022), $L_1$, $\chi^2$, and Kullback–Leibler (KL) sets (Yang et al., 2022). For these tractable uncertainty sets $\mathcal{U}$, the robust policy evaluator ($\widehat{J}_n$) and subgradient evaluator ($\widehat{J}_n^\partial$) can be efficiently implemented using robust DP methods (Iyengar, 2005; Kumar et al., 2022; 2024; Wang & Zou, 2022). As concrete examples, we provide detailed implementations of $\widehat{J}_n$ and $\widehat{J}_n^\partial$ for finite and KL sets in Appendix C. We assumed Assumption 2 because the envelope theorem (Lemmas 9 and 10), together with this assumption, guarantees that $\partial J_{c_n,\mathcal{U}}(\pi)$ is well-defined.

---

**Algorithm 1** Bisection Search with $\min_{\pi \in \Pi} \Delta_{b_0}(\pi)$ Subroutine
(also referred to as `EpiRC-PGS` when using Algorithm 2 as the subroutine)

---

1: **Input:** Iteration length $K \in \mathbb{N}$, evaluator $\widehat{J}_n$ and subroutine $\mathscr{A}$ (see Assumptions 1 and 4)
2: Initialize the search space: $i^{(0)} := 0$ and $j^{(0)} := H$
3: **for** $k = 0, \cdots, K - 1$ **do**
4:     $\pi^{(k)} := \mathscr{A}(b_0^{(k)})$ where $b_0^{(k)} := (i^{(k)} + j^{(k)})/2$        // Compute policy by subroutine
5:     $\widehat{\Delta}^{(k)} := \max_{n \in [\![0,N]\!]} \widehat{J}_n(\pi^{(k)}) - b_n$ where $b_0 = b_0^{(k)}$        // Robust policy evaluation
6:     Compute $i^{(k+1)}$ and $j^{(k+1)}$ by Equation (11) using $\widehat{\Delta}^{(k)}$        // Update search space
7: **end for**
8: **return** $\pi_{\mathrm{ret}}$ computed by $\mathscr{A}(j^{(K)})$

---

## 5.1 BISECTION SEARCH WITH $\min_{\pi \in \Pi} \Delta_{b_0}(\pi)$ SUBROUTINE

This section describes the outer loop of `EpiRC-PGS` to identify $b_0 = J^\star$. The outer loop utilizes the following monotonicity of $\Delta_{b_0}^\star$ for the identification. The proof is deferred to Appendix H.1:

**Lemma 2.** $\Delta_{b_0}^\star$ *is monotonically decreasing in* $b_0$ *and* $\Delta_{J^\star}^\star = 0$.

Thanks to this monotonicity of $\Delta_{b_0}^\star$, **if $\Delta_{b_0}^\star$ can be efficiently computed, a line search over $b_0 \in [0, H]$ will readily find** $b_0 = J^\star$. Increase $b_0$ if $\Delta_{b_0}^\star > 0$, and decrease it if $\Delta_{b_0}^\star \leq 0$. Figure 2 summarizes this idea to solve the epigraph form.

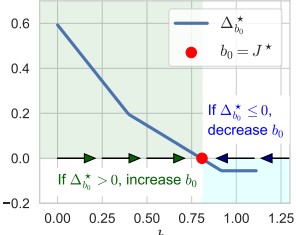

Figure 2: Algorithmic idea to find $b_0 = J^\star$ in Example 1 with $(\gamma, \delta, b_1) = (0.1, 0, 2/3)$.

To implement this idea, let us assume for now that we have a subroutine algorithm $\mathscr{A}$ that computes $\Delta_{b_0}^\star = \min_{\pi \in \Pi} \Delta_{b_0}(\pi)$ with sufficient accuracy. We will implement $\mathscr{A}$ in Section 5.1.

**Assumption 4** (Subroutine algorithm). We have an algorithm $\mathscr{A} : \mathbb{R}_+ \to \Pi$ that takes a value $b_0 \geq 0$ and returns $\pi \in \Pi$ such that $\Delta_{b_0}(\pi) \leq \min_{\pi' \in \Pi} \Delta_{b_0}(\pi') + \varepsilon_{\mathrm{opt}}$, where $\varepsilon_{\mathrm{opt}} \geq 0$.

Using this subroutine $\mathscr{A}$, the outer loop conducts a bisection search over $b_0 \in [0, H]$. Let $K \in \mathbb{N}$ be the number of iterations. For each iteration $k$, let $[i^{(k)}, j^{(k)}] \subseteq [0, H]$ be the search space where $i^{(k)} \leq j^{(k)}$. Set $i^{(0)} = 0$ and $j^{(0)} = H$, and define $b_0^{(k)} := (i^{(k)} + j^{(k)})/2$. Additionally, given $b_0^{(k)}$, we denote the returned policy from $\mathscr{A}$ as $\pi^{(k)} := \mathscr{A}(b_0^{(k)})$ and its estimated $\Delta$ value as

$$\widehat{\Delta}^{(k)} := \max_{n \in [\![0,N]\!]} \widehat{J}_n(\pi^{(k)}) - b_n \text{ where } b_0 = b_0^{(k)} . \tag{10}$$

Based on Figure 2, our bisection search increases $b_0^{(k)}$ if $\widehat{\Delta}^{(k)} > 0$; otherwise, it decreases $b_0^{(k)}$:

$$i^{(k+1)} := \begin{cases} b_0^{(k)} & \text{if } \widehat{\Delta}^{(k)} > 0 \\ i^{(k)} & \text{otherwise} \end{cases} \quad \text{and} \quad j^{(k+1)} := \begin{cases} j^{(k)} & \text{if } \widehat{\Delta}^{(k)} > 0 \\ b_0^{(k)} & \text{otherwise} \end{cases} \tag{11}$$

We summarize the pseudocode of this bisection search in Algorithm 1. The following Theorem 3 ensures that Algorithm 1 returns a near-optimal policy. We provide the proof in Appendix H.5.

**Theorem 3.** *Suppose that Algorithm 1 is run with algorithms $\widehat{J}_n$ and $\mathscr{A}$ that satisfy Assumptions 1 and 4. Then, Algorithm 1 returns an $\tilde{\varepsilon}$-optimal policy, where $\tilde{\varepsilon} := 2(\varepsilon_{\mathrm{opt}} + \varepsilon_{\mathrm{est}}) + 2^{-K} H$.*

## 5.2 SUBROUTINE ALGORITHM TO SOLVE $\min_{\pi \in \Pi} \Delta_{b_0}(\pi)$

The remaining task is to implement the subroutine $\mathscr{A}$ which satisfies Assumption 4. In other words, for a given $b_0$, we need to solve the following auxiliary problem:

$$\textbf{(Epigraph's Auxiliary Problem)} \quad \min_{\pi \in \Pi} \Delta_{b_0}(\pi) = \min_{\pi \in \Pi} \max_{n \in [\![0,N]\!]} \max_{P \in \mathcal{U}} J_{c_n, P}(\pi) - b_n . \tag{12}$$

The right-hand side of Equation (12) can be seen as an RMDP with additional robustness over the set of modified cost functions $\mathcal{C}_{b_0} := \{c_n - b_n/H\mathbf{1}\}_{n \in [\![0,N]\!]}$. Note that since $\mathcal{C}_{b_0}$ is not a rectangular set,

---

**Algorithm 2** Projected Policy Gradient Subroutine

---

1: **Input:** Threshold parameter $b_0 \geq 0$, learning rate $\alpha > 0$, iteration length $T \in \mathbb{N}$, policy evaluator $\widehat{J}_n$ and subgradient evaluator $\widehat{J}_n^\partial$ (see Assumptions 1 and 3)
2: Set an arbitrary initial policy $\pi^{(0)} \in \Pi$
3: **for** $t = 0, \cdots, T - 1$ **do**
4:    $n^{(t)} \in \arg\max_{n \in [\![0,N]\!]} \widehat{J}_n(\pi^{(t)}) - b_n$.        // Select the most violated constraint
5:    $\pi^{(t+1)} := \mathrm{Proj}_\Pi(\pi^{(t)} - \alpha g^{(t)})$ where $g^{(t)} := \widehat{J}_{n^{(t)}}^\partial(\pi^{(t)})$        // Update policy
6: **end for**
7: **return** $\pi^{(t^\star)}$ where $t^\star \in \arg\min_{t \in [\![0,T-1]\!]} \widehat{\Delta}^{(t)}$ and $\widehat{\Delta}^{(t)} := \max_{n \in [\![0,N]\!]} \widehat{J}_n(\pi^{(t)}) - b_n$

---

even if $\mathcal{U}$ is rectangular, the combination $\mathcal{C}_{b_0} \times \mathcal{U}$ does not retain rectangularity. As a result, existing RMDP algorithms designed for rectangular sets, such as DP (Iyengar, 2005), natural policy gradient (Li et al., 2022), and convex optimization (Grand-Clément & Petrik, 2024), are inapplicable.

Due to this non-rectangularity issue, we employ the projected policy gradient method, together with the following subgradient of $\partial\Delta_{b_0}(\pi)$. The proof is deferred to Appendix H.3:

**Lemma 3.** *Define $\mathcal{G}_{b_0}(\pi) := \left\{ \nabla J_{c_n,P}(\pi) \mid n, P \in \arg\max_{(n,P) \in [\![0,N]\!] \times \mathcal{U}} J_{c_n,P}(\pi) - b_n \right\}$ for any $b_0 \in \mathbb{R}$ and $\pi \in \Pi$. Let $\mathrm{conv}\, B$ denote the convex hull of a set $B \subset \mathbb{R}^{SA}$. Under Assumption 2, for any $\pi \in \Pi$ and $b_0 \in \mathbb{R}$, the subgradient of $\Delta_{b_0}(\cdot)$ at $\pi$ is given by $\partial\Delta_{b_0}(\pi) = \mathrm{conv}\, \mathcal{G}_{b_0}(\pi)$.*

We implement the subroutine $\mathscr{A}(b_0)$ based on this subgradient lemma. Starting from an arbitrary policy $\pi^{(0)}$, let $\pi^{(1)}, \ldots, \pi^{(T)}$ be the updated policies where $T \in \mathbb{N}$ is the iteration length. Using the evaluators $\widehat{J}_n, \widehat{J}_n^\partial$, and a learning rate $\alpha > 0$, for a given $b_0$, our subroutine updates policy as follows:

$$\pi^{(t+1)} := \mathrm{Proj}_\Pi(\pi^{(t)} - \alpha g^{(t)}) \text{ where } g^{(t)} := \widehat{J}_{n^{(t)}}^\partial(\pi^{(t)}) \text{ and } n^{(t)} \in \arg\max_{n \in [\![0,N]\!]} \widehat{J}_n(\pi^{(t)}) - b_n .$$
(13)

We summarize the pseudocode of this policy update subroutine in Algorithm 2.

**Remark 1** (Comparison to Lagrange). Recall Equation (5) that the subgradient of the Lagrangian's auxiliary problem, $\partial L_\lambda(\pi)$, involves a summation of policy gradients over different environments. On the other hand, Equation (13) focuses on the policy gradient of a single worst-case environment by taking $\max_{n \in [\![0,N]\!]}$. Intuitively, our policy update avoids the sum of conflicting policy gradients, thereby circumventing the **gradient conflict challenge** discussed in Section 3. Indeed, when the initial distribution satisfies the following coverage assumption[6], **there is no local minimum in** $\Delta_{b_0}(\pi)$:

**Assumption 5** (Initial distribution coverage). The initial distribution $\mu \in \mathscr{P}(\mathcal{S})$ satisfies $\mu > \mathbf{0}$.

**Theorem 4** (Optimality of stationary points). *Under Assumptions 2 and 5, for any $(\pi, b_0) \in \Pi \times \mathbb{R}$,*

$$\Delta_{b_0}(\pi) - \min_{\pi' \in \Pi} \Delta_{b_0}(\pi') \leq DH \max_{\pi' \in \Pi} \langle \pi - \pi', g \rangle \quad \forall g \in \partial\Delta_{b_0}(\pi) ,$$

*where $D := \max_{n,P \in [\![0,N]\!] \times \mathcal{U}} \left\| d_P^{\pi_{n,P}^\star} / \mu \right\|_\infty$ with $\pi_{n,P}^\star \in \arg\min_{\pi' \in \Pi} J_{c_n,P}(\pi')$.*

The detailed proof can be found in Appendix H.4. Our proof is similar to **Theorem 3.2** in Wang et al. (2023), but it is more rigorous and corrects a crucial error that can invalidate their result[7]. Moreover, while their proof is limited to cases where $\arg\max_{P \in \mathcal{U}} J_{c_0,P}(\pi)$ is finite, ours is not. We leverage Sion's minimax theorem (Sion, 1958) for this refinement.

Thanks to the optimality of stationary points of $\Delta_{b_0}(\pi)$ (Theorem 4), Algorithm 2 is guaranteed to solve $\min_{\pi \in \Pi} \Delta_{b_0}(\pi)$ and satisfies the requirement of $\mathscr{A}(b_0)$ in Assumption 4 as follows:

**Theorem 5.** *Suppose Assumptions 1, 2, 3 and 5 hold. Then, there exist problem-dependent constants $C_\partial, C_J, C_\alpha, C_T > 0$ that do not depend on $\varepsilon$ such that, when Algorithm 2 is run with $\alpha = C_\alpha \varepsilon^2$ and $T = C_T \varepsilon^{-4}$, if the evaluators are sufficiently accurate such that $\varepsilon_{\mathrm{grd}} = C_\partial \varepsilon^2$ and $\varepsilon_{\mathrm{est}} = C_J \varepsilon^2$, Algorithm 2 returns a policy $\pi^{(t^\star)}$ satisfying $\Delta_{b_0}(\pi^{(t^\star)}) - \min_{\pi \in \Pi} \Delta_{b_0}(\pi) \leq \varepsilon$.*

---

[6] Such coverage assumption is necessary to ensure the global convergence of policy gradient methods (Mei et al., 2020). Additionally, note that the Lagrange performs poorly even under Assumption 5 (see Theorem 1).

[7] For example, their proof around **Equation (32)** incorrectly bounds a positive value by a negative value.

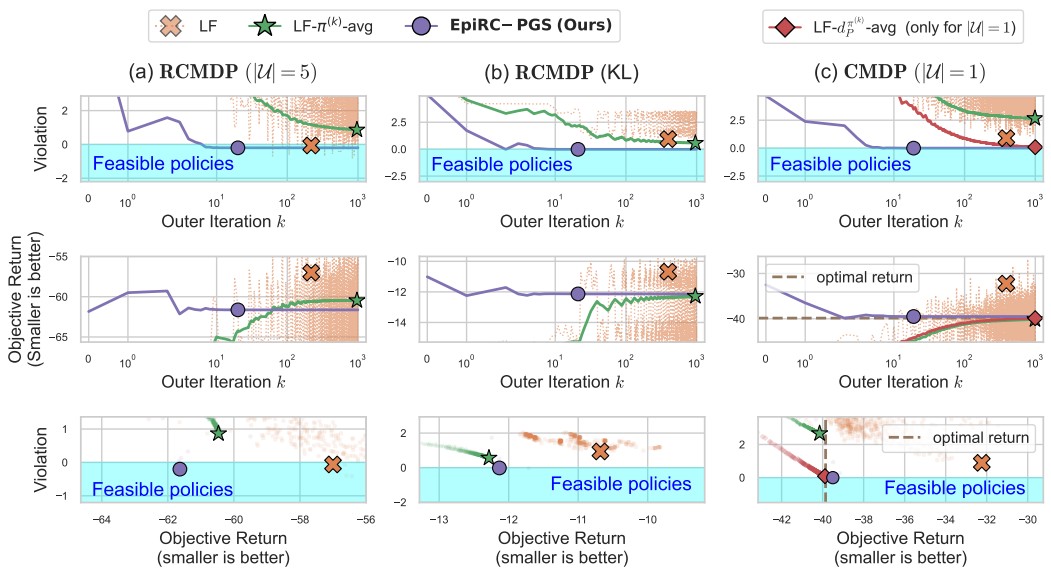

Figure 3: Comparison of the algorithms in different settings **(a)**, **(b)**, and **(c)**, defined in Section 6. The feasible $\pi^{(k)}$ with the smallest return is marked; if none is feasible, the one with the smallest violation is marked. In all the settings, `EpiRC-PGS` quickly identifies a feasible and low-return policy (●). **Top row**: Constraint violation (y-axis: $J_{c_1,\mathcal{U}}(\pi^{(k)}) - b_1$). Policies in the blue area satisfy the constraints. **Middle row**: Objective return relative to the uniform policy (y-axis: $J_{c_0,\mathcal{U}}(\pi^{(k)}) - J_{c_0,\mathcal{U}}(\pi_{\text{unif}})$). Negative values indicate that the policies achieve non-trivial low cumulative objective cots. **Bottom row**: Constraint violation vs. relative objective return.

We provide the proof and the concrete values of $C_\partial, C_J, C_\alpha$ and $C_T$ in Appendix H.6. The proof is primarily based on the weakly convex function analysis (Beck, 2017; Wang et al., 2022).

**Remark 2** (Best-iterate convergence). Lagrangian-based CMDP algorithms typically rely on averaging past policies (e.g., Li et al. (2024); Liu et al. (2021)), which can be impractical in memory-intensive settings, such as deep RL applications. In contrast, Theorem 5 avoids policy averaging and guarantees that the best policy among $\{\pi^{(0)}, \ldots, \pi^{(T-1)}\}$ converges to the solution.

### 5.3 COMBINING BISECTION SEARCH WITH THE POLICY GRADIENT SUBROUTINE

Finally, we combine Algorithm 1 with Algorithm 2 subroutine and refer to the combination as `EpiRC-PGS`. According to Theorems 3 and 5, `EpiRC-PGS` is ensured to find an $\varepsilon$-optimal policy:

**Corollary 1.** *Consider the same settings and notations as in Theorem 5. Set Algorithm 2 as the subroutine $\mathscr{A}$ with parameters $(\widehat{J}_n, \widehat{J}_n^\partial, \alpha, T)$, where we set $\alpha = C_\alpha \varepsilon^2/4$, $T = 16C_T\varepsilon^{-4}$. Then, given inputs $\widehat{J}_n$ and $\mathscr{A}$, Algorithm 1 returns an $\varepsilon$-optimal policy after $K = \lfloor \log(2H\varepsilon^{-1}) \rfloor$ iterations.*

**Remark 3** (Computational complexity). `EpiRC-PGS` outputs an $\varepsilon$-optimal policy by querying $\widehat{J}_n$ and $\widehat{J}_n^\partial$ a total of $\widetilde{\mathcal{O}}((N+1)KT)$ times. Thus, the computational complexity of `EpiRC-PGS` can be expressed as $\widetilde{\mathcal{O}}((N+1)KT \times [\text{querying cost}])$. As a simple example, consider the case where $\mathcal{U}$ is finite, where querying $\widehat{J}_n(\pi)$ and $\widehat{J}_n^\partial(\pi)$ require $\widetilde{\mathcal{O}}(S^2A|\mathcal{U}|)$ operations Kumar et al. (2024). Using the concrete value of $KT$, the computational complexity of `EpiRC-PGS` for finite $\mathcal{U}$ becomes $\widetilde{\mathcal{O}}(D^4S^5A^4H^{14}|\mathcal{U}|(N+1)\varepsilon^{-4})$. Similar analyses can be applied to other types of uncertainty sets.

## 6 EXPERIMENTS

To support the theoretical guarantees of `EpiRC-PGS` and demonstrate the limitations of the Lagrangian methods in identifying near-optimal policies, this section empirically evaluates `EpiRC-PGS` in three settings with five constraints ($N = 5$):

**(a)** RCMDP with $\mathcal{U} = \{P_1, \ldots, P_5\}$, where each environment $P.$ is randomly generated.

**(b)** RCMDP with a $(s, a)$-rectangular KL uncertainty set (Iyengar, 2005), which considers $\mathcal{U} = \times_{s,a}\{p \in \mathscr{P}(\mathcal{S}) \mid \mathrm{KL}[p \,\|\, P(\cdot \mid s, a)] \leq C_{\mathrm{KL}}\}^8$ where $P$ is a nominal environment.

**(c)** CMDP, which is equivalent to RCMDP with $\mathcal{U} = \{P\}$.

**Environment construction.** In **(a)**, **(b)**, and **(c)**, we randomly generate environments following the experimental setup of Dann et al. (2017). For each state-action pair $(s, a)$, transition probabilities $P(\cdot \mid s, a)$ are independently sampled from a $\mathrm{Dirichlet}(0.1, \ldots, 0.1)$ distribution. This produces a concentrated yet non-deterministic transition model, resembling the widely used GARNET benchmark with a *branching factor* of 1 (Archibald et al., 1995). Cost values $c_n(s, a)$ are assigned as 0 with probability 0.1 and 1 otherwise. Initial state probabilities $\mu(\cdot)$ are sampled from a $\mathrm{Dirichlet}(0.5, \ldots, 0.5)$ distribution. Constraint thresholds $b_1, \ldots, b_5$ are set to ensure the existence of a feasible policy. Additional environmental parameters are described in Appendix D.

**Baseline algorithms.** We compare `EpiRC-PGS` with a Lagrangian counterpart, denoted `LF`, which abstracts most existing Lagrangian-based algorithms for RCMDPs (e.g., Russel et al., 2020; Wang et al., 2022). `LF` aims to solve the problem $\max_{\lambda \in \mathbb{R}_+^N} \min_{\pi \in \Pi} L_\lambda(\pi)$ in Equation (3) by performing gradient ascent on $\lambda$ while using a policy gradient subroutine to solve $\min_{\pi \in \Pi} L_\lambda(\pi)$.

We also evaluate the averaged policies generated by `LF`, defined as $\frac{1}{k} \sum_{j=0}^{k} \pi^{(k)}$. Such policy averaging is employed in Lagrangian methods (Miryoosefi et al., 2019; Zhang et al., 2024), though it often lacks theoretical guarantees (see Appendix A.4). In the CMDP setting **(c)**, we further evaluate the averaged occupancy measures, where the $k$-th policy is derived from $\frac{1}{k} \sum_{j=0}^{k} d_P^{\pi^{(k)}}$. Averaging $d_P^{\pi^{(k)}}$ is ensured to identify a near-optimal policy (Zahavy et al., 2021), but is well-defined only when $|\mathcal{U}| = 1$. We refer to these two averagings as `LF-`$\pi^{(k)}$`-avg` and `LF-`$d_P^{\pi^{(k)}}$`-avg`, respectively. Moreover, **(c)** reports the optimal return value computed via an LP method (Altman, 1999). The detailed implementation of algorithms are provided in Appendix D.

**Results.** Figure 3 illustrates the performance of the algorithms averaged over 10 random seeds. Across all settings, `EpiRC-PGS` rapidly converges to a feasible policy with a low objective cost return, while both `LF` and `LF-`$\pi^{(k)}$`-avg` fail to identify feasible policies in certain settings (e.g., `LF-`$\pi^{(k)}$`-avg` in **(a)** and `LF` in **(b)**). In the CMDP **(c)**, `EpiRC-PGS` and `LF-`$d_P^{\pi^{(k)}}$`-avg` converge to a near-optimal policy, but `LF-`$\pi^{(k)}$`-avg` does not. These results empirically validate that `EpiRC-PGS` yields a near-optimal policy in RCMDPs, contrasting with the conventional Lagrangian-based algorithm's inability in robust settings.

## 7 CONCLUSION AND LIMITATIONS

In this work, we propose `EpiRC-PGS`, the first algorithm guaranteed to find a near-optimal policy in an RCMDP (Corollary 1). At its core, `EpiRC-PGS` leverages the epigraph form of RCMDPs, which not only ensures the optimal policy $\pi^\star$ can be obtained (Section 4) but also enables a policy gradient algorithm to efficiently find it (Theorem 4). These features address the optimization challenges encountered in the conventional Lagrangian formulation (Section 3).

**Limitations and future work.** A double-loop algorithm like `EpiRC-PGS` is often inefficient when the inner problem requires high computational cost (Lin et al., 2024). Developing a single-loop algorithm is a promising direction for future research, and we discuss the challenges in Appendix B.

Another research avenue is improving the iteration complexity of our $\widetilde{\mathcal{O}}(\varepsilon^{-4})$. This may not be tight, since for RMDPs with $(s, a)$-rectangularity, the natural policy gradient method is ensured to find an $\varepsilon$-optimal policy with $\widetilde{\mathcal{O}}(\varepsilon^{-2})$ iterations Li et al. (2022).

Finally, the coverage assumption on the initial distribution (Assumption 5) is not necessary in CMDPs (Ding et al., 2024). We leave the removal of Assumption 5 in RCMDPs for future work.

---

[8]$\mathrm{KL}[p \,\|\, q] = \sum_{s \in \mathcal{S}} p(s) \ln p(s)/q(s)$ represents the KL divergence between two probability distributions $p > \mathbf{0}$ and $q > \mathbf{0}$ defined over $\mathcal{S}$. $C_{\mathrm{KL}} > 0$ is a positive constant.

ACKNOWLEDGMENTS

This work is supported by JST Moonshot R&D Program Grant Number JPMJMS2236. We thank Arnob Ghosh for his advice on improving the expression of assumptions in this paper.

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

# Contents

## A  ADDITIONAL RELATED WORK

This section reviews existing approaches for CMDPs (Appendix A.1), RMDPs (Appendix A.2), and RCMDPs (Appendix A.3). It also highlights their inherent limitations and the challenges they face when applied to RCMDPs.

### A.1  CONSTRAINED MARKOV DECISION PROCESSES

CMDP is a specific subclass of RCMDP where the uncertainty set consists of a single element, i.e., $\mathcal{U} = \{P\}$. This section describes the primary approaches to the CMDP problem: the linear programming (LP) approach, the Lagrangian approach, and the epigraph approach.

**Linear programming approach.**  The LP approach has been extensively studied in the theoretical literature (Efroni et al., 2020; Liu et al., 2021; Bura et al., 2022; HasanzadeZonuzy et al., 2021; Zheng & Ratliff, 2020). Although it is a fundamental method in CMDP, it is less popular in practice due to its difficulty in scaling to high-dimensional problem settings, such as those encountered in deep RL. Additionally, incorporating environmental uncertainty into the LP approach for CMDPs is challenging. The LP approach utilizes the fact that the return minimization problem of an MDP can be formulated as a convex optimization problem with respect to the occupancy measure (Altman, 1999; Nachum & Dai, 2020). However, RMDPs do not permit a convex formulation in terms of occupancy measures (Iyengar, 2005; Grand-Clément & Petrik, 2024). While Grand-Clément & Petrik (2024) recently introduced a convex optimization approach for RMDPs, their formulation is convex for the transformed objective value function, not for the occupancy measure, making it challenging to incorporate constraints as seen in RCMDPs.

**Lagrangian approach.**  The Lagrangian approach is perhaps the most popular approach to CMDPs both in theory (Ding et al., 2020; Wei et al., 2021; HasanzadeZonuzy et al., 2021; Kitamura et al., 2024) and practice (Achiam et al., 2017; Tessler et al., 2018; Wang et al., 2022; Le et al., 2019; Russel et al., 2020). This popularity stems from its compatibility with policy gradient methods, making it readily extendable to deep RL. The Lagrangian approach benefits from the strong duality in CMDPs. When $\mathcal{U}$ consists of a single element, it is well established that strong duality holds, meaning that $L^\star = J^\star$ holds, where $L^\star$ is from Equation (3) and $J^\star$ is from Equation (2) (Altman, 1999; Paternain et al., 2019; 2022).

The challenge with the Lagrangian method is the identification of an optimal policy. Even if Equation (3) is solved, there's no guarantee that the solution to the inner minimization problem will represent an optimal policy. In some CMDPs, where feasible policies in $\Pi_\mathrm{F}$ must be stochastic (Altman, 1999), the inner minimization may yield a deterministic solution that is infeasible. Zahavy et al. (2021); Miryoosefi et al. (2019); Chen et al. (2021); Li et al. (2024); Liu et al. (2021) addressed this challenge by averaging policies (or occupancy measures) obtained during the optimization process. However, policy averaging can be impractical for large-scale algorithms (e.g., deep RL) because it necessitates storing all past policies, which is often infeasible. On the other hand, Ying et al. (2022); Ding et al. (2024); Müller et al. (2024); Kitamura et al. (2024) tackled the issue by introducing entropy regularization into the objective return. However, the regularization can lead to biased solutions and result in a policy design that may deviate from what is intended by the cost function.

In contrast, `EpiRC-PGS` requires neither policy averaging nor regularization, thereby offering advantageous properties even in CMDP settings.

**Epigraph approach.**  Few studies have investigated the epigraph form in the CMDP setting. So & Fan (2023) proposed a deep RL algorithm aimed at system stabilization under constraints, and So et al. (2024) developed a deep RL algorithm for goal-reaching tasks with risk-avoidance constraints. Although these studies empirically demonstrated the effectiveness of the epigraph form in constrained RL problems, they did not establish theoretical guarantees, such as global convergence. On the other hand, this paper is the first work to provide theoretical guarantees for the epigraph form in CMDPs. Furthermore, unlike existing constrained RL studies, we consider not only constraints but also the robustness against the transition kernel.

## A.2 ROBUST MARKOV DECISION PROCESSES

RMDP is a specific subclass of RCMDP where there are no constraints, i.e., $N = 0$. RMDP is a crucial research area for the practical success of RL applications, where the environmental mismatch between the training phase and the testing phase is almost unavoidable. Without robust policy design, even a small mismatch can lead to poor performance of the trained policy in the testing phase (Li et al., 2022; Jiang, 2018).

**Dynamic programming approach.** Since the seminal work by Iyengar (2005), numerous studies have explored dynamic programming (DP) approaches for RMDPs (Nilim & El Ghaoui, 2005; Clavier et al., 2023; Panaganti & Kalathil, 2022; Mai & Jaillet, 2021; Grand-Clément & Kroer, 2021; Derman et al., 2021; Wang & Zou, 2021; Kumar et al., 2022; Yang et al., 2023). The DP approach decomposes the original problem into smaller sub-problems using Bellman's principle of optimality (Bellman et al., 1957). To apply this principle, DP approaches enforce rectangularity on the uncertainty set, which assumes independent worst-case transitions at each state or state-action pair. However, as pointed out by Goyal & Grand-Clement (2023), the rectangularity assumption can result in a very conservative optimal policy. Moreover, applying DP to constrained settings is challenging since CMDPs typically do not satisfy the principle of optimality (Haviv, 1996). Although several studies have attempted to apply DP to CMDPs, they face issues such as excessive memory consumption, due to the use of non-stationary policy classes, or are restricted to deterministic policy classes (Chang, 2023; Chen & Blankenship, 2004; Chen & Feinberg, 2006).

**Epigraph application to DP approach.** Chen et al. (2019); Wiesemann et al. (2013); Ho et al. (2021) employed the epigraph form to implement a robust DP algorithm for the $s$-rectangular uncertainty set setting. Specifically, they showed that the $s$-rectangular robust Bellman operator, which is the $s$-rectangular counterpart of Equation (19), can be efficiently implemented using the epigraph form. However, since their algorithms rely on Bellman's principle of optimality, similar to standard DP, they are likely to encounter the same challenges in CMDP settings as those discussed above.

**Policy gradient approach.** Another promising approach for RMDPs is the policy gradient method. Similar to the DP approach, most existing policy gradient algorithms also work only under the rectangularity assumption (Kumar et al., 2024; Wang & Zou, 2022; Li et al., 2022), and thus suffer from the same conservativeness issue. It is important to note that robust policy evaluation can be NP-hard without any structural assumptions on the uncertainty set (Wiesemann et al., 2013), but such assumptions are potentially not required for the robust policy optimization step. Our policy gradient algorithm abstracts the evaluation step by Assumption 1 and avoids the need for the rectangularity assumption during the policy optimization phase, similar to the recent work by Wang et al. (2023).

## A.3 ROBUST CONSTRAINED MARKOV DECISION PROCESSES

Russel et al. (2020); Mankowitz et al. (2020) proposed heuristic algorithms for RCMDPs, but their approaches lack theoretical guarantees for convergence to a near-optimal policy. Wang et al. (2022) introduced a Lagrangian approach with convergence guarantee to a stationary point. However, they do not ensure the optimality of this stationary point. Moreover, their method is heavily dependent on the restrictive $R$-contamination set assumption (Du et al., 2018; Wang & Zou, 2021; 2022).

Sun et al. (2024) applied a trust-region method to RCMDPs. The policy is updated to remain sufficiently similar to the previous one, ensuring that performance and constraint adherence do not degrade, even in the face of environmental uncertainty. However, while they ensure that each policy update step maintains performance, convergence to a near-optimal policy is not guaranteed.

Ghosh (2024) employed a penalty approach which considers the optimization problem of the form $\min_\pi f(\pi) + \lambda \max\{h(\pi), 0\}$, where $f$ and $h$ are defined in Section 1. While this approach can yield a near-optimal policy for a sufficiently large value of $\lambda > 0$, the author does not provide a concrete optimization method for the minimization and instead assumes the availability of an oracle to solve it. As we demonstrate in Section 3, this minimization is intrinsically difficult, making it challenging to implement such an oracle.

Finally, Zhang et al. (2024) tackled RCMDPs using the policy-mixing technique (Miryoosefi et al., 2019; Le et al., 2019). In this technique, a policy is sampled from a finite set of deterministic policies

according to a sampling distribution at the start of each episode, and it remains fixed throughout the episode. However, even if a good sampling distribution is determined, there is no guarantee that the resulting expected policy will be optimal due to the non-convexity of the return function with respect to policies (Agarwal et al., 2021). We discuss the limitations of the policy-mixing technique in Appendix A.4. Additionally, Zhang et al. (2024) assume an $R$-contamination uncertainty set, limiting its applicability similarly to the work of Wang et al. (2022).

Although the RCMDP problem remains unsolved, the control theory community has long studied the computation of safe controllers under environmental uncertainties. Notable methods include robust model predictive control (Bemporad & Morari, 2007) and $H_\infty$ optimal control (Anderson et al., 2019; Zames, 1981; Doyle, 1982). These approaches are specifically tailored for a specialized class of MDPs, known as the linear quadratic regulator (LQR, Du et al. (2021)). However, because LQR and tabular MDPs operate within distinct frameworks, these control methods are unsuitable for tabular RCMDPs. Given that most modern RL algorithms, such as DQN (Mnih et al., 2015), are based on the tabular MDP framework, our results bridge the gap between the RL and control theory communities, laying the foundation for the development of reliable RL applications in the future.

### A.4 NOTES ON THE POLICY-MIXING TECHNIQUE

This section explains the theoretical limitations of the policy-mixing technique (Zhang et al., 2024; Miryoosefi et al., 2019; Le et al., 2019) for identifying a near-optimal policy.

**Policy-mixing technique.** Let $\widetilde{\Pi} := \{\pi_1, \ldots, \pi_m\}$ be a finite set of policies with $m \in \mathbb{N}$. Consider a non-robust, single-constraint CMDP $(\mathcal{S}, \mathcal{A}, \gamma, P, \mathcal{C} = \{c_0, c_1\}, b, \mu)$. Given a distribution $\rho \in \mathscr{P}\left(\widetilde{\Pi}\right)$, define

$$\widetilde{J}_{c_0,P}(\rho) := \sum_{\pi \in \widetilde{\Pi}} \rho(\pi) J_{c_0,P}(\pi) \text{ and } \widetilde{J}_{c_1,P}(\rho) := \sum_{\pi \in \widetilde{\Pi}} \rho(\pi) J_{c_1,P}(\pi) .$$

The policy-mixing technique considers the following optimization problem:

$$\widetilde{J}^\star := \min_{\rho \in \mathscr{P}\left(\widetilde{\Pi}\right)} \widetilde{J}_{c_0,P}(\rho) \text{ such that } \widetilde{J}_{c_1,P}(\rho) \le b_1 \tag{14}$$

$$= \min_{\rho \in \mathscr{P}\left(\widetilde{\Pi}\right)} \max_{\lambda \in \mathbb{R}_+} \sum_{\pi \in \widetilde{\Pi}} \rho(\pi)(J_{c_0,P}(\pi) + \lambda(J_{c_1,P}(\pi) - b_1)) =: \min_{\rho \in \mathscr{P}\left(\widetilde{\Pi}\right)} \max_{\lambda \in \mathbb{R}_+} \widetilde{L}(\rho, \lambda) .$$

Let $\rho^\star$ be the solution of Equation (14) such that $\rho^\star \in \arg\min_{\rho \in \mathscr{P}\left(\widetilde{\Pi}\right)} \max_{\lambda \in \mathbb{R}_+} \widetilde{L}(\rho, \lambda)$.

In this setting, a policy is sampled from $\rho$ at the start of each episode and remains fixed throughout the episode. The term $\widetilde{J}_{c_0,P}(\rho)$ represents the expected return under the distribution $\rho$. Since $\widetilde{L}(\rho, \lambda)$ is convex in $\rho$ and concave in $\lambda$, under some mild assumptions, Equation (14) can be solved efficiently by the following standard optimization procedure for min-max problems: At each iteration $t = 1, \ldots, T$, with initial values $\lambda^{(0)} \in \mathbb{R}_+$ and $\rho^{(0)} \in \mathscr{P}\left(\widetilde{\Pi}\right)$,

1. Update $\lambda^{(t)}$ using a no-regret algorithm. For example, with gradient ascent and a learning rate $\alpha > 0$:
$$\lambda^{(t)} := \max\left\{\lambda^{(t-1)} + \alpha\left(\widetilde{J}_{c_1,P}(\rho^{(t-1)}) - b_1\right), 0\right\} .$$

2. Update $\rho^{(t)}$ as $\rho^{(t)}(\pi) = \mathbb{1}\left\{\pi = \pi^{(t)}\right\}$ where
$$\pi^{(t)} \in \arg\min_{\pi \in \widetilde{\Pi}} J_{c_0,P}(\pi) + \lambda^{(t)}(J_{c_1,P}(\pi) - b_1) .$$

Then, the averaged distribution $\bar{\rho}^{(T)} := \frac{1}{T}\sum_{t=0}^{T} \rho^{(t)}$ converges to $\rho^\star$ as $T \to \infty$ (Abernethy & Wang, 2017; Zahavy et al., 2021). When $\widetilde{\Pi}$ is sufficiently large, we can expect that the optimal value of Equation (14) is equivalent to that of the CMDP problem, i.e., $\widetilde{J}^\star = J^\star$, where $J^\star$ is defined in Equation (2) with $\mathcal{U} = \{P\}$.

**Limitation of policy-mixing.** Even when $\widetilde{J}^\star = J^\star$, it is crucial to note that, while $\bar{\rho}^{(T)}$ converges to $\rho^\star$, **there is no guarantee that $\bar{\pi}^{(T)} := \frac{1}{T} \sum_{t=0}^T \pi^{(t)}$ will converge to $\pi^\star$.**

Let $\bar{\lambda}^{(T)} := \frac{1}{T} \sum_{t=0}^T \lambda^{(t)}$. Zhang et al. (2024); Miryoosefi et al. (2019); Le et al. (2019) argued for the convergence of $\bar{\pi}^{(t)}$ by asserting that the equality (a) in the following equation holds:

$$\frac{1}{T} \sum_{t=1}^T \widetilde{L}\left(\rho^{(t)}, \lambda^{(t)}\right) = \frac{1}{T} \sum_{t=1}^T \left( J_{c_0,P}(\pi^{(t)}) + \lambda^{(t)}\left( J_{c_1,P}(\pi^{(t)}) - b_1 \right) \right)$$
$$\overset{(a)}{=} J_{c_0,P}(\bar{\pi}^{(T)}) + \bar{\lambda}^{(T)}\left( J_{c_1,P}(\bar{\pi}^{(T)}) - b_1 \right) \tag{15}$$

(see, for example, **Equation (14)** in Zhang et al. (2024), **Equation (1)** in Le et al. (2019), and around **Equation (13)** in Miryoosefi et al. (2019)).

However, (a) in Equation (15) does not hold in general because the return function is neither convex nor concave in policy. Even when $T = 2$, there is an example where Equation (15) fails (see **Proof of Lemma 3.1** in Agarwal et al. (2021)). This invalidates the results of Miryoosefi et al. (2019); Le et al. (2019); Zhang et al. (2024), thus illustrating the theoretical limitations of the policy-mixing approach for near-optimal policy identification.

## B    DISCUSSION ON SINGLE-LOOP ALGORITHM

Although Algorithm 1 can identify a near-optimal policy, it uses a double-loop structure that repetitively solves $\min_{\pi \in \Pi} \Delta_{b_0}(\pi)$ by Algorithm 2. In practice, single-loop algorithms, such as primal-dual algorithms for CMDPs (e.g., Efroni et al. (2020); Ding et al. (2024)), are typically more efficient and preferable compared to double-loop algorithms. This section discusses the challenge of designing a single-loop algorithm for the epigraph form.

Since the epigraph form is a constrained optimization problem, we can further transform it using a Lagrangian multiplier $\lambda \in \mathbb{R}_+$, yielding:

$$J^\star = \min_{b_0 \in [0,H]} \max_{\lambda \in \mathbb{R}_+} L_{\text{epi}}(b_0, \lambda) \quad \text{where} \quad L_{\text{epi}}(b_0, \lambda) := b_0 + \lambda \Delta_{b_0}^\star . \tag{16}$$

Similar to the typical Lagrangian approach, let's swap the min-max order. We call the resulting formulation the "epigraph-Lagrange" formulation:

$$(\textbf{Epigraph-Lagrange}) \quad L_{\text{epi}}^\star = \max_{\lambda \in \mathbb{R}_+} \min_{b_0 \in [0,H]} \min_{\pi \in \Pi} b_0 + \lambda \Delta_{b_0}(\pi) . \tag{17}$$

Does the strong duality, $J^\star = L_{\text{epi}}^\star$, hold? If it does, we could design a single-loop algorithm similar to primal-dual CMDP algorithms, performing gradient ascent and descent on Equation (17). Unfortunately, proving the strong duality is challenging.

Essentially, the min-max can be swapped when $L_{\text{epi}}(b_0, \lambda)$ in Equation (16) is quasiconvex-quasiconcave (Sion, 1958). While $L_{\text{epi}}(b_0, \lambda)$ is clearly concave in $\lambda$, the quasiconvexity in $b_0$ is not obvious. Although $\Delta_{b_0}^\star$ is decreasing due to Lemma 2 and thus a quasi-convex function, there is no guarantee on the quasi-convexity of $b_0 + \lambda \Delta_{b_0}^\star$. The situation would be resolved if $\Delta_{b_0}^\star$ were convex in $b_0$. However, since $\Delta_{b_0}^\star = \min_{\pi \in \Pi} \Delta_{b_0}(\pi)$ is a pointwise minimum and $\Delta_{b_0}(\pi)$ may not be convex in $\pi$ (Agarwal et al., 2021), $\Delta_{b_0}^\star$ may not be convex in $b_0$ (Boyd & Vandenberghe, 2004).

Therefore, algorithms for the epigraph-Lagrange formulation face a problem similar to the $\boldsymbol{\pi}^\star$ **solution challenge** of the Lagrangian formulation (Section 3). Proving strong duality or finding alternative ways to circumvent this challenge is a promising direction for future RCMDP research.

## C    UNCERTAINTY SETS AND ALGORITHMS FOR $\Delta_{b_0}$ AND SUBGRADIENT EVALUATORS

This section provides examples of uncertainty set structures and algorithms that realize $\widehat{J}_n$ in Assumption 1 and $\widehat{J}_n^\partial$ in Assumption 3. $\widehat{J}_n$ evaluates $J_{c_n,\mathcal{U}}(\pi)$, and $\widehat{J}_n^\partial$ evaluates one of the

---

**Algorithm 3** Evaluators for Finite Uncertainty Set

---

1: **Input:** Policy $\pi$, uncertainty set $\mathcal{U} = \{P_1, \ldots, P_M\}$, and index $n \in [\![0, N]\!]$.
2: **for** $m \in [\![1, M]\!]$ **do**
3: $\quad Q^\pi_{c_n, P_m} := (I - \gamma P_m \Pi^\pi)^{-1} c_n \in \mathbb{R}^{SA}$.
4: $\quad J_{c_n, P_m}(\pi) := \sum_{(s,a) \in \mathcal{S} \times \mathcal{A}} \mu(s) \pi(s,a) Q^\pi_{c_n, P_m}(s,a)$
5: **end for**
6: Let $m^\star \in \arg\max_{m \in [\![1,M]\!]} J_{c_n, P_m}(\pi)$
7: $d^\pi_{P_{m^\star}} = (1 - \gamma) \mu (I - \gamma \Pi^\pi P_{m^\star})^{-1} \in \mathbb{R}^S$
8: **return** (for $\widehat{J}_n$): $J_{c_n, P_{m^\star}}(\pi)$
9: **return** (for $\widehat{J}_n^\partial$): $H d^\pi_{P_{m^\star}}(s) Q^\pi_{c_n, P_{m^\star}}(s,a) \quad \forall(s,a) \in \mathcal{S} \times \mathcal{A}$

---

elements in $\partial J_{c_n, \mathcal{U}}(\pi)$. In this section, we frequently use the following useful matrix formulations (Pirotta et al., 2013): for a cost $c \in \mathbb{R}^{SA}$,

$$Q^\pi_{c,P} = (I - \gamma P \Pi^\pi)^{-1} c \in \mathbb{R}^{SA}$$

$$J_{c,P}(\pi) = \sum_{(s,a) \in \mathcal{S} \times \mathcal{A}} \mu(s) \pi(s,a) Q^\pi_{c,P}(s,a) \in \mathbb{R} \tag{18}$$

$$d^\pi_P = \mu^\top (I - \gamma \Pi^\pi P)^{-1} \in \mathbb{R}^S \,,$$

where $I$ is an identity matrix and $\Pi^\pi$ is a $\mathbb{R}^{\mathcal{S} \times (\mathcal{S} \times \mathcal{A})}$ matrix such that $\Pi^\pi(s, (s,a)) = \pi(s,a)$. Due to Lemma 1, an element in $\partial J_{c_n, \mathcal{U}}(\pi)$ takes the form of:

$$(\nabla J_{c_n, P}(\pi))(s,a) = H d^\pi_P(s) Q^\pi_{c_n, P}(s,a) \; \forall(s,a) \in \mathcal{S} \times \mathcal{A} \,.$$

### C.1 FINITE UNCERTAINTY SET

Let $\mathcal{U}$ be a finite uncertainty set such that $\mathcal{U} = \{P_1, \ldots, P_M\}$ where $M \in \mathbb{N}$. $\mathcal{U}$ clearly satisfies Assumption 2. The implementation of $\widehat{J}_n$ is trivial by Equation (18). The implementation of $\widehat{J}_n^\partial$ is also straightforward due to Lemma 10. Specifically, it can be implemented by the following subgradient representation:

$$\partial J_{c_n, \mathcal{U}} = \text{conv} \left\{ \nabla J_{c_n, P_m}(\pi) \; \middle| \; m \in \arg\max_{m \in [\![1,M]\!]} J_{c_n, P_m}(\pi) \right\} \,.$$

Algorithm 3 summarizes the implementations of $\widehat{J}_n$ and $\widehat{J}_n^\partial$. Our experiment **(a)** in Section 6 uses Algorithm 3.

### C.2 $(s,a)$-RECTANGULAR KL UNCERTAINTY SET

$(s,a)$**-rectangularity.** An uncertainty set $\mathcal{U}$ is called $(s,a)$-*rectangular* if it satisfies:

$$\mathcal{U} = \times_{s,a} \mathcal{U}_{s,a} \text{ where } \mathcal{U}_{s,a} \subseteq \mathscr{P}(\mathcal{S}) \,.$$

For such $(s,a)$-rectangular uncertainty sets, the following **robust DP** update is a widely-used approach to compute the worst-case $Q$-function:

$$\textbf{(Robust DP)} \quad Q^{(t+1)}_{c_n}(s,a) = c_n(s,a) + \gamma \max_{p \in \mathcal{U}_{s,a}} \sum_{s' \in \mathcal{S}} p(s') V^{(t)}_{c_n}(s')$$

$$\text{where } V^{(t)}_{c_n}(s') := \sum_{a' \in \mathcal{A}} \pi(s', a') Q^{(t)}_{c_n}(s', a') \,. \tag{19}$$

By repeatedly applying Equation (19), $Q^{(t)}_{c_n}$ converges linearly to $Q^\pi_{c_n, P^\star_n}$, where $P^\star_n \in \arg\max_{P \in \mathcal{U}} J_{c_n, P}(\pi)$ (see, e.g., **Corollary 2** in Iyengar (2005)).

---

**Algorithm 4** Evaluators for KL Uncertainty Set

---

1: **Input:** Policy $\pi$, nominal transition kernel $P$, regularization parameter $C'_{\mathrm{KL}} > 0$, and index $n \in [\![0, N]\!]$.
2: Repeat Equation (25) and compute its fixed point $Q_{c_n}^{(\infty)}$.
3: $P_n^\star(\cdot \mid s, a) \propto P(\cdot \mid s, a) \exp\left(V_{c_n}^{(\infty)}(\cdot)/C'_{\mathrm{KL}}\right)$      // Compute the worst-case environment
4: $Q_{c_n, P_n^\star}^\pi := (I - \gamma P_n^\star \Pi^\pi)^{-1} c_n \in \mathbb{R}^{SA}$.
5: $J_{c_n, P_n^\star}(\pi) := \sum_{(s,a) \in \mathcal{S} \times \mathcal{A}} \mu(s) \pi(s,a) Q_{c_n, P_n^\star}^\pi(s,a)$
6: $d_{P_n^\star}^\pi := (1 - \gamma)\mu(I - \gamma \Pi^\pi P_n^\star)^{-1} \in \mathbb{R}^S$
7: **return (for $\widehat{J}_n$):** $J_{c_n, P_n^\star}(\pi)$
8: **return (for $\widehat{J}_n^\partial$):** $H d_{P_n^\star}^\pi(s) Q_{c_n, P_n^\star}^\pi(s,a)$    $\forall(s,a) \in \mathcal{S} \times \mathcal{A}$

---

Once we have $Q_{c_n, P_n^\star}^\pi$, thanks to the rectangularity, the worst-case environment can be computed by:

$$P_n^\star(\cdot \mid s, a) = \arg\max_{p \in \mathcal{U}_{s,a}} \sum_{s' \in \mathcal{S}} p(s') V_{c_n, P_n^\star}^\pi(s') \quad \forall(s,a) \in \mathcal{S} \times \mathcal{A} . \tag{20}$$

**KL uncertainty set.** Both Equation (19) and Equation (20) require an efficient computation of $\max_{p \in \mathcal{U}_{s,a}} \langle p, v \rangle$ for some vector $v \in \mathbb{R}^S$. The KL uncertainty set is one of the most popular choices, as it allows for efficient computation of this maximization (Iyengar, 2005; Yang et al., 2022).

For some positive constant $C_{\mathrm{KL}} > 0$, if $\mathcal{U}_{s,a}$ satisfies

$$\mathcal{U}_{s,a} = \{p \in \mathscr{P}(\mathcal{S}) \mid \mathrm{KL}[p \,\|\, P(\cdot \mid s,a)] \le C_{\mathrm{KL}}\} ,$$

we call $\mathcal{U}$ a $(s,a)$-rectangular KL uncertainty set. Here, $\mathrm{KL}[p \,\|\, q] = \sum_{s \in \mathcal{S}} p(s) \ln \frac{p(s)}{q(s)}$ denotes the KL divergence between $p \in \mathscr{P}(\mathcal{S})$ and $q \in \mathscr{P}(\mathcal{S})$. Due to Lemma 1, this $\mathcal{U}$ clearly satisfies Assumption 2.

The following lemma is useful for implementing the robust DP update with a KL uncertainty set.

**Lemma 4 (Lemma 4 in Iyengar (2005)).** *Let $v \in \mathbb{R}^S$ and $0 < q \in \mathscr{P}(\mathcal{S})$. The value of the optimization problem:*

$$\min_{p \in \mathscr{P}(\mathcal{S})} \langle p, v \rangle \text{ such that } \mathrm{KL}[p \,\|\, q] \le C_{\mathrm{KL}} \tag{21}$$

*is equal to*

$$-\min_{\theta \ge 0} \theta \cdot C_{\mathrm{KL}} + \theta \ln \left\langle q, \exp\left(-\frac{v}{\theta}\right) \right\rangle . \tag{22}$$

*Let $\theta^\star$ be the solution of Equation (22). Then, the solution of Equation (21) is*

$$p \propto q \exp\left(-\frac{v}{\theta^\star}\right) .$$

Using this lemma, Equation (19) can be implemented by

$$Q_{c_n}^{(t+1)}(s,a) = c_n(s,a) + \gamma \sum_{s' \in \mathcal{S}} P_{s,a}^\star(s') V_{c_n}^{(t)}(s')$$

$$\text{where } P_{s,a}^\star \propto P(\cdot \mid s, a) \exp\left(\frac{V_{c_n}^{(t)}(\cdot)}{\theta_{s,a}^\star}\right) \tag{23}$$

$$\text{and } \theta_{s,a}^\star := \arg\min_{\theta \ge 0} \theta \cdot C_{\mathrm{KL}} + \theta \ln \left\langle P(\cdot \mid s, a), \exp\left(\frac{V_{c_n}^{(t)}(\cdot)}{\theta}\right) \right\rangle .$$

**Regularized alternative.** While Equation (22) is a convex optimization problem, solving it for all $P(\cdot \mid s, a) \; \forall(s,a) \in \mathcal{S} \times \mathcal{A}$ in Equation (23) is computationally extensive in practice.

Rather than the exact constrained problem of Equation (21), Yang et al. (2023) proposed the following regularized robust DP update:

$$Q_{c_n}^{(t+1)}(s,a) = c_n(s,a) + \gamma \max_{p \in \mathscr{P}(\mathcal{S})} \left( \sum_{s' \in \mathcal{S}} p(s') V_{c_n}^{(t)}(s') - C'_{\mathrm{KL}} \mathrm{KL}[p \,\|\, P(\cdot \mid s,a)] \right), \qquad (24)$$

where $C'_{\mathrm{KL}} > 0$ is a constant. This regularized form has the following efficient analytical solution:

$$Q_{c_n}^{(t+1)}(s,a) = c_n(s,a) + \gamma \sum_{s' \in \mathcal{S}} P_{s,a}^{\star}(s') V_{c_n}^{(t)}(s')$$

$$\text{where } P_{s,a}^{\star} \propto P(\cdot \mid s,a) \exp \left( \frac{V_{c_n}^{(t)}(\cdot)}{C'_{\mathrm{KL}}} \right). \qquad (25)$$

While this is a regularized approximation, the following lemma shows that Equation (25) solves some exact robust DP with a KL uncertainty set:

**Lemma 5** (Adaptation of **Proposition 3.1** and **Theorem 3.1** in Yang et al. (2023)). *For any $C'_{\mathrm{KL}} > 0$, there exists $C_{\mathrm{KL}} > 0$ such that Equation (24) converges linearly to the fixed point of Equation (23).*

Algorithm 4 summarizes the regularized DP update to implement the algorithms. Our experiment **(b)** in Section 6 uses Algorithm 4.

## D   EXPERIMENT DETAILS

The source code for the experiment is available at `https://github.com/matsuolab/RCMDP-Epigraph`.

**Environment construction.**   For the settings with finite uncertainty sets **(a)**, the parameters are set as $S = 7$, $A = 4$, $\gamma = 0.995$. We set $\gamma = 0.99$ for the CMDP setting **(c)**. For the setting with KL uncertainty set **(b)**, we set $S = 5$, $A = 3$, and $\gamma = 0.99$.

**`EpiRC-PGS` implementation.**   For the policy gradient subroutine (Algorithm 2), we set the iteration length $T$ and the learning rate $\alpha$ to ensure that Assumption 4 is satisfied with a sufficiently small $\varepsilon_{\mathrm{opt}}$. Specifically, for **(a)** and **(c)**, we set the iteration length to $T = 10^4$ and the learning rate to $\alpha = 5 \times 10^{-5}$. For **(b)**, we set $T = 10^3$ and $\alpha = 5 \times 10^{-4}$.

Notably, parameter tuning for `EpiRC-PGS` is straightforward, as any sufficiently large $T$ and small $\alpha$ should meet the conditions of Assumption 4. Since the initial policy in Algorithm 2 can be chosen arbitrarily, the $(k-1)$-th policy from the outer loop is used as the initial policy for the $k$-th policy computation.

For the finite uncertainty set settings **(a)** and **(c)**, we implement the evaluators $\widehat{J}_n$ and $\widehat{J}_n^{\partial}$ using Algorithm 3. For the KL uncertainty set setting **(b)**, we implement $\widehat{J}_n$ and $\widehat{J}_n^{\partial}$ using Algorithm 4 with $C'_{\mathrm{KL}} = 2.0$.

**`LF` implementation.**   The pseudocode for `LF` is shown in Algorithm 5. We set the iteration length and learning rate for the inner policy optimization to $T = 10^4$ and $\alpha_{\pi} = 5 \times 10^{-5}$ in **(a, c)**, and $T = 10^3$ and $\alpha_{\pi} = 5 \times 10^{-4}$ in **(b)**. Similar to `EpiRC-PGS`, these values are chosen to expect sufficient optimization in the inner loop. We choose $\alpha_{\lambda} = 0.01$ from $\{0.1, 0.01, 0.001\}$ for the outer updates, balancing between the convergence speed and performance.

## E   ADDITIONAL DEFINITIONS

Throughout this section, let $\mathcal{X}$ denote a set such that $\mathcal{X} \subset \mathbb{R}^d$ with $d \in \mathbb{N}$.

---

**Algorithm 5** Lagrangian Formulation Policy Gradient Search (`LF`)

---

1: **Input:** Outer iteration length $K \in \mathbb{N}$, inner iteration length $T \in \mathbb{N}$, learning rate for Lagrangian multipliers $\alpha_\lambda > 0$, learning rate for policy $\alpha_\pi > 0$
2: Initialize the Lagrangian multipliers $\lambda^{(0)} = \mathbf{0} \in \mathbb{R}^N$
3: Set an arbitrary initial policy $\pi^{(0)} \in \Pi$
4: **for** $k = 0, \cdots, K-1$ **do**
5:     Set the initial policy $\pi^{(k,0)} := \pi^{(k)}$ for the inner loop
6:     **for** $t = 0, \cdots, T-1$ **do**
7:         $g^{(k,t)} \in \partial L_{\lambda^{(k)}}(\pi^{(k,t)})$               // Compute policy gradient
8:         $\pi^{(k,t+1)} := \text{Proj}_\Pi(\pi^{(k,t)} - \alpha_\pi g^{(k,t)})$        // Policy update
9:     **end for**
10:    Set the new policy: $\pi^{(k+1)} := \pi^{(k,t^\star)}$ where $t^\star \in \arg\min_{t \in [\![0, T-1]\!]} L_{\lambda^{(k)}}(\pi^{(k,t)})$
11:    Update Lagrangian multipliers: $\lambda_n^{(k+1)} := \max\left\{\lambda_n^{(k)} + \alpha_\lambda\big(J_{n,\mathcal{U}}(\pi^{(k+1)}) - b_n\big), 0\right\}$ for all $n \in [\![1, N]\!]$
12: **end for**

---

**Definition 2** (Subgradient (Kruger, 2003)). Let $\mathcal{X} \subset \mathbb{R}^d$ be an open set where $d \in \mathbb{N}$. The set of (Fréchet) subgradients of a function $f : \mathcal{X} \to \mathbb{R}$ at a point $x \in \mathcal{X}$ is defined as the set

$$\partial f(x) := \left\{ u \in \mathcal{X} \;\middle|\; \liminf_{x' \to x, x' \neq x} \frac{f(x') - f(x) - \langle u, x' - x \rangle}{\|x' - x\|_2} \geq 0 \right\} .$$

Furthermore, if $\partial f(\mathbf{x})$ is a singleton, its element is denoted as $\nabla f(\mathbf{x})$ and called the (Fréchet) gradient of $f$ at $x$.

**Definition 3** (Lipschitz continuity). Let $\ell \geq 0$. A function $f : \mathcal{X} \to \mathbb{R}$ is $\ell$-Lipschitz if for any $x_1, x_2 \in \mathcal{X}$, we have that
$$\|f(x_1) - f(x_2)\|_2 \leq \ell \|x_1 - x_2\|_2 .$$

**Definition 4** (Smoothness). Let $\ell \geq 0$. A function $f : \mathcal{X} \to \mathbb{R}$ is $\ell$-smooth if for any $x_1, x_2 \in \mathcal{X}$, we have that
$$\|\nabla f(x_1) - \nabla f(x_2)\|_2 \leq \ell \|x_1 - x_2\|_2 .$$

**Definition 5** (Weak convexity). Let $\ell > 0$. A function $f : \mathcal{X} \to \mathbb{R}$ is $\ell$-weakly convex if for any $g \in \partial f(x)$ and $x, x' \in \mathcal{X}$,

$$f(x') - f(x) \geq \langle g, x' - x \rangle - \frac{\ell}{2} \|x' - x\|_2^2 .$$

Note that $f(x) + \frac{\ell}{2}\|x\|_2^2$ is convex in $\mathcal{X}$ if and only if $f$ is $\ell$-weakly convex.

**Definition 6** (Moreau envelope of a weakly convex function). Given a $\ell$-weakly convex function $f : \mathcal{X} \to \mathbb{R}$ and a parameter $0 < \tau < \ell^{-1}$, the Moreau envelope function of $f$ is given by $\text{M}_\tau \circ f : \mathbb{R}^d \to \mathbb{R}$ such that

$$(\text{M}_\tau \circ f)(x) = \min_{x' \in \mathcal{X}} \left\{ f(x') + \frac{1}{2\tau} \|x - x'\|_2^2 \right\} .$$

## F   Useful Lemmas

Throughout this section, $\mathcal{X}$ denotes a compact set such that $\mathcal{X} \subset \mathbb{R}^d$, where $d \in \mathbb{N}$.

**Lemma 6** (**Lemma D.2.** in Wang et al. (2023)). *Let $\ell \geq 0$ and $h : \mathcal{X} \to \mathbb{R}$ be an $\ell$-smooth function. Then, $h$ is a $\ell$-weakly convex function.*

**Lemma 7** (e.g., **Proposition 13.37** in Rockafellar & Wets (2009)). *Let $f : \mathcal{X} \to \mathbb{R}$ be an $\ell$-weakly convex function, and let $0 < \tau < \ell$ be a parameter. The Moreau envelope function $\text{M}_\tau \circ f : \mathbb{R}^d \to \mathbb{R}$ is differentiable, and its gradient is given by*

$$\nabla(\text{M}_\tau \circ f)(x) = \frac{1}{\tau}\left(x - \arg\min_{x' \in \mathcal{X}}\left(f(x') + \frac{1}{2\tau}\|x - x'\|_2^2\right)\right) .$$

**Lemma 8** (Sion's minimax theorem (Sion, 1958)). *Let $n, m \in \mathbb{N}$. Let $\mathcal{X} \subset \mathbb{R}^n$ be a compact convex set and $\mathcal{Y} \subset \mathbb{R}^m$ a convex set. Suppose that $f : \mathcal{X} \times \mathcal{Y} \to \mathbb{R}$ satisfies the following two properties:*

- *$f(x, \cdot)$ is upper semicontinuous and quasi-concave on $\mathcal{Y}$ for any $x \in \mathcal{X}$.*

- *$f(\cdot, y)$ is lower semicontinuous and quasi-convex on $\mathcal{X}$ for any $y \in \mathcal{Y}$.*

*Then, $\min_{x \in \mathcal{X}} \sup_{y \in \mathcal{Y}} f(x, y) = \sup_{y \in \mathcal{Y}} \min_{x \in \mathcal{X}} f(x, y)$.*

**Lemma 9** (e.g., **Problem 9.13, Page 99** in Clarke et al. (2008)). *Let $\mathcal{Y} \subset \mathbb{R}^m$ be a compact set and $f : \mathbb{R}^d \times \mathcal{Y} \to \mathbb{R}$ be a continuous function of two arguments. Consider a point $\bar{x} \in \mathbb{R}^d$ and let $\Omega(\bar{x}) \subset \mathbb{R}^d$ be its neighborhood. For any $(x, y) \in \Omega(\bar{x}) \times \mathcal{Y}$, suppose that the gradient $\nabla_x f(x, y)$ exists and is jointly continuous.*

*Let $h(\bar{x}) := \max_{y \in \mathcal{Y}} f(\bar{x}, y)$. Then, the subgradient of $h$ at $\bar{x}$ is given by*

$$\partial h(\bar{x}) = \operatorname{conv}\left\{ \nabla_x f(\bar{x}, y) \,\middle|\, y \in \arg\max_{y \in \mathcal{Y}} f(\bar{x}, y) \right\} .$$

**Lemma 10.** *Let $N \in \mathbb{N}$. Let $f_i : \mathcal{X} \to \mathbb{R}$ for $i \in [\![1, N]\!]$ be $\ell$-weakly convex functions for some $\ell \geq 0$. Define the pointwise maximum function $f : \mathcal{X} \to \mathbb{R}$ as*

$$f(x) = \max\{f_1(x), \cdots, f_N(x)\} \quad \forall x \in \mathcal{X} .$$

*Then, for any $x \in \mathcal{X}$,*

$$\partial f(x) = \operatorname{conv}\left\{ g \in \mathbb{R}^d \,\middle|\, g \in \partial f_i(x), f_i(x) = f(x) \right\} .$$

*Proof.* The claim directly follows from **Theorem 1.3** and **Theorem 1.5** in Mikhalevich et al. (2024). $\square$

**Lemma 11** (Maximum difference inequality). *Let $N \in \mathbb{N}$. For two sets of real numbers $\{x_i\}_{i \in [\![1,N]\!]}$ and $\{y_i\}_{i \in [\![1,N]\!]}$, where $x_i, y_i \in \mathbb{R}$,*

$$\left| \max_{i \in [\![1,N]\!]} x_i - \max_{i' \in [\![1,N]\!]} y_{i'} \right| \leq \max_{i \in [\![1,N]\!]} |x_i - y_i| .$$

*Proof.* For any $i \in [\![1, N]\!]$,

$$\max_{i \in [\![1,N]\!]} x_i = \max_{i \in [\![1,N]\!]} x_i - y_i + y_i \leq \max_{i \in [\![1,N]\!]} (x_i - y_i) + \max_{i' \in [\![1,N]\!]} y_{i'}$$
$$\implies \max_{i \in [\![1,N]\!]} x_i - \max_{i' \in [\![1,N]\!]} y_{i'} \leq \max_{i \in [\![1,N]\!]} x_i - y_i .$$

By the symmetry of $x_i$ and $y_i$, we have $\max_{i \in [\![1,N]\!]} y_i - \max_{i' \in [\![1,N]\!]} x_{i'} \leq \max_{i \in [\![1,N]\!]} y_i - x_i$. Therefore,

$$\left| \max_{i \in [\![1,N]\!]} x_i - \max_{i' \in [\![1,N]\!]} y_{i'} \right| \leq \max_{i \in [\![1,N]\!]} |x_i - y_i| .$$

$\square$

**Lemma 12** (Point-wise maximum preserves weak convexity). *Let $h : \mathcal{X} \to \mathbb{R}$ and $f : \mathcal{X} \to \mathbb{R}$ be $\ell_h$- and $\ell_f$-weakly convex functions, respectively. Then, $g : \mathcal{X} \to \mathbb{R}$ defined by $g(x) = \max\{h(x), f(x)\}$ for any $x \in \mathcal{X}$ is $\ell$-weakly convex, where $\ell := \max\{\ell_h, \ell_f\}$.*

*Proof.* By the definition of weak convexity, for any $\theta \in [0, 1]$ and $x, y \in \mathcal{X}$,

$$h(\theta x + (1-\theta)y) + \frac{\ell_h}{2}\|\theta x + (1-\theta)y\|_2^2 \leq \theta\left(h(x) + \frac{\ell_h}{2}\|x\|_2^2\right) + (1-\theta)\left(h(y) + \frac{\ell_h}{2}\|y\|_2^2\right) .$$

A similar inequality holds for $f$. Then,

$$g(\theta x + (1-\theta)y) + \frac{\ell}{2}\|\theta x + (1-\theta)y\|_2^2$$

$$= \max\{h(\theta x + (1-\theta)y), f(\theta x + (1-\theta)y)\} + \frac{\ell}{2}\|\theta x + (1-\theta)y\|_2^2$$

$$= \max\left\{h(\theta x + (1-\theta)y) + \frac{\ell}{2}\|\theta x + (1-\theta)y\|_2^2, f(\theta x + (1-\theta)y) + \frac{\ell}{2}\|\theta x + (1-\theta)y\|_2^2\right\}$$

$$\leq \max\left\{\theta h(x) + (1-\theta)h(y) + \frac{\ell}{2}\left(\theta\|x\|_2^2 + (1-\theta)\|y\|_2^2\right), \theta f(x) + (1-\theta)f(y) + \frac{\ell}{2}\left(\theta\|x\|_2^2 + (1-\theta)\|y\|_2^2\right)\right\}$$

$$= \max\{\theta h(x) + (1-\theta)h(y), \theta f(x) + (1-\theta)f(y)\} + \frac{\ell}{2}\left(\theta\|x\|_2^2 + (1-\theta)\|y\|_2^2\right)$$

$$\leq \theta\left(\max\{h(x), f(x)\} + \frac{\ell}{2}\|x\|_2^2\right) + (1-\theta)\left(\max\{h(y), f(y)\} + \frac{\ell}{2}\|y\|_2^2\right)$$

$$= \theta\left(g(x) + \frac{\ell}{2}\|x\|_2^2\right) + (1-\theta)\left(g(y) + \frac{\ell}{2}\|y\|_2^2\right).$$

Therefore, $g$ is $\ell$-weakly convex. □

**Lemma 13.** *Let $N_\mathcal{X}(x)$ be the normal cone of $\mathcal{X}$ at $x \in \mathcal{X}$, defined as*

$$N_\mathcal{X}(x) := \left\{g \in \mathbb{R}^d \mid \langle g, y\rangle \leq \langle g, x\rangle \quad \forall y \in \mathcal{X}\right\}.$$

*Define the indicator function $\mathbb{I}_\mathcal{X} : \mathbb{R}^d \to \mathbb{R}$ such that*

$$\mathbb{I}_\mathcal{X}(x) = \begin{cases} 0 & \text{if } x \in \mathcal{X} \\ \infty & \text{otherwise} \end{cases}.$$

*Then, $\partial\mathbb{I}_\mathcal{X}(x) = N_\mathcal{X}(x)$ for any $x \in \mathcal{X}$.*

*Proof.* Note that any $g \in \partial\mathbb{I}_\mathcal{X}(x)$ satisfies

$$\mathbb{I}_\mathcal{X}(y) \geq \mathbb{I}_\mathcal{X}(x) + \langle g, y - x\rangle \quad \forall y \in \mathbb{R}^d. \tag{26}$$

Suppose that $g \notin N_\mathcal{X}(x)$. Then, there exists $y' \in \mathcal{X}$ such that $\langle g, x\rangle < \langle g, y'\rangle$, which contradicts Equation (26). Therefore, $g \in N_\mathcal{X}(x)$ for any $g \in \partial\mathbb{I}_\mathcal{X}(x)$ and thus $\partial\mathbb{I}_\mathcal{X}(x) \subseteq N_\mathcal{X}(x)$.

Consider $g \in N_\mathcal{X}(x)$. It satisfies $0 \geq \langle g, y - x\rangle$ for any $y \in \mathcal{X}$. Since $x \in \mathcal{X}$ and by the definition of $\mathbb{I}_\mathcal{X}$, Equation (26) holds for any $y \in \mathbb{R}^d$. Therefore, $N_\mathcal{X}(x) \subseteq \partial\mathbb{I}_\mathcal{X}(x)$. This concludes the proof. □

**Lemma 14.** *Let $h : \mathcal{X} \to \mathbb{R}$ be an $\ell$-weakly convex function. For $0 < \tau < 1/\ell$, define*

$$\bar{x}_\tau \in \arg\min_{x' \in \mathcal{X}} h(x') + \frac{1}{2\tau}\|x - x'\|_2^2.$$

*Then, there exists a subgradient $g \in \partial h(\bar{x}_\tau)$ such that, for any $y \in \mathcal{X}$,*

$$\langle g, \bar{x}_\tau - y\rangle \leq \langle\nabla(\mathrm{M}_\tau \circ h)(x), \bar{x}_\tau - y\rangle$$

*Proof.* Let $f : \mathbb{R}^d \to \mathbb{R}$ be a function such that $f(x) = h(x) + \mathbb{I}_\mathcal{X}(x)$.

The Moreau envelope function of $f$ satisfies that, for any $x \in \mathbb{R}^d$,

$$(\mathrm{M}_\tau \circ f)(x) = \min_{x' \in \mathbb{R}^d}\left\{h(x') + \mathbb{I}_\mathcal{X}(x') + \frac{1}{2\tau}\|x - x'\|_2^2\right\} = \min_{x' \in \mathcal{X}}\left\{h(x') + \frac{1}{2\tau}\|x - x'\|_2^2\right\}.$$

It holds that $\nabla(\mathrm{M}_\tau \circ f)(x) = \frac{1}{\tau}(x - \bar{x}_\tau)$ due to Lemma 7.

Note that

$$\bar{x}_\tau \in \arg\min_{x' \in \mathcal{X}} h(x') + \frac{1}{2\tau}\|x - x'\|_2^2 = \arg\min_{x' \in \mathbb{R}^d} h(x') + \mathbb{I}_\mathcal{X}(x') + \frac{1}{2\tau}\|x - x'\|_2^2.$$

It is clear that $\bar{x}_\tau$ is a minimizer of the function $\phi_x(x') := h(x') + \mathbb{I}_\mathcal{X}(x') + \frac{1}{2\tau}\|x - x'\|_2^2$. Therefore, it holds that $\mathbf{0} \in \partial\phi_x(\bar{x}_\tau)$. Accordingly,

$$\mathbf{0} \in \partial\left(h(y) + \mathbb{I}_\mathcal{X}(y) + \frac{1}{2\tau}\|x - y\|_2^2\right)\bigg|_{y=\bar{x}_\tau} \implies -\frac{1}{\tau}(\bar{x}_\tau - x) \in \partial(h(y) + \mathbb{I}_\mathcal{X}(y))|_{y=\bar{x}_\tau} .$$

Due to Lemma 13, $\partial\mathbb{I}_\mathcal{X}(x) = N_\mathcal{X}(x)$. Therefore, there exists a subgradient $g \in \partial h(\bar{x}_\tau)$ such that

$$-g - \frac{1}{\tau}(\bar{x}_\tau - x) \in N_\mathcal{X}(\bar{x}_\tau) .$$

Since any $z \in N_\mathcal{X}(\bar{x}_\tau)$ satisfies $\langle z, y - \bar{x}_\tau \rangle \leq 0$ for any $y \in \mathcal{X}$, it holds that

$$\langle -g, y - \bar{x}_\tau \rangle \leq \left\langle \frac{1}{\tau}(\bar{x}_\tau - x), y - \bar{x}_\tau \right\rangle, \quad \forall y \in \mathcal{X} .$$

Then the claim follows from the fact that $\frac{1}{\tau}(x - \bar{x}_\tau) = \nabla(\mathrm{M}_\tau \circ h)(x)$ due to Lemma 7. $\qquad\square$

**Lemma 15** (Linear optimization on convex hull). *Given $c \in \mathbb{R}^d$ and a compact set $\mathcal{X} \subset \mathbb{R}^d$, it holds that*

$$\min_{x\in\mathcal{X}}\langle c, x\rangle = \min_{x\in\mathrm{conv}\{\mathcal{X}\}}\langle c, x\rangle .$$

*Proof.* Let $x^\star \in \arg\min_{x\in\mathrm{conv}\{\mathcal{X}\}}\langle c, x\rangle$. The claim holds for $x^\star \in \mathcal{X}$. Suppose that $x^\star \notin \mathcal{X}$. Then, by the definition of the convex hull, there exist $y, z \in \mathcal{X}$ and $\theta \in (0, 1)$ such that $y \neq z$ and

$$x^\star = \theta y + (1 - \theta)z .$$

Since $x^\star$ is a minimizer, we have

$$\langle c, x^\star \rangle \leq \langle c, y \rangle \quad \text{and} \quad \langle c, x^\star \rangle \leq \langle c, z \rangle .$$

Accordingly,

$$\langle c, x^\star \rangle = \theta\langle c, x^\star \rangle + (1 - \theta)\langle c, x^\star \rangle \leq \theta\langle c, y \rangle + (1 - \theta)\langle c, z \rangle = \langle c, x^\star \rangle .$$

The inequality must be an equality, and thus

$$\theta\underbrace{(\langle c, y \rangle - \langle c, x^\star \rangle)}_{\geq 0} + (1 - \theta)\underbrace{(\langle c, z \rangle - \langle c, x^\star \rangle)}_{\geq 0} = 0 .$$

Since $\theta \in (0, 1)$, it holds that

$$\langle c, y \rangle = \langle c, z \rangle = \langle c, x^\star \rangle .$$

The above equality means that both $y$ and $z \in \mathcal{X}$ satisfy $\langle c, y \rangle = \langle c, z \rangle = \min_{x\in\mathrm{conv}\{\mathcal{X}\}}\langle c, x\rangle$. Therefore, $\min_{x\in\mathcal{X}}\langle c, x\rangle = \min_{x\in\mathrm{conv}\{\mathcal{X}\}}\langle c, x\rangle$. $\qquad\square$

**Lemma 16** (**Lemma 3.1** in Wang et al. (2023)). *Let*

$$\ell_{\mathrm{Lp}} := H^2\sqrt{A} \quad \text{and} \quad \ell_{\mathrm{sm}} := 2\gamma A H^3 .$$

*For any $\pi, \pi' \in \Pi$, $P : \mathcal{S} \times \mathcal{A} \to \mathscr{P}(\mathcal{S})$, $\mu \in \mathscr{P}(\mathcal{S})$, and $c \in [0, 1]^{SA}$,*

$$|J_{c,P}(\pi) - J_{c,P}(\pi')| \leq \ell_{\mathrm{Lp}}\|\pi - \pi'\|_2 , \quad \|\nabla J_{c,P}(\pi) - \nabla J_{c,P}(\pi)\|_2 \leq \ell_{\mathrm{sm}}\|\pi - \pi'\|_2 ,$$
$$\text{and } |J_{c,\mathcal{U}}(\pi) - J_{c,\mathcal{U}}(\pi)| \leq \ell_{\mathrm{Lp}}\|\pi - \pi'\|_2 .$$

*Furthermore, $J_{c,P}(\pi)$ is $\ell_{\mathrm{sm}}$-weakly convex in $\Pi$, as follows directly from Lemma 6.*

**Lemma 17** (e.g., **Lemma 4.1** in Agarwal et al. (2021) and **Lemma E.2** in Wang et al. (2023)). *Let $\mu \in \mathscr{P}(\mathcal{S})$ such that $\min_{s\in\mathcal{S}}\mu(s) > 0$. For any $\pi \in \Pi$, $P : \mathcal{S} \times \mathcal{A} \to \mathscr{P}(\mathcal{S})$, and $c \in [0, 1]^{SA}$,*

$$J_{c,P}(\pi) - J_{c,P}(\pi_{c,P}^\star) \leq H\left\|\frac{d_P^{\pi_{c,P}^\star}}{\mu}\right\|_\infty \max_{\pi'\in\Pi}\langle \pi - \pi', \nabla J_{c,P}(\pi) \rangle ,$$

*where $\pi_{c,P}^\star \in \arg\min_{\pi'\in\Pi} J_{c,P}(\pi')$.*

## G  PROOF OF THEOREM 1

*Proof of Theorem 1.* Consider the deterministic RCMDP shown in Figure 1a with $N = 1$, $\mathcal{U} = \{P_1, P_2\}$, $\mathcal{S} = \{s_1, s_2, s_3, s_4\}$, and $\mathcal{A} = \{a_1, a_2\}$. Set the initial distribution such that $\mu(s_1) = \mu(s_2) = \mu(s_3) = \mu(s_4) = 1/4$.

**First part of Theorem 1.**  We set $\lambda = 1$. The threshold $b_1$ can be arbitrary.

Let $\pi_1$ and $\pi_2$ be two policies such that $\pi_1$ always chooses $a_1$ and $\pi_2$ always chooses $a_2$ in any state. For any $\delta > 0$, we will show two results:

- Equation (28): $L_\lambda(\pi_2) - \min_{\pi \in \Pi} L_\lambda(\pi) \geq \frac{H\gamma}{4} - \frac{3H\delta}{4}$ .

- Equation (30): $(\nabla L_\lambda(\pi_2))(\cdot, a_1) > (\nabla L_\lambda(\pi_2))(\cdot, a_2)$.

The former demonstrates the suboptimality of $\pi_2$, while the latter shows that choosing $a_2$ always decreases $L_\lambda$, indicating that $\pi_2$ is a local minimum.

According to the RCMDP construction, for any $\pi \in \Pi$, we have

$$\mu(s_3)V^\pi_{c_0,P_1}(s_3) + \mu(s_4)V^\pi_{c_0,P_1}(s_4) = \frac{H}{4}(1+\gamma) \,,$$

$$\mu(s_3)V^\pi_{c_0,P_2}(s_3) + \mu(s_4)V^\pi_{c_0,P_2}(s_4) = \frac{H}{4}(1-\gamma) \,,$$

$$\mu(s_3)V^\pi_{c_1,P_1}(s_3) + \mu(s_4)V^\pi_{c_1,P_1}(s_4) = \frac{H}{4}(1-\gamma) \,,$$

$$\mu(s_3)V^\pi_{c_1,P_2}(s_3) + \mu(s_4)V^\pi_{c_1,P_2}(s_4) = \frac{H}{4}(1+\gamma) \,.$$

For $\pi_1$ and $\pi_2$, it is easy to verify that

$$\mu(s_1)V^{\pi_1}_{c_0,P_1}(s_1) + \mu(s_2)V^{\pi_1}_{c_0,P_1}(s_2) = \frac{1}{4}\big(\delta + \gamma + \gamma^2\delta + \cdots\big) + \frac{1}{4}\big(1 + \gamma\delta + \gamma^2 + \cdots\big) = \frac{H}{4}(1+\delta) \,,$$

$$\mu(s_1)V^{\pi_1}_{c_0,P_2}(s_1) + \mu(s_2)V^{\pi_1}_{c_0,P_2}(s_2) = \frac{1}{4}\big(\delta + \gamma + \gamma^2 + \cdots\big) + \frac{1}{4}\big(1 + \gamma + \gamma^2 + \cdots\big) = \frac{H}{4}(1+\gamma) + \frac{\delta}{4} \,,$$

$$\mu(s_1)V^{\pi_1}_{c_1,P_1}(s_1) + \mu(s_2)V^{\pi_1}_{c_1,P_1}(s_2) = \frac{H}{2} \,, \quad \mu(s_1)V^{\pi_1}_{c_1,P_2}(s_1) + \mu(s_2)V^{\pi_1}_{c_1,P_2}(s_2) = \frac{H}{2} \,,$$

$$\mu(s_1)V^{\pi_2}_{c_0,P_1}(s_1) + \mu(s_2)V^{\pi_2}_{c_0,P_1}(s_2) = \frac{H}{2} \,, \quad \mu(s_1)V^{\pi_2}_{c_0,P_2}(s_1) + \mu(s_2)V^{\pi_2}_{c_0,P_2}(s_2) = \frac{H}{2} \,,$$

$$\mu(s_1)V^{\pi_2}_{c_1,P_1}(s_1) + \mu(s_2)V^{\pi_2}_{c_1,P_1}(s_2) = \frac{H}{4}(1 + \gamma - 2\delta) \,,$$

$$\mu(s_1)V^{\pi_2}_{c_1,P_2}(s_1) + \mu(s_2)V^{\pi_2}_{c_1,P_2}(s_2) = \frac{H}{4}(1 + \gamma - 2\delta) \,.$$

Therefore,

$$J_{c_0,P_1}(\pi_1) = \frac{H}{2} + \frac{H}{4}(\gamma + \delta) \,, \quad J_{c_0,P_2}(\pi_1) = \frac{H}{2} + \frac{\delta}{4} \,,$$

$$J_{c_1,P_1}(\pi_1) = \frac{H}{4}(3 - \gamma) \,, \quad J_{c_1,P_2}(\pi_1) = \frac{H}{4}(3 + \gamma) \,,$$

$$J_{c_0,P_1}(\pi_2) = \frac{H}{4}(3 + \gamma) \,, \quad J_{c_0,P_2}(\pi_2) = \frac{H}{4}(3 - \gamma) \,, \tag{27}$$

$$J_{c_1,P_1}(\pi_2) = \frac{H}{2} - \frac{H\delta}{2} \,, \quad J_{c_1,P_2}(\pi_2) = \frac{H}{2} + \frac{H\gamma}{2} - \frac{H\delta}{2} \,,$$

Hence,

$$J_{c_0,\mathcal{U}}(\pi_1) = J_{c_0,P_1}(\pi_1) = \frac{H}{2} + \frac{H\gamma}{4} + \frac{H\delta}{4} \,, \quad J_{c_1,\mathcal{U}}(\pi_1) = J_{c_1,P_2}(\pi_1) = \frac{H}{4}(3 + \gamma) \,,$$

$$J_{c_0,\mathcal{U}}(\pi_2) = J_{c_0,P_1}(\pi_2) = \frac{H}{4}(3 + \gamma) \,, \quad J_{c_1,\mathcal{U}}(\pi_2) = J_{c_1,P_2}(\pi_2) = \frac{H}{2} + \frac{H\gamma}{2} - \frac{H\delta}{2} \,.$$

Accordingly, since $\lambda = 1$, we have

$$L_\lambda(\pi_2) - \min_{\pi \in \Pi} L_\lambda(\pi) \geq L_\lambda(\pi_2) - L_\lambda(\pi_1) \geq \frac{H\gamma}{4} - \frac{3H\delta}{4} \ . \tag{28}$$

The next task is to show that $(\nabla L_\lambda(\pi_2))(\cdot, a_1) > (\nabla L_\lambda(\pi_2))(\cdot, a_2)$.

By using Lemma 10, it is easy to show that

$$\nabla L_\lambda(\pi_2) = \nabla J_{c_0, P_1}(\pi_2) + \nabla J_{c_1, P_2}(\pi_2) \ .$$

Since $d_{P_1}^{\pi_2} = d_{P_2}^{\pi_2} = 0.25\mathbf{1}$ and due to Lemma 1, we have

$$\frac{4}{H} \nabla L_\lambda(\pi_2) = Q_{c_0, P_1}^{\pi_2} + Q_{c_1, P_2}^{\pi_2} \ .$$

Note that

$$
\begin{aligned}
\frac{4}{H}(\nabla L_\lambda(\pi_2))(s_1, a_1) &= Q_{c_0, P_1}^{\pi_2}(s_1, a_1) + Q_{c_1, P_2}^{\pi_2}(s_1, a_1) = (\delta + H\gamma) + (H - H\gamma\delta) \ , \\
\frac{4}{H}(\nabla L_\lambda(\pi_2))(s_1, a_2) &= Q_{c_0, P_1}^{\pi_2}(s_1, a_2) + Q_{c_1, P_2}^{\pi_2}(s_1, a_2) = H + H(\gamma - \delta) \ , \\
\frac{4}{H}(\nabla L_\lambda(\pi_2))(s_2, a_1) &= Q_{c_0, P_1}^{\pi_2}(s_2, a_1) + Q_{c_1, P_2}^{\pi_2}(s_2, a_1) = 2H \ , \\
\frac{4}{H}(\nabla L_\lambda(\pi_2))(s_2, a_2) &= Q_{c_0, P_1}^{\pi_2}(s_2, a_2) + Q_{c_1, P_2}^{\pi_2}(s_2, a_2) = 2H(1 - \delta) \ .
\end{aligned}
\tag{29}
$$

Therefore, since $\delta > 0$,

$$
\begin{aligned}
\frac{4}{H}((\nabla L_\lambda(\pi_2))(s_1, a_1) - (\nabla L_\lambda(\pi_2))(s_1, a_2)) &= \delta - H\gamma\delta + H\delta = 2\delta > 0 \ , \\
\frac{4}{H}((\nabla L_\lambda(\pi_2))(s_2, a_1) - (\nabla L_\lambda(\pi_2))(s_2, a_2)) &= 2H\delta > 0 \ .
\end{aligned}
\tag{30}
$$

Now, with a sufficiently small $R > 0$, let $\widetilde{\Pi}_2 := \{\pi \in \Pi \mid \|\pi - \pi_2\|_2 \leq R, \ \pi \neq \pi_2\}$ be policies near $\pi_2$. When $R$ is sufficiently small, due to the Lipshictz continuity of $J_{c_n, P}(\pi)$ by Lemma 16, Equation (27) indicates that

$$J_{c_0, \mathcal{U}}(\pi) = J_{c_0, P_1}(\pi) \ \text{ and } \ J_{c_1, \mathcal{U}}(\pi) = J_{c_1, P_2}(\pi) \quad \forall \pi \in \widetilde{\Pi}_2 \ . \tag{31}$$

Similarly, due to Equation (31) with the Lipshictz continuity of $\nabla J_{n, P}(\pi)$ by Lemma 16, Equation (29) and Equation (30) indicate that,

$$(\nabla L_\lambda(\pi))(\cdot, a_1) > (\nabla L_\lambda(\pi))(\cdot, a_2) \quad \forall \pi \in \widetilde{\Pi}_2 \ .$$

Therefore, since $\pi_2$ always chooses $a_2$, we have $L_\lambda(\pi_2) < L_\lambda(\pi) \ \forall \pi \in \widetilde{\Pi}_2$ for a sufficiently small $R > 0$. The first part of the claim holds by setting $\delta = \gamma/4$ with Equation (28).

**Second part of Theorem 1.** Consider again the deterministic RCMDP given in the previous part of the proof with $\delta = \gamma/4$. For a value $b_1 \in \mathbb{R}$, define a function $\Psi_{b_1} : \mathbb{R} \to \mathbb{R}$ such that

$$\Psi_{b_1}(\lambda) = \min_{\pi \in \Pi} J_{c_0, \mathcal{U}}(\pi) + \lambda J_{c_1, \mathcal{U}}(\pi) - \lambda b_1 = \min_{\pi \in \Pi} L_\lambda(\pi) \ .$$

We first show that, when $b_1$ ranges from 0 to $H$, the $\arg\sup_{\lambda \in \mathbb{R}_+} \Psi_{b_1}(\lambda)$ ranges from $\infty$ to 0.

Since $\mathcal{U} = \{P_1, P_2\}$, Lemma 10 and Lemma 1 indicates that, for any $\lambda \in \mathbb{R}$ and $b_1 \in \mathbb{R}$,

$$\partial_\lambda \Psi_{b_1}(\lambda) \subseteq \mathrm{conv}\{J_{c_1, P}(\pi) - b_1 \mid \pi \in \Pi, P \in \{P_1, P_2\}\} \ . \tag{32}$$

Since $\mu = \frac{1}{4} \cdot \mathbf{1}$ and due to the construction of the RCMDP in Figure 1a, it is easy to verify that,

$$
\begin{aligned}
\min_{\pi \in \Pi} \min_{P \in \mathcal{U}} J_{c_1, P}(\pi) &\geq \min_{\pi \in \Pi} \min_{P \in U} \min_{s \in \{s_3, s_4\}} \frac{1}{4} V_{c_1, P}^\pi(s) = \frac{1}{4} \ , \\
\max_{\pi \in \Pi} \max_{P \in \mathcal{U}} J_{c_1, P}(\pi) &\leq \frac{H}{2} + \max_{\pi \in \Pi} \max_{P \in U} \frac{1}{4}\left(V_{c_1, P}^\pi(s_3) + V_{c_1, P}^\pi(s_4)\right) = H - \frac{1}{4} \ .
\end{aligned}
$$

By inserting this to Equation (32), for any $\lambda \in \mathbb{R}$ and $b_1 \in \mathbb{R}$, we have

$$\frac{1}{4} - b_1 \leq g \leq H - \frac{1}{4} - b_1 \quad \forall g \in \partial_\lambda \Psi_{b_1}(\lambda) . \tag{33}$$

Therefore,

- When $b_1 \in [0, 1/4)$, $g \in \partial_\lambda \Psi_0(\lambda)$ must satisfy $g > 0$ for any $\lambda$. Thus, $\{\infty\} = \arg\sup_{\lambda \in \mathbb{R}_+} \Psi_{b_1}(\lambda)$ for any $b_1 \in [0, 1/4)$. Moreover, $\sup_{\lambda \in \mathbb{R}_+} \Psi_{b_1}(\lambda) = \infty$.

- When $b_1 \in (H - 1/4, H]$, $g \in \partial_\lambda \Psi_{b_1}(\lambda)$ must satisfy $g < 0$ for any $\lambda$. Thus, $\{0\} = \arg\max_{\lambda \in \mathbb{R}_+} \Psi_{b_1}(\lambda)$ for any $b_1 \in (H - 1/4, H]$. Moreover, $\max_{\lambda \in \mathbb{R}_+} \Psi_{b_1}(\lambda) \leq H$.

Next, we will show that there exists $b_1$ such that $1 \in \arg\max_{\lambda \in \mathbb{R}_+} \Psi_{b_1}(\lambda)$. From now, we only consider sufficiently large $b_1$ such that the value of $\arg\max_{\lambda \in \mathbb{R}_+} \Psi_{b_1}(\lambda)$ becomes finite.

Let $\Psi_{b_1}^\star := \max_{\lambda \in \mathbb{R}_+} \Psi_{b_1}(\lambda)$ and $f(b_1) := \frac{\Psi_{b_1}^\star - \Psi_H^\star}{b_1 - H}$. Since $f(H - 1/5) = 0$, $f(0) = -\infty$, and $f(b_1)$ is continuous in $b_1$, the intermediate value theorem ensures that there exists $b_1' \in [0, H]$ such that $f(b_1') = -1$. Moreover, the generalized mean value theorem (**Theorem 2.3.7** in Clarke (1990)) states that there exists $b_1^\star \in [b_1', H]$ such that $-1 = f(b_1') \in \partial_{b_1} \Psi_{b_1^\star}^\star$.

Let $\Lambda_{b_1}$ be the set that provides maximums of $\Psi_{b_1}$, i.e., $\Lambda_{b_1} := \arg\max_{\lambda \in \mathbb{R}_+} \Psi_{b_1}(\lambda)$. Using Lemma 9,

$$-1 \in \partial_{b_1} \Psi_{b_1^\star}^\star = \text{conv}\{\nabla_{b_1} \Psi_{b_1^\star}(\lambda) \mid \lambda \in \Lambda_{b_1^\star}\} = \text{conv}\{-\lambda \mid \lambda \in \Lambda_{b_1^\star}\} = [-\max \Lambda_{b_1^\star}, -\min \Lambda_{b_1^\star}] .$$

Since $\Psi_{b_1}(\lambda)$ is concave in $\lambda$, any $\lambda \in [\min \Lambda_{b_1}, \max \Lambda_{b_1}]$ provides $\max_{\lambda \in \mathbb{R}_+} \Psi_{b_1}(\lambda)$. Thus, $1 \in \arg\max_{\lambda \in \mathbb{R}_+} \Psi_{b_1^\star}(\lambda)$. This proves the existence of $b_1$ such that $1 \in \arg\max_{\lambda \in \mathbb{R}_+} \min_{\pi \in \Pi} L_\lambda(\pi)$.

$\square$

# H  MISSING PROOFS

## H.1  PROOF OF LEMMA 2

*Proof of Lemma 2.* We prove the first claim. Recall the definition of $\Delta_{b_0}^\star$:

$$\Delta_{b_0}^\star = \min_{\pi \in \Pi} \Delta_{b_0}(\pi) = \min_{\pi \in \Pi} \max_{n \in [\![0, N]\!]} J_{c_n, \mathcal{U}}(\pi) - b_n . \tag{34}$$

It is easy to see that $\Delta_{b_0}(\pi)$ is monotonically decreasing in $b_0$. Consider two real numbers $x \leq y$ and let $\pi^x \in \arg\min_{\pi \in \Pi} \Delta_x(\pi)$. Then,

$$\Delta_y^\star = \min_{\pi \in \Pi} \Delta_y(\pi) \leq \Delta_y(\pi^x) \leq \Delta_x(\pi^x) = \min_{\pi \in \Pi} \Delta_x(\pi) = \Delta_x^\star .$$

Therefore, $\Delta_{b_0}^\star$ is monotonically decreasing in $b_0$.

Next, we prove the second claim. Suppose that $\Delta_{J^\star}^\star < 0$. Then, there exists a feasible policy $\pi \in \Pi_{\text{F}}$ such that $J_{c_0, \mathcal{U}}(\pi) < J^\star = J_{c_0, \mathcal{U}}(\pi^\star)$. This contradicts the definition of the optimal policy. Therefore, $\Delta_{J^\star}^\star \geq 0$.

Suppose that $\Delta_{J^\star}^\star > 0$. Since $\min_{\pi \in \Pi} \Delta_{J^\star}(\pi) > 0$, no feasible policy achieves the objective return $J^\star$. This also contradicts the existence of the optimal policy. Therefore, $\Delta_{J^\star}^\star = 0$. $\square$

## H.2  PROOF OF THEOREM 2

*Proof of Theorem 2.* We first prove Equation (9) by contradiction. Let $x := \min\{b_0 \in [0, H] \mid \Delta_{b_0}^\star \leq 0\}$ and suppose that $x < J^\star$. Since $\Delta_{J^\star}^\star = 0$ by Lemma 2, there exists a feasible policy $\pi \in \Pi_{\text{F}}$ such that $J_{c_0, \mathcal{U}}(\pi) \leq x < J^\star = J_{c_0, \mathcal{U}}(\pi^\star)$. This contradicts the definition of the optimal policy.

We then show that Equation (9) provides $\pi^\star$. Since $\Delta_{J^\star}^\star = 0$ by Lemma 2, any policy $\pi \in \arg\min_{\pi \in \Pi} \Delta_{b_0}(\pi)$ is feasible and satisfies $J_{c_0, \mathcal{U}}(\pi) = J^\star$. The claim directly follows from the definition of an optimal policy. $\square$

## H.3 PROOF OF LEMMA 3

Instead of Lemma 3, we prove the following lemma that includes Lemma 3.

**Lemma 18** (Properties of $\Delta_{b_0}$). *The following properties hold for any $b_0 \in \mathbb{R}$.*

1. *(Lipschitz continuity): For any $\pi, \pi' \in \Pi$, $|\Delta_{b_0}(\pi) - \Delta_{b_0}(\pi')| \leq \ell_{\mathrm{Lp}}\|\pi - \pi'\|_2$ with $\ell_{\mathrm{Lp}} := H^2\sqrt{A}$.*

2. *(Weak convexity): $\Delta_{b_0}(\pi) + \frac{\ell_{\mathrm{sm}}}{2}\|\pi\|_2^2$ is convex in $\pi$ with $\ell_{\mathrm{sm}} := 2\gamma A H^3$.*

3. *(Subdifferentiability): For any $\pi \in \Pi$, the subgradient of $\Delta_{b_0}$ at $\pi$ is given by*

$$\partial \Delta_{b_0}(\pi) = \mathrm{conv}\{\nabla_\pi J_{c_n, P}(\pi) \mid n, P \in \mathcal{W}\},$$

*where $\mathrm{conv}\, B$ represents the convex hull of a set $B \subset \mathbb{R}^{SA}$.*

*Proof of Lipschitz continuity.*

$$|\Delta_{b_0}(\pi) - \Delta_{b_0}(\pi')| \leq \left| \max_{n \in [\![0,N]\!]} \{J_{c_n, \mathcal{U}}(\pi) - b_n\} - \max_{m \in [\![0,N]\!]} \{J_{m, \mathcal{U}}(\pi') - b_m\} \right|$$

$$\overset{(a)}{\leq} \max_{n \in [\![0,N]\!]} |J_{c_n, \mathcal{U}}(\pi) - b_n - (J_{c_n, \mathcal{U}}(\pi') - b_n)| \overset{(b)}{\leq} \ell_{\mathrm{Lp}}\|\pi - \pi'\|_2$$

where (a) uses Lemma 11 and (b) is due to Lemma 16. This concludes the proof of the Lipschitz continuity. □

*Proof of weak convexity.* The weak convexity of $\Delta_{b_0}(\pi) = \max_{n \in [\![0,N]\!]} J_{c_n, \mathcal{U}}(\pi) - b_n$ immediately follows from the weak convexity of $J_{c_n, \mathcal{U}}(\pi)$ due to Lemma 16 with Lemma 12. □

*Proof of subdifferentiability.* Suppose that $\mathcal{U}$ is a finite set. The claim directly follows from Lemma 10 with the weak convexity of $J_{c_n, P}(\pi)$ due to Lemma 16.

Suppose that $\mathcal{U}$ is a compact set such that, for any $\pi \in \Pi$, $\nabla J_{c_n, P}(\pi)$ is continuous with respect to $P \in \mathcal{U}$. The envelope theorem (Lemma 9) indicates that, for any $n \in [\![0, N]\!]$,

$$\partial J_{c_n, \mathcal{U}}(\pi) = \mathrm{conv}\left\{ \nabla J_{c_n, P}(\pi) \,\middle|\, P \in \underset{P \in \mathcal{U}}{\arg\max}\, J_{c_n, P}(\pi) - b_n \right\}.$$

Then, using Lemma 10 with the weak convexity of $J_{c_n, \mathcal{U}}(\pi)$ due to Lemma 18, we have

$$\partial \Delta_{b_0}(\pi) = \mathrm{conv}\left\{ g \,\middle|\, g \in \partial J_{c_n, \mathcal{U}}(\pi) \text{ where } n \in \underset{n \in [\![0,N]\!]}{\arg\max}\, J_{c_n, \mathcal{U}}(\pi) - b_n \right\}$$

$$= \mathrm{conv}\left\{ \nabla J_{c_n, P}(\pi) \,\middle|\, n, P \in \underset{(n,P) \in [\![0,N]\!] \times \mathcal{U}}{\arg\max}\, J_{c_n, P}(\pi) - b_n \right\}.$$

□

## H.4 PROOF OF THEOREM 4

*Proof of Theorem 4.* We introduce shorthands $\mathcal{G}$ and $\mathcal{W}$ such that

$$\mathcal{G} = \{\nabla J_{c_n, P}(\pi) \mid n, P \in \mathcal{W}_{b_0}(\pi)\} \text{ and } \mathcal{W} = \underset{(n,P) \in [\![0,N]\!] \times \mathcal{U}}{\arg\max}\, J_{c_n, P}(\pi) - b_n. \quad (35)$$

Let $\pi_{b_0}^\star \in \arg\min_{\pi \in \Pi} \Delta_{b_0}(\pi)$. For any $\pi \in \Pi$ and $b_0 \in \mathbb{R}$, we have

$$
\begin{aligned}
&\Delta_{b_0}(\pi) - \Delta_{b_0}(\pi_{b_0}^\star) \\
&= \left( \max_{n \in [\![0,N]\!]} \max_{P \in \mathcal{U}} J_{c_n,P}(\pi) - b_n \right) - \left( \max_{n \in [\![0,N]\!]} \max_{P \in \mathcal{U}} J_{c_n,P}(\pi_{b_0}^\star) - b_n \right) \\
&= \left( \min_{n,P \in \mathcal{W}} J_{c_n,P}(\pi) - b_n \right) - \left( \max_{n \in [\![0,N]\!]} \max_{P \in \mathcal{U}} J_{c_n,P}(\pi_{b_0}^\star) - b_n \right) \\
&\le \min_{n,P \in \mathcal{W}} \left( J_{c_n,P}(\pi) - b_n \right) - \left( J_{c_n,P}(\pi_{b_0}^\star) - b_n \right) \\
&= \min_{n,P \in \mathcal{W}} J_{c_n,P}(\pi) - J_{c_n,P}(\pi_{b_0}^\star) \\
&\le \min_{n,P \in \mathcal{W}} J_{c_n,P}(\pi) - \min_{\pi' \in \Pi} J_{c_n,P}(\pi') \\
&\overset{(a)}{\le} H \min_{n,P \in \mathcal{W}} \left\| \frac{d_P^{\pi_{n,\star}^\star,P}}{\mu} \right\|_\infty \underbrace{\max_{\pi' \in \Pi} \langle \pi - \pi', \nabla_\pi J_{c_n,P}(\pi) \rangle}_{\ge 0 \text{ when } \pi' \text{ is greedy to } \nabla_\pi J_{c_n,P}(\pi)} \\
&\le DH \min_{n,P \in \mathcal{W}} \max_{\pi' \in \Pi} \langle \pi - \pi', \nabla_\pi J_{c_n,P}(\pi) \rangle \\
&= DH \min_{g \in \mathcal{G}} \max_{\pi' \in \Pi} \langle \pi - \pi', g \rangle \ ,
\end{aligned}
\tag{36}
$$

where (a) uses Lemma 17.

The claim holds by showing that

$$
\min_{g \in \mathcal{G}} \max_{\pi' \in \Pi} \langle \pi - \pi', g \rangle = \min_{g \in \partial \Delta_{b_0}(\pi)} \max_{\pi' \in \Pi} \langle \pi - \pi', g \rangle \ .
\tag{37}
$$

Since $\mathrm{conv}\{\mathcal{G}\} = \partial \Delta_{b_0}(\pi)$ due to Lemma 18, Equation (37) holds when there exists a $g^\star \in \arg\min_{g \in \mathrm{conv}\{\mathcal{G}\}} \max_{\pi' \in \Pi} \langle \pi - \pi', g \rangle$ such that $g^\star \in \mathcal{G}$.

Let $z^\star \in \arg\max_{\pi' \in \Pi} \min_{g \in \mathrm{conv}\{\mathcal{G}\}} \langle \pi - \pi', g \rangle$. For any $g^\star \in \arg\min_{g \in \mathrm{conv}\{\mathcal{G}\}} \max_{\pi' \in \Pi} \langle \pi - \pi', g \rangle$, it holds that

$$
\begin{aligned}
\max_{\pi' \in \Pi} \langle \pi - \pi', g^\star \rangle &= \min_{g \in \mathrm{conv}\{\mathcal{G}\}} \max_{\pi' \in \Pi} \langle \pi - \pi', g \rangle \\
&\overset{(a)}{=} \max_{\pi' \in \Pi} \min_{g \in \mathrm{conv}\{\mathcal{G}\}} \langle \pi - \pi', g \rangle \\
&= \min_{g \in \mathrm{conv}\{\mathcal{G}\}} \langle \pi - z^\star, g \rangle \\
&\overset{(b)}{=} \min_{g \in \mathcal{G}} \langle \pi - z^\star, g \rangle
\end{aligned}
\tag{38}
$$

where (a) uses Sion's minimax theorem (Lemma 8) with the convexity of $\Pi$ and $\mathrm{conv}\{\mathcal{G}\}$, and (b) uses Lemma 15.

Note that

$$
\langle \pi - z^\star, g^\star \rangle \le \max_{\pi' \in \Pi} \langle \pi - \pi', g^\star \rangle \overset{(a)}{=} \min_{g \in \mathrm{conv}\{\mathcal{G}\}} \langle \pi - z^\star, g \rangle \le \langle \pi - z^\star, g^\star \rangle \ ,
\tag{39}
$$

where (a) is due to the third line of Equation (38). The inequality must be equality. Accordingly,

$$
\langle \pi - z^\star, g^\star \rangle \overset{(a)}{=} \max_{\pi' \in \Pi} \langle \pi - \pi', g^\star \rangle \overset{(b)}{=} \min_{g \in \mathcal{G}} \langle \pi - z^\star, g \rangle \ ,
$$

where (a) uses Equation (39) and (b) uses Equation (38). Therefore, $g^\star \in \mathcal{G}$ and thus Equation (37) holds. This concludes the proof. $\qquad\square$

## H.5 PROOF OF THEOREM 3

To facilitate the analysis with estimation error, we present a slightly modified version of the epigraph form. Let $\varepsilon \in \mathbb{R}$ be an admissible violation parameter. We introduce the following formulation:

$$
(\textbf{Epigraph}_\varepsilon) \quad J_\varepsilon^\star := \min_{b_0 \in [0,H]} b_0 \text{ such that } \Delta_{b_0}^\star \le \varepsilon \ .
\tag{40}
$$

Note that $J_\varepsilon^\star$ is monotonically decreasing in $\varepsilon$.

Additionally, we introduce a slightly generalized version of Theorem 2:

**Lemma 19.** *For any $\varepsilon_1, \varepsilon_2 \geq 0$, if $b_0$ and a policy $\pi \in \Pi$ satisfy $b_0 \leq J^\star + \varepsilon_2$ and $\Delta_{b_0}(\pi) \leq \varepsilon_1$, then $\pi$ is an $(\varepsilon_1 + \varepsilon_2)$-optimal policy.*

*Proof.* Note that $J_{c_0, \mathcal{U}}(\pi) \leq J^\star + \varepsilon_1 + \varepsilon_2$ and $J_{n, \mathcal{U}}(\pi) \leq b_n + \varepsilon_1$ for any $n \in [\![1, N]\!]$. The claim directly follows from Definition 1 and the fact that $J^\star = J_{c_0, \mathcal{U}}(\pi^\star)$. $\square$

For any $\bar{b}_0 \in [J_{\varepsilon_1}^\star, J^\star + \varepsilon_2]$ with some $\varepsilon_1, \varepsilon_2 \geq 0$, the subroutine returns a policy $\bar{\pi} = \mathscr{A}(\bar{b}_0)$ such that

$$\Delta_{\bar{b}_0}(\bar{\pi}) \overset{(a)}{\leq} \min_{\pi' \in \Pi} \Delta_{\bar{b}_0}(\pi') + \varepsilon_{\mathrm{opt}} \overset{(b)}{\leq} \min_{\pi' \in \Pi} \Delta_{J_{\varepsilon_1}^\star}(\pi') + \varepsilon_{\mathrm{opt}} \overset{(c)}{\leq} \varepsilon_1 + \varepsilon_{\mathrm{opt}},$$

where (a) is due to Assumption 4, (b) holds since $\Delta_{b_0}(\pi)$ is monotonically decreasing in $b_0$, and (c) follows from Equation (40). Consequently, by applying Lemma 19, $\bar{\pi}$ is $(\varepsilon_1 + \varepsilon_2 + \varepsilon_{\mathrm{opt}})$-optimal.

The following intermediate lemma guarantees that the search space of Algorithm 1 always contains such $\bar{b}_0$ with $\varepsilon_1 = \varepsilon_{\mathrm{est}}$ and $\varepsilon_2 = \varepsilon_{\mathrm{est}} + \varepsilon_{\mathrm{opt}}$.

**Lemma 20.** *Suppose that Algorithm 1 is run with algorithms $\widehat{J}_n$ and $\mathscr{A}$ that satisfy Assumptions 1 and 4. For any $k \in [\![0, K]\!]$, $[i^{(k)}, j^{(k)}] \cap [J_{\varepsilon_{\mathrm{est}}}^\star, J^\star + \varepsilon_{\mathrm{est}} + \varepsilon_{\mathrm{opt}}] \neq \emptyset$.*

*Proof.* The claim holds for $k = 0$. Suppose that the claim holds for a fixed $k$. Recall $\widehat{\Delta}^{(k)}$ defined in Equation (10). If $\widehat{\Delta}^{(k)} > 0$, it holds that

$$-\varepsilon_{\mathrm{est}} - \varepsilon_{\mathrm{opt}} \overset{(a)}{<} \widehat{\Delta}^{(k)} - \left| \widehat{\Delta}^{(k)} - \Delta_{b_0^{(k)}}(\pi^{(k)}) \right| - \varepsilon_{\mathrm{opt}} \leq \Delta_{b_0^{(k)}}(\pi^{(k)}) - \varepsilon_{\mathrm{opt}} \overset{(b)}{\leq} \Delta_{b_0^{(k)}}(\pi^\star) \overset{(c)}{=} J^\star - b_0^{(k)} \tag{41}$$

where (a) is due to Assumption 1 with $\widehat{\Delta}^{(k)} > 0$, (b) is due to Assumption 4, and (c) holds since $\pi^\star$ is a feasible policy. Combining this with the induction assumption and the update rule of Equation (11), we have $i^{(k+1)} = b_0^{(k)} \leq J^\star + \varepsilon_{\mathrm{est}} + \varepsilon_{\mathrm{opt}}$ and $J_{\varepsilon_{\mathrm{est}}}^\star \leq j^{(k+1)}$. Hence, $[i^{(k+1)}, j^{(k+1)}] \cap [J_{\varepsilon_{\mathrm{est}}}^\star, J^\star + \varepsilon_{\mathrm{est}} + \varepsilon_{\mathrm{opt}}] \neq \emptyset$ when $\widehat{\Delta}^{(k)} > 0$.

On the other hand, if $\widehat{\Delta}^{(k)} \leq 0$, we have

$$\min_\pi \Delta_{b_0^{(k)}}(\pi) \leq \Delta_{b_0^{(k)}}(\pi^{(k)}) \leq \widehat{\Delta}^{(k)} + \varepsilon_{\mathrm{est}} \leq \varepsilon_{\mathrm{est}}. \tag{42}$$

Since $b_0^{(k)}$ is the feasible solution to Equation (40), it holds that $J_{\varepsilon_{\mathrm{est}}}^\star \leq b_0^{(k)} = j^{(k+1)}$. Accordingly, we have $[i^{(k+1)}, j^{(k+1)}] \cap [J_{\varepsilon_{\mathrm{est}}}^\star, J^\star + \varepsilon_{\mathrm{est}} + \varepsilon_{\mathrm{opt}}] \neq \emptyset$. Therefore, the claim holds for any $k \in [\![0, K]\!]$. $\square$

We are now ready to prove Theorem 3.

*Proof of Theorem 3.* Note that $j^{(k)} - i^{(k)} \leq (j^{(0)} - i^{(0)}) 2^{-k} = H 2^{-k}$ due to the update rule of Equation (11). According to Lemma 20, we have $J_{\varepsilon_{\mathrm{est}}}^\star \leq j^{(K)} \leq J^\star + \varepsilon_{\mathrm{est}} + \varepsilon_{\mathrm{opt}} + H 2^{-K}$. Additionally, the returned policy $\pi_{\mathrm{ret}}$ satisfies

$$\Delta_{j^{(K)}}(\pi_{\mathrm{ret}}) \overset{(a)}{\leq} \min_{\pi \in \Pi} \Delta_{j^{(K)}}(\pi) + \varepsilon_{\mathrm{opt}} \overset{(b)}{\leq} \min_{\pi \in \Pi} \Delta_{J_{\varepsilon_{\mathrm{est}}}^\star}(\pi) + \varepsilon_{\mathrm{opt}} \leq \varepsilon_{\mathrm{est}} + \varepsilon_{\mathrm{opt}},$$

where (a) uses Assumption 4 and (b) is due to $J_{\varepsilon_{\mathrm{est}}}^\star \leq j^{(K)}$ and the fact that $\min_\pi \Delta_{b_0}(\pi)$ is monotonically decreasing in $b_0$. Applying this to Lemma 19 with $j^{(K)} \leq J^\star + \varepsilon_{\mathrm{est}} + \varepsilon_{\mathrm{opt}} + H 2^{-K}$ concludes the proof. $\square$

## H.6 Proof of Theorem 5

We prove the following restatement of Theorem 5 with concrete values.

**Theorem 6** (Restatement of Theorem 5). *Suppose Assumptions 2 and 5 hold. Suppose that Algorithm 2 is run with algorithms $\widehat{J}_n$ and $\widehat{J}^\partial$ that satisfy Assumption 1 and Assumption 3. Let*

$$C := \frac{\ell_{\mathrm{Lp}}}{2\ell_{\mathrm{sm}}} + 2DH\sqrt{S} = \frac{1}{2\gamma H\sqrt{A}} + 2DH\sqrt{S} \underbrace{=}_{when\ \gamma \approx 1} \widetilde{\mathcal{O}}(DH\sqrt{S}),$$

*where $\ell_{\mathrm{Lp}}$ and $\ell_{\mathrm{sm}}$ are defined in Lemma 18 and $D$ is defined in Theorem 4. Assume that the evaluators $\widehat{J}$ and $\widehat{J}^\partial$ are sufficiently accurate such that*

$$\varepsilon_{\mathrm{grd}} = C_\partial \varepsilon^2 \ \text{ where } \ C_\partial := \frac{1}{1024C^2\ell_{\mathrm{sm}}\sqrt{S}} \quad \text{and} \quad \varepsilon_{\mathrm{est}} = C_J \varepsilon^2 \ \text{ where } \ C_J := \frac{1}{1024C^2\ell_{\mathrm{sm}}}.$$

*Set $\alpha = C_\alpha \varepsilon^2$ and $T = C_T \varepsilon^{-4}$ such that*

$$C_\alpha := \frac{1}{64C^2\ell_{\mathrm{sm}}(\ell_{\mathrm{Lp}}^2 + \varepsilon_{\mathrm{grd}})} \quad \text{and} \quad C_T := 4096C^4\ell_{\mathrm{sm}}^2 S(\ell_{\mathrm{Lp}}^2 + \varepsilon_{\mathrm{grd}}^2) = \widetilde{\mathcal{O}}(D^4 S^3 A^3 H^{14}).$$

*Then, Algorithm 1 returns a policy $\pi^{(t^\star)}$ such that*

$$\Delta_{b_0}(\pi^{(t^\star)}) - \min_{\pi \in \Pi} \Delta_{b_0}(\pi) \le \varepsilon.$$

We first introduce the following useful lemma.

**Lemma 21.** *Let $\left(\mathrm{M}_{\frac{1}{2\ell_{\mathrm{sm}}}} \circ \Delta_{b_0}\right) : \pi \mapsto \min_{\pi' \in \Pi}\left\{\Delta_{b_0}(\pi') + \ell_{\mathrm{sm}}\|\pi - \pi'\|_2^2\right\}$ be the Moreau envelope function of $\Delta_{b_0}(\pi)$ with parameter $1/2\ell_{\mathrm{sm}}$. For any policy $\pi \in \Pi$,*

$$\Delta_{b_0}(\pi) - \min_{\pi' \in \Pi}\Delta_{b_0}(\pi') \le C\left\|\nabla\left(\mathrm{M}_{\frac{1}{2\ell_{\mathrm{sm}}}} \circ \Delta_{b_0}\right)(\pi)\right\|_2.$$

*Proof.* Define $\overline{\pi} := \arg\min_{\pi' \in \Pi}\Delta_{b_0}(\pi') + \ell_{\mathrm{sm}}\|\pi - \pi'\|_2^2$. According to Lemma 14 with $\tau = 1/2\ell_{\mathrm{sm}}$, there exists a subgradient $g \in \partial\Delta_{b_0}(\overline{\pi})$ such that, for any $\pi' \in \Pi$,

$$\begin{aligned}
\langle\overline{\pi} - \pi', g\rangle &\le \left\langle\nabla\left(\mathrm{M}_{\frac{1}{2\ell_{\mathrm{sm}}}} \circ \Delta_{b_0}\right)(\pi), \overline{\pi} - \pi'\right\rangle \\
&\overset{(a)}{\le} \left\|\nabla\left(\mathrm{M}_{\frac{1}{2\ell_{\mathrm{sm}}}} \circ \Delta_{b_0}\right)(\pi)\right\|_2 \|\overline{\pi} - \pi'\|_2 \\
&\overset{(b)}{\le} 2\sqrt{S}\left\|\nabla\left(\mathrm{M}_{\frac{1}{2\ell_{\mathrm{sm}}}} \circ \Delta_{b_0}\right)(\pi)\right\|_2,
\end{aligned} \tag{43}$$

where (a) is due to the Cauchy–Schwarz inequality and (b) uses that, for any $\pi' \in \Pi$

$$\begin{aligned}
\|\overline{\pi} - \pi'\|_2 &= \sqrt{\sum_{s\in\mathcal{S}}\sum_{a\in\mathcal{A}}(\overline{\pi}(s,a) - \pi'(s,a))^2} \le \sqrt{S}\max_{s\in\mathcal{S}}\sqrt{\sum_{a\in\mathcal{A}}(\overline{\pi}(s,a) - \pi'(s,a))^2} \\
&\le \sqrt{S}\max_{s\in\mathcal{S}}\sum_{a\in\mathcal{A}}|\overline{\pi}(s,a) - \pi'(s,a)| \le 2\sqrt{S}.
\end{aligned} \tag{44}$$

Let $\pi^\star_{b_0} \in \arg\min_{\pi\in\Pi}\Delta_{b_0}(\pi)$. Inserting this result into Theorem 4, we have

$$\begin{aligned}
\Delta_{b_0}(\overline{\pi}) - \Delta_{b_0}(\pi^\star_{b_0}) &\le DH\max_{\pi'\in\Pi}\langle\overline{\pi} - \pi', g\rangle \quad \forall g \in \partial\Delta_{b_0}(\overline{\pi}) \\
&\le 2DH\sqrt{S}\left\|\nabla\left(\mathrm{M}_{\frac{1}{2\ell_{\mathrm{sm}}}} \circ \Delta_{b_0}\right)(\pi)\right\|_2.
\end{aligned}$$

Therefore,

$$\begin{aligned}
\Delta_{b_0}(\pi) - \Delta_{b_0}(\pi^\star_{b_0}) &= \Delta_{b_0}(\pi) - \Delta_{b_0}(\overline{\pi}) + \Delta_{b_0}(\overline{\pi}) - \Delta_{b_0}(\pi^\star_{b_0}) \\
&\overset{(a)}{\le} \Delta_{b_0}(\pi) - \Delta_{b_0}(\overline{\pi}) + 2DH\sqrt{S}\left\|\nabla\left(\mathrm{M}_{\frac{1}{2\ell_{\mathrm{sm}}}} \circ \Delta_{b_0}\right)(\pi)\right\|_2 \\
&\overset{(b)}{\le} \ell_{\mathrm{Lp}}\|\pi - \overline{\pi}\|_2 + 2DH\sqrt{S}\left\|\nabla\left(\mathrm{M}_{\frac{1}{2\ell_{\mathrm{sm}}}} \circ \Delta_{b_0}\right)(\pi)\right\|_2 \\
&\overset{(c)}{\le} \frac{\ell_{\mathrm{Lp}}}{2\ell_{\mathrm{sm}}}\left\|\nabla\left(\mathrm{M}_{\frac{1}{2\ell_{\mathrm{sm}}}} \circ \Delta_{b_0}\right)(\pi)\right\|_2 + 2DH\sqrt{S}\left\|\nabla\left(\mathrm{M}_{\frac{1}{2\ell_{\mathrm{sm}}}} \circ \Delta_{b_0}\right)(\pi)\right\|_2,
\end{aligned}$$

where (a) is due to Theorem 4, (b) is due to the Lipschitz continuity by Lemma 18, and (c) uses Lemma 7. This concludes the proof. $\qquad\square$

**Lemma 22.** *Under the settings of Theorem 6,*

$$\sum_{t=0}^{T-1}\left\|\nabla\Big(\mathrm{M}_{\frac{1}{2\ell_{\mathrm{sm}}}}\circ\Delta_{b_0}\Big)(\pi^{(t)})\right\|_2^2 \le \frac{16\ell_{\mathrm{sm}}S}{\alpha} + 4T\Big(\alpha\ell_{\mathrm{sm}}(\ell_{\mathrm{Lp}}^2+\varepsilon_{\mathrm{grd}}) + 4\ell_{\mathrm{sm}}\varepsilon_{\mathrm{grd}}\sqrt{S} + 2\ell_{\mathrm{sm}}\varepsilon_{\mathrm{est}}\Big).$$

*Proof.* Recall that

$$\pi^{(t+1)} \in \arg\min_{\pi\in\Pi}\left\langle g^{(t)},\pi-\pi^{(t)}\right\rangle + \frac{1}{2\alpha}\left\|\pi-\pi^{(t)}\right\|^2 = \mathrm{Proj}_\Pi\Big(\pi^{(t)}-\alpha g^{(t)}\Big).$$

Define $\bar\pi^{(t)} := \arg\min_{\pi'}\Delta_{b_0}(\pi') + \ell_{\mathrm{sm}}\left\|\pi^{(t)}-\pi'\right\|_2^2$. Then, we have

$$\begin{aligned}
\Big(\mathrm{M}_{\frac{1}{2\ell_{\mathrm{sm}}}}\circ\Delta_{b_0}\Big)(\pi^{(t+1)}) &= \min_{\pi\in\Pi}\Delta_{b_0}(\pi) + \ell_{\mathrm{sm}}\left\|\pi^{(t+1)}-\pi\right\|_2^2 \\
&\le \Delta_{b_0}(\bar\pi^{(t)}) + \ell_{\mathrm{sm}}\left\|\pi^{(t+1)}-\bar\pi^{(t)}\right\|_2^2 \\
&= \Delta_{b_0}(\bar\pi^{(t)}) + \ell_{\mathrm{sm}}\left\|\mathrm{Proj}_\Pi\Big(\pi^{(t)}-\alpha g^{(t)}\Big) - \mathrm{Proj}_\Pi\Big(\bar\pi^{(t)}\Big)\right\|_2^2 \\
&\le \Delta_{b_0}(\bar\pi^{(t)}) + \ell_{\mathrm{sm}}\left\|\pi^{(t)}-\alpha g^{(t)}-\bar\pi^{(t)}\right\|_2^2 \\
&= \underbrace{\Delta_{b_0}(\bar\pi^{(t)}) + \ell_{\mathrm{sm}}\left\|\pi^{(t)}-\bar\pi^{(t)}\right\|_2^2}_{=\Big(\mathrm{M}_{\frac{1}{2\ell_{\mathrm{sm}}}}\circ\Delta_{b_0}\Big)(\pi^{(t)})} + \underbrace{2\ell_{\mathrm{sm}}\alpha\left\langle g^{(t)},\bar\pi^{(t)}-\pi^{(t)}\right\rangle}_{=:①} + \underbrace{\alpha^2\ell_{\mathrm{sm}}\left\|g^{(t)}\right\|_2^2}_{=:②}.
\end{aligned}$$

$$(45)$$

We further upper bound ① and ②. Recall $n^{(t)} \in \arg\max_{n\in[\![0,N]\!]}\widehat{J}_n(\pi^{(t)}) - b_n$ and $g^{(t)} = \widehat{J}_{n^{(t)}}^\partial(\pi^{(t)})$. Due to Assumption 3, there exists a vector $g'\in\mathbb{R}^{SA}$ that satisfies

$$g' \in \left\{\nabla J_{c_{n^{(t)}},P^{(t)}}(\pi^{(t)}) \;\middle|\; P^{(t)}\in\arg\max_{P\in\mathcal{U}}J_{c_{n^{(t)}},P}(\pi^{(t)})\right\} \text{ and } \left\|g^{(t)}-g'\right\|_2^2 \le \varepsilon_{\mathrm{grd}}^2. \quad (46)$$

Thus, using Lemma 16, we have $② \le \|g'\|_2^2 + \left\|g^{(t)}-g'\right\|_2^2 \le \ell_{\mathrm{Lp}}^2 + \varepsilon_{\mathrm{grd}}^2$. Furthermore, we have

$$\begin{aligned}
① &= \left\langle g^{(t)},\bar\pi^{(t)}-\pi^{(t)}\right\rangle = \left\langle g',\bar\pi^{(t)}-\pi^{(t)}\right\rangle + \left\langle g^{(t)}-g',\bar\pi^{(t)}-\pi^{(t)}\right\rangle \\
&\overset{(a)}{\le} \left\langle g',\bar\pi^{(t)}-\pi^{(t)}\right\rangle + \left\|g^{(t)}-g'\right\|_2\left\|\bar\pi^{(t)}-\pi^{(t)}\right\|_2 \overset{(b)}{\le} \left\langle g',\bar\pi^{(t)}-\pi^{(t)}\right\rangle + 2\varepsilon_{\mathrm{grd}}\sqrt{S}
\end{aligned}$$

where (a) is due to the Cauchy–Schwarz inequality and (b) uses Equation (44).

By inserting the above inequalities to Equation (45), using $g'$ defined in Equation (46), we have

$$\begin{aligned}
③ &:= 2\ell_{\mathrm{sm}}\alpha\left\langle g',\pi^{(t)}-\bar\pi^{(t)}\right\rangle \\
&\le \Big(\mathrm{M}_{\frac{1}{2\ell_{\mathrm{sm}}}}\circ\Delta_{b_0}\Big)(\pi^{(t)}) - \Big(\mathrm{M}_{\frac{1}{2\ell_{\mathrm{sm}}}}\circ\Delta_{b_0}\Big)(\pi^{(t+1)}) + \alpha^2\ell_{\mathrm{sm}}(\ell_{\mathrm{Lp}}^2+\varepsilon_{\mathrm{grd}}^2) + 4\ell_{\mathrm{sm}}\alpha\varepsilon_{\mathrm{grd}}\sqrt{S}.
\end{aligned}$$

$$(47)$$

Next, we are going to derive the lower bound of ③. Define $\Delta_{b_0}^{(t)}(\pi)$ such that

$$\Delta_{b_0}^{(t)}(\pi) := J_{c_{n^{(t)}},\mathcal{U}}(\pi) - b_{n^{(t)}}. \quad (48)$$

Additionally, let $\delta_n$ and $\widehat{\delta}_n$ be shorthand such that $\delta_n := J_{c_n,\mathcal{U}}(\pi) - b_n$ and $\widehat{\delta}_n := \widehat{J}_n(\pi) - b_n$. Then, Due to Assumption 1, for any $\pi$, we have

$$\begin{aligned}
\left|\Delta_{b_0}^{(t)}(\pi) - \Delta_{b_0}(\pi)\right| &\overset{(a)}{\le} \underbrace{\left|\Delta_{b_0}^{(t)}(\pi) - \widehat{\delta}_{n^{(t)}}\right|}_{\le\varepsilon_{\mathrm{est}}\text{ by Assumption 1}} + \left|\max_{n\in[\![0,N]\!]}\widehat{\delta}_n - \Delta_{b_0}(\pi)\right| \\
&\overset{(b)}{\le} \varepsilon_{\mathrm{est}} + \max_{n\in[\![0,N]\!]}\left|\widehat{\delta}_n - \delta_n\right| \le 2\varepsilon_{\mathrm{est}}.
\end{aligned}$$

$$(49)$$

where (a) is due to the definition of $n^{(t)}$ and (b) uses Lemma 11.

Due to the weak convexity of $\Delta_{b_0}^{(t)}(\pi)$ with respect to $\pi$ (Lemma 18) and since $g' \in \partial \Delta_{b_0}^{(t)}(\pi^{(t)})$,

$$\text{③}/2\ell_{\text{sm}}\alpha = \left\langle g', \pi^{(t)} - \bar{\pi}^{(t)} \right\rangle$$

$$\geq \Delta_{b_0}^{(t)}(\pi^{(t)}) - \Delta_{b_0}^{(t)}(\bar{\pi}^{(t)}) - \frac{\ell_{\text{sm}}}{2}\left\| \bar{\pi}^{(t)} - \pi^{(t)} \right\|_2^2$$

$$\geq -\underbrace{\left\lfloor \Delta_{b_0}^{(t)}(\pi^{(t)}) - \Delta_{b_0}(\pi^{(t)}) \right\rfloor}_{\leq \varepsilon_{\text{est}} \text{ by Equation (49)}} - \underbrace{\left\lfloor \Delta_{b_0}(\bar{\pi}^{(t)}) - \Delta_{b_0}^{(t)}(\bar{\pi}^{(t)}) \right\rfloor}_{\leq \varepsilon_{\text{est}} \text{ by Equation (49)}} + \Delta_{b_0}(\pi^{(t)}) - \Delta_{b_0}(\bar{\pi}^{(t)}) - \frac{\ell_{\text{sm}}}{2}\left\| \bar{\pi}^{(t)} - \pi^{(t)} \right\|_2^2$$

$$= \Delta_{b_0}(\pi^{(t)}) + \ell_{\text{sm}}\left\| \pi^{(t)} - \pi^{(t)} \right\|_2^2 - \Delta_{b_0}(\bar{\pi}^{(t)}) - \ell_{\text{sm}}\left\| \bar{\pi}^{(t)} - \pi^{(t)} \right\|_2^2 + \frac{\ell_{\text{sm}}}{2}\left\| \bar{\pi}^{(t)} - \pi^{(t)} \right\|_2^2 - 2\varepsilon_{\text{est}}$$

$$= \Delta_{b_0}(\pi^{(t)}) + \ell_{\text{sm}}\left\| \pi^{(t)} - \pi^{(t)} \right\|_2^2 - \min_{\pi' \in \Pi}\left( \Delta_{b_0}(\pi') + \ell_{\text{sm}}\left\| \pi' - \pi^{(t)} \right\|_2^2 \right) + \frac{\ell_{\text{sm}}}{2}\left\| \bar{\pi}^{(t)} - \pi^{(t)} \right\|_2^2 - 2\varepsilon_{\text{est}}$$

$$\geq \frac{\ell_{\text{sm}}}{2}\left\| \bar{\pi}^{(t)} - \pi^{(t)} \right\|_2^2 - 2\varepsilon_{\text{est}}$$

$$\overset{(a)}{=} \frac{\ell_{\text{sm}}}{2}\left\| \frac{1}{2\ell_{\text{sm}}}\nabla\left( M_{\frac{1}{2\ell_{\text{sm}}}} \circ \Delta_{b_0} \right)(\pi^{(t)}) \right\|_2^2 = \frac{1}{8\ell_{\text{sm}}}\left\| \nabla\left( M_{\frac{1}{2\ell_{\text{sm}}}} \circ \Delta_{b_0} \right)(\pi^{(t)}) \right\|_2^2 - 2\varepsilon_{\text{est}} ,$$

where (a) uses Lemma 7. By inserting this to Equation (47),

$$\frac{\alpha}{4}\left\| \nabla\left( M_{\frac{1}{2\ell_{\text{sm}}}} \circ \Delta_{b_0} \right)(\pi^{(t)}) \right\|_2^2 - 8\alpha\ell_{\text{sm}}\varepsilon_{\text{est}}$$

$$\leq \left( M_{\frac{1}{2\ell_{\text{sm}}}} \circ \Delta_{b_0} \right)(\pi^{(t)}) - \left( M_{\frac{1}{2\ell_{\text{sm}}}} \circ \Delta_{b_0} \right)(\pi^{(t+1)}) + \alpha^2\ell_{\text{sm}}(\ell_{\text{Lp}}^2 + \varepsilon_{\text{grd}}^2) + 4\alpha\ell_{\text{sm}}\varepsilon_{\text{grd}}\sqrt{S} .$$

By taking summation over $\sum_{t=0}^{T-1}$,

$$\frac{\alpha}{4}\sum_{t=0}^{T-1}\left\| \nabla\left( M_{\frac{1}{2\ell_{\text{sm}}}} \circ \Delta_{b_0} \right)(\pi^{(t)}) \right\|_2^2 \leq \left( M_{\frac{1}{2\ell_{\text{sm}}}} \circ \Delta_{b_0} \right)(\pi^{(0)}) - \left( M_{\frac{1}{2\ell_{\text{sm}}}} \circ \Delta_{b_0} \right)(\pi^{(T)})$$

$$+ T\left( \alpha^2\ell_{\text{sm}}(\ell_{\text{Lp}}^2 + \varepsilon_{\text{grd}}^2) + 4\alpha\ell_{\text{sm}}\varepsilon_{\text{grd}}\sqrt{S} + 8\alpha\ell_{\text{sm}}\varepsilon_{\text{est}} \right) .$$

Note that

$$\left( M_{\frac{1}{2\ell_{\text{sm}}}} \circ \Delta_{b_0} \right)(\pi^{(0)}) - \left( M_{\frac{1}{2\ell_{\text{sm}}}} \circ \Delta_{b_0} \right)(\pi^{(T)})$$

$$= \min_{\pi \in \Pi}\left\{ \Delta_{b_0}(\pi) + \ell_{\text{sm}}\left\| \pi^{(0)} - \pi \right\|_2^2 \right\} - \min_{\pi \in \Pi}\left\{ \Delta_{b_0}(\pi) + \ell_{\text{sm}}\left\| \pi^{(T)} - \pi \right\|_2^2 \right\}$$

$$= \Delta_{b_0}(\bar{\pi}^{(0)}) + \ell_{\text{sm}}\left\| \pi^{(0)} - \bar{\pi}^{(0)} \right\|_2^2 - \Delta_{b_0}(\bar{\pi}^{(T)}) - \ell_{\text{sm}}\left\| \pi^{(T)} - \bar{\pi}^{(T)} \right\|_2^2$$

$$\leq \Delta_{b_0}(\bar{\pi}^{(T)}) + \ell_{\text{sm}}\left\| \pi^{(0)} - \bar{\pi}^{(T)} \right\|_2^2 - \Delta_{b_0}(\bar{\pi}^{(T)}) - \ell_{\text{sm}}\left\| \pi^{(T)} - \bar{\pi}^{(T)} \right\|_2^2$$

$$\leq \ell_{\text{sm}}\left\| \pi^{(0)} - \bar{\pi}^{(T)} \right\|_2^2 \leq 4\ell_{\text{sm}}S ,$$

where the last inequality uses Equation (44).

By combining all the results, we obtain

$$\sum_{t=0}^{T-1}\left\| \nabla\left( M_{\frac{1}{2\ell_{\text{sm}}}} \circ \Delta_{b_0} \right)(\pi^{(t)}) \right\|_2^2 \leq \frac{16\ell_{\text{sm}}S}{\alpha} + 4T\left( \alpha\ell_{\text{sm}}(\ell_{\text{Lp}}^2 + \varepsilon_{\text{grd}}^2) + 4\ell_{\text{sm}}\varepsilon_{\text{grd}}\sqrt{S} + 2\ell_{\text{sm}}\varepsilon_{\text{est}} \right) .$$

This concludes the proof. □

We are now ready to prove Theorem 6.

*Proof of Theorem 6.* Let $\pi_{b_0}^\star \in \arg\min_{\pi \in \Pi} \Delta_{b_0}(\pi)$. Then,

$$\min_{t \in [\![0,T-1]\!]} \Delta_{b_0}(\pi^{(t)}) - \Delta_{b_0}(\pi_{b_0}^\star)$$

$$\leq \frac{1}{T} \sum_{t=0}^{T-1} \Delta_{b_0}(\pi^{(t)}) - \Delta_{b_0}(\pi_{b_0}^\star)$$

$$\stackrel{(a)}{\leq} \frac{1}{T} C \sum_{t=0}^{T-1} \left\| \nabla \left( \mathrm{M}_{\frac{1}{2\ell_{\mathrm{sm}}}} \circ \Delta_{b_0} \right)(\pi) \right\|_2$$

$$\leq C \sqrt{\frac{1}{T} \sum_{t=0}^{T-1} \left\| \nabla \left( \mathrm{M}_{\frac{1}{2\ell_{\mathrm{sm}}}} \circ \Delta_{b_0} \right)(\pi) \right\|_2^2}$$

$$\stackrel{(b)}{\leq} C \sqrt{\frac{16\ell_{\mathrm{sm}}S}{T\alpha} + 4\left(\alpha\ell_{\mathrm{sm}}(\ell_{\mathrm{Lp}}^2 + \varepsilon_{\mathrm{grd}}^2) + 4\ell_{\mathrm{sm}}\varepsilon_{\mathrm{grd}}\sqrt{S} + 2\ell_{\mathrm{sm}}\varepsilon_{\mathrm{est}}\right)}$$

$$\stackrel{(c)}{=} C \sqrt{\frac{16\ell_{\mathrm{sm}}S}{\delta\sqrt{T}} + \frac{4\delta}{\sqrt{T}}\ell_{\mathrm{sm}}(\ell_{\mathrm{Lp}}^2 + \varepsilon_{\mathrm{grd}}^2) + 16\ell_{\mathrm{sm}}\varepsilon_{\mathrm{grd}}\sqrt{S} + 8\ell_{\mathrm{sm}}\varepsilon_{\mathrm{est}}}$$

$$\stackrel{(d)}{\leq} 4C\sqrt{\ell_{\mathrm{sm}}S\delta^{-1}}T^{-\frac{1}{4}} + 2C\sqrt{\ell_{\mathrm{sm}}(\ell_{\mathrm{Lp}}^2 + \varepsilon_{\mathrm{grd}}^2)\delta}T^{-\frac{1}{4}} + 4C\sqrt{\ell_{\mathrm{sm}}}\left(S^{\frac{1}{4}}\sqrt{\varepsilon_{\mathrm{grd}}} + \sqrt{\varepsilon_{\mathrm{est}}}\right)$$

$$\stackrel{(e)}{=} 4C\sqrt{\ell_{\mathrm{sm}}}S^{\frac{1}{4}}(\ell_{\mathrm{Lp}}^2 + \varepsilon_{\mathrm{grd}}^2)^{\frac{1}{4}}T^{-\frac{1}{4}} + 4C\sqrt{\ell_{\mathrm{sm}}}\left(S^{\frac{1}{4}}\sqrt{\varepsilon_{\mathrm{grd}}} + \sqrt{\varepsilon_{\mathrm{est}}}\right) .$$

where (a) uses Lemma 21, (b) uses Lemma 22, (c) replaces $\alpha$ with $\delta/\sqrt{T}$, (d) uses $\sqrt{x+y} \leq \sqrt{x} + \sqrt{y}$, and (e) sets $\delta = \sqrt{S/(\ell_{\mathrm{Lp}}^2 + \varepsilon_{\mathrm{grd}}^2)}$.

Therefore, when $\varepsilon_{\mathrm{est}}, \varepsilon_{\mathrm{grd}}, \alpha$, and $T$ satisfy:

$$\varepsilon_{\mathrm{grd}} = \frac{\varepsilon^2}{1024C^2\ell_{\mathrm{sm}}\sqrt{S}} \ , \ \varepsilon_{\mathrm{est}} = \frac{\varepsilon^2}{1024C^2\ell_{\mathrm{sm}}} \ ,$$

$$T = 4096C^4\ell_{\mathrm{sm}}^2 S(\ell_{\mathrm{Lp}}^2 + \varepsilon_{\mathrm{grd}}^2)\varepsilon^{-4} \ , \ \text{and} \ \alpha = \frac{\delta}{\sqrt{T}} = \frac{\varepsilon^2}{64C^2\ell_{\mathrm{sm}}(\ell_{\mathrm{Lp}}^2 + \varepsilon_{\mathrm{grd}}^2)} \ ,$$

we have

$$\min_{t \in [\![0,T-1]\!]} \Delta_{b_0}(\pi^{(t)}) \leq \Delta_{b_0}(\pi_{b_0}^\star) + \frac{3}{4}\varepsilon \ .$$

Finally, $t^\star \in \arg\min_{t \in [\![0,T-1]\!]} \widehat{\Delta}^{(t)}$ satisfies that

$$\Delta_{b_0}(\pi^{(t^\star)}) = \widehat{\Delta}^{(t^\star)} + \Delta_{b_0}(\pi^{(t^\star)}) - \widehat{\Delta}^{(t^\star)}$$

$$\leq \min_{t \in [\![0,T-1]\!]} \widehat{\Delta}^{(t)} + \varepsilon_{\mathrm{est}}$$

$$\leq \min_{t \in [\![0,T-1]\!]} \Delta_{b_0}(\pi^{(t)}) + \widehat{\Delta}^{(t)} - \Delta_{b_0}(\pi^{(t)}) + \varepsilon_{\mathrm{est}}$$

$$\leq \min_{t \in [\![0,T-1]\!]} \Delta_{b_0}(\pi^{(t)}) + 2\varepsilon_{\mathrm{est}}$$

$$\leq \Delta_{b_0}(\pi_{b_0}^\star) + \frac{3}{4}\varepsilon + 2\varepsilon_{\mathrm{est}}$$

$$\leq \Delta_{b_0}(\pi_{b_0}^\star) + \varepsilon \ .$$

This concludes the proof. $\square$

