# OpenReview forum: "Near-Optimal Policy Identification in Robust Constrained Markov Decision Processes via Epigraph Form"
_ICLR.cc/2025/Conference — ICLR 2025 Poster_

### Official Review · Reviewer_FLm4 · 2024-11-03

**Soundness:** 4
**Presentation:** 3
**Contribution:** 3
**Rating:** 6
**Confidence:** 4

**Summary:**

The paper introduces Epigraph Robust Constrained Policy Gradient Search, an algorithm designed to identify near-optimal policies in RCMDPs. The epigraph form can address gradient conflicts and theoretical techniques of prior approaches, ensuring convergence to an
$ϵ-optimal$ policy across uncertain environments while satisfying safety constraints.

**Strengths:**

This paper is well structured and written, it addresses the difficulties of applying existing CMDP algorithms in RCMDP by providing the first algorithm with theoretical performance guarantees for near-optimal policy identification. The proposed epigraph form effectively resolves gradient conflicts inherent in the Lagrangian formulation. They also include a toy example to show the challenges in RCMDP which makes the paper easy to follow and the empirical results demonstrate that EpiRC-PGS outperforms baselines in various RCMDP settings.

**Weaknesses:**

The proposed algorithm’s double-loop structure may become computationally intensive in environments with high-dimensional action or state spaces, limiting real-time applicability.  The binary search and the projection in the policy updates are highly computationally inefficient. Assumption 2 requires an initial distribution over all the states, which is impractical, especially in the RCMDP setting, since some states are unsafe. Since the design of the algorithm is based on the monotonical property of $\Delta_b^*,$ it requires assumptions 3,4,5 to have accurate estimations. The empirical evaluation could benefit from complicated or additional real-world applications to validate the robustness in practical MDP scenarios.

comments:
It is better to discuss all the assumptions before presenting the main theorem.

**Questions:**

In robust RL, sample complexity and convergence results typically depend on the gap between the transition kernels in the uncertainty set. Could the authors elaborate on how the results depend on this gap or specify which parts of the analysis are impacted by it?

---

> ### Author Response · Authors · 2024-11-18
>
> **Thank you very much for the valuable comments. We are addressing the concerns raised in the review.**
>
> > comments: It is better to discuss all the assumptions before presenting the main theorem.
>
> Thank you very much for pointing this out. We noticed that we have a typo in Theorem 3 in the previous draft, which referred to the later Assumption 5; the correct assumptions are "Assumption 1 and 2" in Theorem 3. We have corrected this typo in the current revision. Additionally, we have made improvements to the readability and clarity of the draft. For further details, please see the [General Response.](https://openreview.net/forum?id=G5sPv4KSjR&noteId=kPGMvQckyn)
>
> ## About Weakness1
>
> > The proposed algorithm’s double-loop structure may become computationally intensive in environments with high-dimensional action or state spaces, limiting real-time applicability. The binary search and the projection in the policy updates are highly computationally inefficient.
>
> We note that binary search is generally exponentially fast, making our algorithm computationally efficient from a theoretical standpoint. Yet, as you mention, the double loop may present a computational burden in practice. Thus, our Appendix B offers a potential empirical solution and theoretical challenge. Specifically, it provides a Lagrangian extension of our epigraph form, and describes the hardness to show its strong duality.
>
> **Additionally, we remark that algorithmic development in MDPs often begins with double-loop algorithms [2] [3] before extending to a single-loop architecture [4] [5].** Thus, we believe our research, while currently based on a double-loop structure, represents an important stepping stone for future advancements in the RCMDP study.
>
> [2] A Dual Approach to Constrained Markov Decision Processes with Entropy Regularization: https://arxiv.org/abs/2110.08923
>
> [3] Policy Gradient for Rectangular Robust Markov Decision Processes: https://arxiv.org/abs/2212.10439
>
> [4] Last-Iterate Convergent Policy Gradient Primal-Dual Methods for Constrained MDPs: https://arxiv.org/abs/2306.11700
>
> [5] A Single-Loop Robust Policy Gradient Method for Robust Markov Decision Processes: https://arxiv.org/abs/2406.00274
>
> ## About Weakness 2
>
> > Assumption 2 requires an initial distribution over all the states, which is impractical, especially in the RCMDP setting, since some states are unsafe.
>
> The initial distribution coverage assumption is an almost necessary condition for the policy gradient approach[1]. As demonstrated in Remark 1 of [6], the term $D$ in Theorem 4 cannot be omitted. **Therefore, without the coverage assumption, $D$ would become infinite, and as a result, global convergence in policy gradient methods cannot be guaranteed.**
>
> However, as [our answer to Question 2 in the reviewer YFQK](https://openreview.net/forum?id=G5sPv4KSjR&noteId=HAzxT4rtkt) explains, our policy subgradient method seems to be the only viable approach. Exploring whether the coverage assumption is an interesting future work, but due to the above reasons, we currently expect that this assumption is almost necessary for the RCMDP problem.
>
> [6] On the Global Convergence Rates of Softmax Policy Gradient Methods: https://arxiv.org/abs/2005.06392
>
> ## About Weakness 3
>
> > Since the design of the algorithm is based on the monotonical property of Δb∗, it requires assumptions 3,4,5 to have accurate estimations.
>
> Computing optimal policy must require accurate policy evaluation. For example, when we do not have an accurate policy evaluator, identifying optimal policy is impossible in robust MDPs [7]. As a simple example, consider a simple MDP where we have two actions a1 and a2. If do not know the expected return values from a1 and a2, we cannot answer which action results in a better return.
> Therefore, we do not consider these accurate evaluator assumptions be weaknesses.
>
> However, we recognize that the previous draft may not well convey this explanation. **To enhance intuition about the assumptions, we have updated the notations,** particularly those related to the evaluators of $\Delta_{b_0}(\pi)$. In Assumption 1 of the new draft, we introduce the robust policy evaluator $\widehat{J}_n(\pi)$, which estimates the robust return value of $\pi$. Similarly, a new notation for the robust policy gradient evaluator is introduced in Assumption 3. These updates aim to clarify that our EpiRC-PGS algorithm can be implemented using existing robust policy gradient techniques.
>
> [7] Robust Markov Decision Processes: https://pubsonline.informs.org/doi/abs/10.1287/moor.1120.0566?journalCode=moor

---

> ### Author Response · Authors · 2024-11-18
>
> ## About Weakness 4
>
> > The empirical evaluation could benefit from complicated or additional real-world applications to validate the robustness in practical MDP scenarios. As you suggest, conducting such practical experiments is an important future work.
>
> Since our focus is on proposing first theoretically guaranteed algorithm for RCMDP and its theoretical analysis, the aim of experiments is to confirm the theoretical guarantees of EpiRC-PGS (Corollary 1) and demonstrate the limitations of the Lagrangian method (Theorem 1) in identifying near-optimal policies, rather than to showcase the effectiveness in practical MDP benchmarks. We believe our experimental results (Figure 3) sufficiently achieve this aim, as the Lagrangian baselines failed to identify feasible policies, while EpiRC-PGS successfully identified them.
>
> Therefore, we believe the current experiments are sufficient for our study.
>
> ## About Question
> > In robust RL, sample complexity and convergence results typically depend on the gap between the transition kernels in the uncertainty set. Could the authors elaborate on how the results depend on this gap or specify which parts of the analysis are impacted by it?
>
> In typical robust MDP analyses, the dependence on the gap primarily arises from the robust policy evaluation step. We abstract this evaluation step through Assumptions 1 and 2 (in the current draft), thereby eliminating the need for specific structural assumptions about the uncertainty set.
>
> This abstraction is a technical strength because **it enables future theoretical research on RCMDPs to extend their results in specific uncertainty sets to our general theoretical results.** We note that a similar abstraction has been used in previous robust MDP literature [8].
>
> [8] Policy Gradient in Robust MDPs with Global Convergence Guarantee: https://arxiv.org/abs/2212.10439
>
> =================
>
>
> **Finally, we want to thank the reviewer again for the detailed and valuable comments. If our response resolves your concerns satisfactorily, we want to kindly ask the reviewer to consider raising the score rating of our work.**
>
> **Please let us know if you have any further questions, and we will be happy to answer them. Additionally, if you have any further concerns, we will be happy to make changes and improve the paper.**

---

> > ### Comment · Reviewer_FLm4 · 2024-11-29
> >
> > I thank the authors for their response. I think the current score represents my evaluation of the paper.

---

### Official Review · Reviewer_yT6U · 2024-11-04

**Soundness:** 3
**Presentation:** 3
**Contribution:** 3
**Rating:** 8
**Confidence:** 4

**Summary:**

The paper present the first algorithm to solve Robust Constrained Markov Decision Process (RCMDP). The proposed algorithm achieves $\epsilon$-optimality with $\mathcal{O}(\epsilon^{-4})$ iteration complexity provided the uncertainty set is known apriori. The paper uses epigraph method which is different from the standard Lagrange approach used in the literature.

**Strengths:**

1. Good intuition and presentation.
2. Mathematically sound results.

**Weaknesses:**

**Minor Comments**

1. Assumption $5$ is used in Theorem $3$, however it appears later. Please correct it.

**Major Comments**

1. To my understanding, this algorithm works if the uncertainty set is exactly characterized and known (otherwise, getting a subgradient evaluator might be difficult). However, one of the potential application of Robust MDP is to ensure that policies trained on an unknown model $P_0$ does not deviate too much if the actual model $P$ is close to the model $P_0$ in some sense. In this case, one can probably guess the shape of the uncertainty set but cannot fully characterize it since $P_0$ is unknown. For example, if $P$ lies in a ball of radius $r$ centered around $P_0$, then the shape of the uncertainty set is a high dimensional sphere but its centre, $P_0$ is unknown. Information about $P_0$ is available only through a simulator with model $P_0$. My question is: how can the proposed algorithm provide guarantees in such cases? If it cannot, then that should be highlighted in the abstract and introduction as a drawback.

**Questions:**

1. Please categorize the related works into two groups: one that assumes the uncertainty set is fully characterized apriori (see the weakness above) and others that do not. This will help readers put the work into context.

2. What are the challenges in extending this result to the case where the uncertainty set is not fully known apriori?

3. Does the algorithm scale with the size of the state space? Does it scale with the size of the uncertainty set? Some comments on these should be provided in the paper.

4. What is the typical computational complexity to generate a subgradient estimate with $\mathcal{O}(\epsilon^2)$ error? Such complexities should be elaborated in the paper.

---

> ### Author Response · Authors · 2024-11-18
>
> **Thank you very much for the valuable comments. We are addressing the concerns raised in the review.**
>
> ## About Minor Comment
>
> > Assumption 5 is used in Theorem 3, however it appears later. Please correct it.
>
> Thank you very much for pointing this out. This is indeed a typo; the correct assumptions are "Assumption 1 and 2" in Theorem 3. Our current revision have corrected this typo. Additionally, we have made improvements to the readability and clarity of the draft. For further details, please see the [General Response](https://openreview.net/forum?id=G5sPv4KSjR&noteId=kPGMvQckyn).
>
> ## Major Comments
>
> > To my understanding, this algorithm works if the uncertainty set is exactly characterized and known (otherwise, getting a subgradient evaluator might be difficult).
>
> Our algorithms and theoretical guarantees do not necessarily require the Uncertainty set to be known. In fact, in Assumptions 1 and 3 in the current revision (3 and 4 in the previous draft), we only assume that robust policy evaluation and robust policy gradients can be approximated with sufficient accuracy. It does not matter whether the approximation error arises from computational errors or from sample approximations via simulators or other means.
>
> It is worth noting that, **we employed this abstracted assumption to allow the future RCMDP papers can easily reuse our result.** Our current draft emphasizes this point at line 306. Specifically, it is described as:
>
> “Note that without any assumptions about the uncertainty set $\mathcal{U}$, solving an RCMDP is NP-hard. However, imposing concrete structures on $\mathcal{U}$ can restrict the applicability of EpiRC-PGS. To enable our algorithm to handle a broader class of $\mathcal{U}$, we consider $\mathcal{U}$ where we can approximate the robust return value $J_{c_n, \mathcal{U}}(\pi)$ and its subgradient $\partial J_{c_n, \mathcal{U}}(\pi)$ as follows: … “
>
> > My question is: how can the proposed algorithm provide guarantees in such cases? If it cannot, then that should be highlighted in the abstract and introduction as a drawback.
>
> **As long as you can approximate the uncertainty set with sufficiently accurate robust policy evaluation and robust policy gradient estimations, our paper can offer theoretical guarantees.** Such uncertainty set construction is also important and has been studied in several papers [1], but is not within the scope of the current paper.
>
> For example, consider the simulator setting. A naive extension should be estimating $P_0$ using samples from $P_0$ in a model-based manner like [1]. By estimating $P_0$ to a sufficiently accurate level, it should be possible to control the estimation error in robust policy evaluation and robust policy gradients under the estimated transition kernel $\hat{P}$. By substituting the controlled estimation errors into $\varepsilon_{est}$ and $\varepsilon_{grd}$ in Assumptions 3 and 4 (Assumptions 1 and 3 in the current revision), our Corollary 1 becomes applicable, providing theoretical guarantees even in cases where $ P_0$ is unknown.
>
> Like this, if there are some "$\varepsilon$" error in the estimation of the uncertainty set, then one could just apply our theorem with the estimation error, so extending to such estimated uncertainty setting should be trivial. We do not focus on specific uncertainty sets or estimation settings, allowing future RCMDP research to apply their chosen uncertainty set and associated error bounds to our Corollary 1.
>
> [1] Towards Theoretical Understandings of Robust Markov Decision Processes: Sample Complexity and Asymptotics: https://arxiv.org/abs/2105.03863
>
>
> ## About Question 2
> > What are the challenges in extending this result to the case where the uncertainty set is not fully known apriori?
>
> For uncertainty sets like the $L_1$ ball or KL sets, we can derive the sample approximation error bound for robust policy evaluation by applying [1]. However, as far as we know, no paper has derived sample approximation errors for robust policy gradients. We anticipate that recent alternative formulations of robust policy gradients, such as [2], may be helpful in deriving these error bounds, but we must need some effort for the exact derivation. The concrete error-bound derivation lies beyond the scope of this paper.
>
> [1] Towards Theoretical Understandings of Robust Markov Decision Processes: Sample Complexity and Asymptotics: https://arxiv.org/abs/2105.03863
>
> [2] Policy Gradient for Rectangular Robust Markov Decision Processes: https://arxiv.org/abs/2301.13589

---

> ### Author Response · Authors · 2024-11-18
>
> ## About Question 1
>
> > Please categorize the related works into two groups: one that assumes the uncertainty set is fully characterized apriori (see the weakness above) and others that do not. This will help readers put the work into context.
>
> Thank you for the suggestion. Our related work is structured to highlight the limitations of the existing MDP, CMDP, and RMDP approaches—such as DP, LP, and Lagrangian—in addressing the RCMDP setting. Additionally, as we explain in the answer to your major comment, our theorem does not depend on the type of the robust policy (gradient) evaluation errors in Assumptions 1 and 3. Therefore, we think that categorizing based on uncertainty set knowledge falls outside the scope of our work, so the current version retains the previous related work.
>
> ## Question 3
>
> >Does the algorithm scale with the size of the state space? Does it scale with the size of the uncertainty set? Some comments on these should be provided in the paper.
>
> Since we consider the tabular setting, our theoretical guarantee depends on the size of the state space. Essentially, the iteration length $T$ scales to $S^3$. On the other hand, $T$ does not depend on the size of the uncertainty set. The dependence on the uncertainty set is abstracted by the general robust policy evaluation algorithm in Assumption 1.
>
> To help readers understand this point, our current revision includes this description in Remark 2 as follows:
>
> “**Remark 2**: EpiRC-PGS outputs an $\varepsilon$-optimal policy by querying [robust policy gradient computation algorithms] a total of $\widetilde{O}((N+1)K T)$ times. Thus, the computational complexity of EpiRC-PGS algorithm can be expressed as $\widetilde{O}((N+1)KT \times [\text{querying cost}])$.
>
> As a simple example, consider the case where $\mathcal{U}$ is finite, where a single query requires $\widetilde{O}(S^2 A|\mathcal{U}|)$ operations. Using the concrete value of $KT$, the computational complexity of EpiRC-PGS for finite $\mathcal{U}$ becomes $\widetilde{O}(S^5 A^4 |\mathcal{U}| (N+1) \varepsilon^{-4})$.
> Similar analyses can be applied to other types of uncertainty sets.”
>
> ## About Question 4
>
> > What is the typical computational complexity to generate a subgradient estimate with O(ϵ2) error? Such complexities should be elaborated in the paper.
>
> The computational complexity depends on the type of uncertainty set. We think Table 1 in [3] provides a good summary of the existing computational complexity for computing robust policy gradients. Extending their results to our setting requires simply multiplying their complexity by the factor $(\text{Num of constraints} + 1)$. For instance, in the $s$-rectangular ball setting, achieving an $\varepsilon^{-1}$ error results in a computational complexity of $\widetilde{O}((\text{Num of constraints} + 1) \times S^2 A \ln \varepsilon^{-1})$. We have added this computational discussion in Remark 2 in the current revision.
>
> [3] Policy Gradient for Rectangular Robust Markov Decision Processes: https://arxiv.org/abs/2301.13589
>
> =========
>
> **Finally, we want to thank the reviewer again for the detailed and valuable comments. If our response resolves your concerns satisfactorily, we want to kindly ask the reviewer to consider raising the score rating of our work.**
>
> **Please let us know if you have any further questions, and we will be happy to answer them. Additionally, if you have any further concerns, we will be happy to make changes and improve the paper.**

---

### Official Review · Reviewer_YFQK · 2024-11-05

**Soundness:** 3
**Presentation:** 3
**Contribution:** 3
**Rating:** 6
**Confidence:** 4

**Summary:**

The paper aims to develop an algorithm to identify a near-optimal policy in Robust Constrained Markov Decision Processes (RCMDPs). RCMDPs aim to minimize cumulative costs while satisfying constraints in the worst-case scenarios. The authors note that conventional policy gradient approaches to the Lagrangian formulation for RCMDPs often get trapped in suboptimal solutions due to gradient conflicts. To address this, they propose using the epigraph form of the RCMDP problem, which circumvents these conflicts by optimizing the objective or the constraints individually. This approach allows the policy gradient to avoid local minima associated with conflicting gradients. They introduce the Epigraph Robust Constrained Policy Gradient Search (EpiRC-PGS) algorithm, a binary search-based method that iteratively refines policy selection, with theoretical guarantees to find an ε-optimal policy in RCMDPs.

**Strengths:**

1. Applying the epigraph form to RCMDPs is a novel solution to address gradient conflict issues that arise with traditional Lagrangian formulations.

2. The authors offer theoretical proofs that guarantee the proposed algorithm’s convergence to an ε-optimal policy.

3. The paper includes empirical evaluations across several RCMDP settings. The experiments show that the proposed EpiRC-PGS algorithm reliably converges to feasible and low-cost policies, significantly outperforming Lagrangian-based methods in robust settings.

**Weaknesses:**

1. The double-loop structure of the EpiRC-PGS algorithm, while effective, is computationally intensive. The paper mentions this as a limitation, suggesting that a single-loop alternative could improve efficiency, which is a potential area for improvement.

2. The algorithm’s effectiveness relies on the coverage of the initial distribution (Assumption 2). This assumption may not always hold, especially when the state space is large.

**Questions:**

1. How does the algorithm perform when Assumption 2 on the initial distribution is partially met but not fully satisfied? Would the relaxation of this assumption degrade the policy quality significantly?

2. Have alternative subgradient methods or other non-gradient-based approaches been considered for this epigraph-form RCMDP problem? If so, how do they compare with the EpiRC-PGS algorithm in terms of robustness and feasibility?

---

> ### Author Response · Authors · 2024-11-18
>
> **Thank you very much for the valuable comments. We are addressing the concerns raised in the review.**
>
> ## About Weakness 1
>
> > The double-loop structure of the EpiRC-PGS algorithm, while effective, is computationally intensive. The paper mentions this as a limitation, suggesting that a single-loop alternative could improve efficiency, which is a potential area for improvement.
>
> Thank you for pointing this out. As for this Weakness 1, we have already discussed the potential extension to single-loop and the theoretical challenges in Appendix B. Specifically, it provides a Lagrangian extension of our epigraph form, and describes the hardness to show its strong duality.
>
> Therefore, we will address the other weaknesses and questions. To motivate our answer to Question 1, let us start by answering Question 2 and Weakness 2 first.
>
> ## About Question 2
>
> > Have alternative subgradient methods or other non-gradient-based approaches been considered for this epigraph-form RCMDP problem? If so, how do they compare with the EpiRC-PGS algorithm in terms of robustness and feasibility?
>
> We have explored several subgradient and non-gradient-based approaches to the Epigraph problem but have not discovered any viable solutions due to the non-rectangularity of the epigraph’s auxiliary problem (Equation 12 in the current draft). Specifically, solving the epigraph form requires addressing the robust MDP (RMDP) problem in Equation 12:
> $$ \min_\pi \max_{n \in [0, N]} \max_{P \in \mathcal{U}} J_{c_n, P}(\pi) - b_n \quad \text{ (12)}  $$
>
> Although this is an RMDP problem, we remark that existing non-gradient-based methods are not applicable to Equation 12. **Even when $\mathcal{U}$ is $(s, a)$-rectangular, the term $\max_{n \in [0, N]} \max_{P \in \mathcal{U}}$ no longer preserves the rectangular structure, because $\max_{n \in [0, N]}$ is not rectangular.** As a result, we cannot utilize dynamic programming-based algorithms, such as robust value iteration [1].
>
> An alternative approach is to cast the RMDP as a convex optimization problem like [2]. However, their method also relies on rectangularity, which we cannot leverage in this case.
>
> We also considered a natural policy subgradient approach [3]. However, the proof for the existing natural robust policy subgradient requires rectangularity, making it inapplicable to Equation 12.
>
> Consequently, we have not found any viable solutions except for our subgradient method. Line 377 of the current revision briefly describes this point.
>
> [1] Robust dynamic programming: http://www.corc.ieor.columbia.edu/reports/techreports/tr-2002-07.pdf
>
> [2] On the convex formulations of robust Markov decision processes: https://arxiv.org/abs/2209.10187
>
> [3] First-order Policy Optimization for Robust Markov Decision Process: https://arxiv.org/abs/2209.10579
>
> ## About Weakness 2:
>
> The algorithm’s effectiveness relies on the coverage of the initial distribution (Assumption 2). This assumption may not always hold, especially when the state space is large.
> The initial distribution coverage assumption is an almost necessary condition for the policy gradient approach[1]. As demonstrated in Remark 1 of [4], the term $D$ in Theorem 4 cannot be omitted. **Therefore, without the coverage assumption, $D$ would become infinite, and as a result, global convergence in policy gradient methods cannot be guaranteed.**
>
> However, as the answer to Question 2 explains, **our policy subgradient method seems to be the only viable approach even for rectangular settings.** Investigating whether the coverage assumption stems from the algorithmic design or an inherent limitation of the RCMDP problem is a promising direction for future work. Yet, we currently have no solutions to remove the assumption due to the above reasons.
>
> [4] On the Global Convergence Rates of Softmax Policy Gradient Methods: https://arxiv.org/abs/2005.06392

---

> ### Author Response · Authors · 2024-11-18
>
> ## About Question 1
>
> > How does the algorithm perform when Assumption 2 on the initial distribution is partially met but not fully satisfied? Would the relaxation of this assumption degrade the policy quality significantly?
>
> Since robust design is typically performed in simulation environments where the initial distribution can be freely adjusted [5], let us answer your question through a simple relaxation that assigns very small initial probabilities to all the states.
> Given that the robust return function should be Lipschitz continuous with respect to the initial distribution, we expect that this relaxation will not significantly degrade the quality of the policy, as long as the assigned probabilities remain sufficiently small.
>
> However, **we think that proving the impact of this relaxation is non-trivial.** One potential proof technique is a variant of the so-called simulation lemma [6], which provides an upper bound on the effect of disturbances in the transition kernel on the policy's return. However, the existing simulation lemma considers non-robust-constrained settings. Therefore, we must develop a new simulation lemma to assess the impact of disturbances in the initial distribution on the optimal policy's return in RCMDPs. This would require a novel theoretical tool and beyond the scope of this paper.
>
> [5] Introduction to Quality Engineering: Designing Quality Into Products and Processes: https://www.tandfonline.com/doi/pdf/10.1080/00401706.1989.10488520
>
> [6] Reinforcement Learning: Theory and Algorithms: https://rltheorybook.github.io/
>
> =================
>
> **Finally, we want to thank the reviewer again for the detailed and valuable comments. If our response resolves your concerns satisfactorily, we want to kindly ask the reviewer to consider raising the score rating of our work.**
>
> **Please let us know if you have any further questions, and we will be happy to answer them. Additionally, if you have any further concerns, we will be happy to make changes and improve the paper.**

---

> > ### Comment · Reviewer_YFQK · 2024-11-26
> >
> > I thank the authors for the detailed reply. After reviewing it, I have decided to maintain my score.

---

### Official Review · Reviewer_KWav · 2024-11-11

**Soundness:** 2
**Presentation:** 1
**Contribution:** 1
**Rating:** 3
**Confidence:** 4

**Summary:**

This paper reformulate robust constrained MDPs via epigraphical reformulation and propose a bisection algorithm to solve this problem.

**Strengths:**

- This paper addresses an important problem in the field of MDPs

- The idea of bisection search is interesting.

**Weaknesses:**

- The writing is extremely confusing. For example line 205, it is not clear what does "Does pi^star \in...hold?" mean, given that the exact phrase was given in line 204. Also, there are many diamond-shape notation in the draft, and they should be typos.

- The idea, although is interesting, is very simple. The use of epigraphical reformulation is standard and not surprising and technical. The most technical part should be the evaluation of \Delta, but the proposed algorithm relies on existing methods to do this step.

- The underlying idea of Algorithm 1 is not clear and the presentation is poor.

- The convergence of the proposed algorithm is not fast, O(eps^-4), and there is no discuss on the complexity.

- Experiments are not working on large-scale problems.

**Questions:**

See weaknesses.

- What do you mean by "is an unknown value" in Assumption 4?

- Algorithm 1, how do you set T?

---

> ### Author Response · Authors · 2024-11-18
>
> **Thank you very much for the valuable comments. We are addressing the concerns raised in the review.**
>
> ## About Weakness 1
>
> > The writing is extremely confusing. For example line 205, it is not clear what does "Does pi^star \in...hold?" mean, given that the exact phrase was given in line 204.
>
> Thank you for highlighting this point. We recognize that the previous draft contains some misleading phrases and that its readability could be improved. To improve readability and clarity, **we have restructured the draft and provided more detailed explanations.** A summary of the revisions is listed in [the general response](https://openreview.net/forum?id=G5sPv4KSjR&noteId=kPGMvQckyn), so here we specifically provide the revision about the example you mentioned ("what does "Does pi^star \in...hold?" mean,).
>
> * **Previous draft:** The Lagrangian approach aims to solve Equation (3) by expecting $\pi^\star \in \arg\min_\pi L_{\lambda^\star} (\pi)$. To make this approach work, the two questions must be addressed: …
>
> * **Updated draft:** The Lagrangian approach aims to solve Equation (3) by expecting $\pi^\star \in \arg\min_{\pi \in \Pi} L_{\lambda^\star}(\pi)$. However, this expectation may not hold, as swapping the min-max is not necessarily equivalent to the original max-min problem. Therefore, to guarantee the performance of the Lagrange approach, the two questions must be addressed:
> $\mathrm{(i)}$ Can we ensure $\pi^\star \in \arg\min_{\pi \in \Pi} L_{\lambda^\star}(\pi)$?  $\mathrm{(ii)}$ If so, is it tractable to solve $\min_{\pi \in \Pi} L_{\lambda}(\pi)$?
>
> ## About Weakness 1 (cont.)
>
> > Also, there are many diamond-shape notations in the draft, and they should be typos.
>
> Thank you for pointing this out. We think that the “diamond-shape notation,” you mention refers to $\diamondsuit$ used in Appendix A.4. $\diamondsuit$ is not a typo; it’s a symbol properly defined in Equation (12) in the previous draft. However, we recognized that the notation may be confusing, so our current revision has changed $\diamondsuit$ to $\widetilde{J}^\star$. We also updated other notations for better readability (please refer to the [the general response](https://openreview.net/forum?id=G5sPv4KSjR&noteId=kPGMvQckyn)).
>
> ## About Weakness 2
>
> > The idea, although is interesting, is very simple.The use of epigraphical reformulation is standard and not surprising and technical. The most technical part should be the evaluation of \Delta, but the proposed algorithm relies on existing methods to do this step.
>
> While epigraph reformulation is standard in general optimization, **it is not standard in MDPs.** No prior RCMDP research has used Epigraph, and even in CMDP settings, there exist few papers that used Epigraph [2][3]. Moreover, they did not establish theoretical guarantees, such as global convergence like ours. We added these references in the related work A.1.
>
> Furthermore, although the epigraph form is standard in constrained optimization, our discovery that the epigraph can solve the RCMDP problem is non-trivial. Our section 3 (Challenges of Lagrangian Formulation) shows that the Lagrangian form, which is as standard as the epigraph in constrained optimization, fails in RCMDPs. Furthermore, other MDP methods like dynamic programming and linear programming are not applicable here (line 043). This highlights the non-triviality of the RCMDP problem as it seems unsolvable at first glance. Indeed, while several papers have tackled RCMDPs (Russel et al. (2020); Mankowitz et al. (2020), Sun et al. (2024), Ghosh (2024), Zhang et al. (2024) in the draft),  none have succeeded in solving the RCMDP. **In contrast, this is the first work to theoretically solve RCMDPs, and the approach is non-standard in the MDP literature.**
>
> Finally, as described by the [KISS (Keep It Short and Simple) principle](https://en.wikipedia.org/wiki/KISS_principle), we believe that simplicity is a strength, not a weakness. The value of the contribution should be assessed based on "what problem the paper solves," rather than "what techniques were used to solve it." Our paper makes a substantial contribution to the RCMDP field by solving the previously unsolved problem with a simple yet effective approach.
>
> [2] Solving Stabilize-Avoid Optimal Control via Epigraph Form and Deep Reinforcement Learning: https://arxiv.org/abs/2305.14154
>
> [3] Solving Minimum-Cost Reach Avoid using Reinforcement Learning: https://arxiv.org/abs/2410.22600

---

> ### Author Response · Authors · 2024-11-18
>
> ## About Weakness 3
>
> > The underlying idea of Algorithm 1 is not clear and the presentation is poor.
>
> Thank you very much for your suggestion regarding readability. As you suggest, we recognize that the previous draft is not well-organized: For example, in the earlier draft, the algorithmic Idea and the algorithm implementations were discussed in separate sections, which complicated the motivation of each component of our EpiRC-PGS algorithm.
>
> To improve the readability, **we have restructured the draft, specifically by combining the motivation behind the algorithm with its theory and implementation into a single section.** As an example of the update, the revised Section 5.1 now describes the motivation, theories, and implementation of the binary search, while Section 5.2 does the same for the policy gradient subroutine.
>
> **Additionally, we have updated the notations to provide better intuition about our algorithm.** For example, in Assumption 1 of the new draft, we introduce the robust policy evaluator $\widehat{J}_n(\pi)$, which estimates the robust return value of $\pi$. Similarly, a new notation for the robust policy gradient evaluator is introduced in Assumption 3. These updates aim to clarify that our EpiRC-PGS algorithm can be implemented using existing techniques for robust policy gradient algorithms.
>
> Please refer to [the General Response](https://openreview.net/forum?id=G5sPv4KSjR&noteId=kPGMvQckyn) for other updates of the current revision.
>
> ## About Weakness 4
>
> > The convergence of the proposed algorithm is not fast, O(eps^-4),
>
> The convergence rate $O(\epsilon^{-4})$ is not tight in CMDP and RMDP settings. **However, it is important to note that no previous work has addressed the RCMDP problem, and ours is the first algorithm to do so.** Consequently, while $O(\epsilon^{-4})$ is slow in the RMDP or CMDP problems, it may not be slow for the RCMDP setting, as there is currently no established lower bound or faster algorithm for RCMDPs.
>
> > and there is no discuss on the complexity.
>
> Thank you for highlighting the importance of the complexity discussion. In Remark 2 in the current draft, we have added the discussion on the computational complexity analysis as follows:
>
> **Remark 2:** EpiRC-PGS outputs an $\varepsilon$-optimal policy by querying [robust policy gradient computation algorithms] a total of $\widetilde{O}((N+1)K T)$ times. Thus, the computational complexity of EpiRC-PGS can be expressed as $\widetilde{O}((N+1)KT \times [\text{querying cost}])$.
> As a simple example, consider the case where $\mathcal{U}$ is finite, where a single query requires $\widetilde{O}(S^2 A|\mathcal{U}|)$ operations. Using the concrete value of $KT$, the computational complexity of EpiRC-PGS for finite $\mathcal{U}$ becomes $\widetilde{O}(S^5 A^4 |\mathcal{U}| (N+1) \varepsilon^{-4})$.
> Similar analyses can be applied to other types of uncertainty sets.
>
> ## About Weakness 5
>
> > Experiments are not working on large-scale problems.
>
> Since our focus is on proposing the first theoretically guaranteed algorithm for RCMDP and its theoretical analysis in the tabular setting, the aim of experiments is to confirm the theoretical guarantees of EpiRC-PGS (Corollary 1) and demonstrate the limitations of the Lagrangian method (Theorem 1) in identifying near-optimal policies. Since our theory does not cover the high-dimensional settings, we do not conduct high-dimensional experiment. We believe our experimental results (Figure 3) sufficiently achieve this aim, as the Lagrangian baselines failed to identify feasible policies, while EpiRC-PGS successfully identified them.
>
> It is also noteworthy that previous research on RCMDPs has predominantly focused on large-scale empirical algorithms, overlooking the foundational theory of tabular MDPs. **However, as evidenced by the history of RL, a solid theoretical foundation in tabular MDPs is essential for the advancement of large-scale algorithms.** For example, DQN is based on Q-learning, TRPO stems from Safe Policy Iteration [3], and SAC originates from Soft Q-learning. Without theoretical guarantees in the tabular setting, it would be impossible to establish reliable RL algorithms that perform well in high-dimensional problems. This study contributes to addressing this gap by establishing foundational theories for RCMDPs, though its extension to high-dimensional implementations is beyond the current scope.
>
> [3]: Safe Policy Iteration: https://proceedings.mlr.press/v28/pirotta13.html

---

> > ### Author Response · Authors · 2024-11-18
> >
> > ## About Quesitions
> >
> > > What do you mean by "is an unknown value" in Assumption 4?
> >
> > Thank you for pointing this out. In the previous draft, we have used "unknown" in the sense that our EpiRC-PGS algorithm does not require $\varepsilon_{eps}$ and $\varepsilon_{grd}$ as its inputs.
> >
> > However, we recognized that the term “unknown” is not proper here because producing an $\varepsilon$-optimal policy with EpiRC-PGS requires adjusting the iteration length based on the value of $\varepsilon_{eps}$ (see Theorem 6 for details). Consequently, in our revision, we have removed references to $\varepsilon_{eps}$ and $\varepsilon_{grd}$ as "unknown."
> >
> > > Algorithm 1, how do you set T?
> >
> > $T$ is set to $\varepsilon^-4$ to output $\varepsilon$-optimal policy. Theorem 5 and Corollary 1 in the current draft describe the iteration complexity.
> >
> > =======
> >
> > **Finally, we want to thank the reviewer again for the detailed and valuable comments. If our response resolves your concerns satisfactorily, we want to kindly ask the reviewer to consider raising the score rating of our work.**
> >
> > **Please let us know if you have any further questions, and we will be happy to answer them. Additionally, if you have any further concerns, we will be happy to make changes and improve the paper.**

---

> > > ### Comment · Reviewer_KWav · 2024-11-26
> > >
> > > Thank you for your response. I still think papers in ICLR deserve more and larger experiments. Also, this algorithm does not appear to be useful with the high complexity. I decide to keep my score.

---

> ### Author Response · Authors · 2024-11-27
>
> Thank you for your comments. However, we respectfully disagree with your opinion that ICLR deserves more and larger experiments. According to the [ICLR website](https://iclr.cc/About), ICLR’s scope is not limited to large-scale experimental results but extends to contributions in the field of learning as a whole. Indeed, many theoretical papers without extensive experiments have been accepted and recognized as spotlight or notable top-5% papers in ICLR, such as:
>
> * Pessimistic Model-based Offline Reinforcement Learning under Partial Coverage (ICLR 2022) :  https://openreview.net/forum?id=tyrJsbKAe6
> * Harnessing Density Ratios for Online Reinforcement Learning (ICLR 2024 Spotlight) https://openreview.net/forum?id=THJEa8adBn
> * The Role of Coverage in Online Reinforcement Learning (ICLR 2023 notable-top-5%) https://openreview.net/forum?id=LQIjzPdDt3q
>
> Our paper presents a significant contribution by being the first to solve robust constrained MDPs, a fundamental framework for reliable reinforcement learning algorithms. Furthermore, as detailed in Section 5.2, our algorithm employs gradient-based updates, making it inherently scalable to large-scale problems. Rejecting such foundational work solely due to its lack of immediate large-scale applications risks a significant loss for the research community. For example, while the famous value iteration algorithm cannot directly solve large-scale problems, without it, the recent deep RL algorithms like DQN would not exist.

---

### Official Review · Reviewer_1d8S · 2024-11-13

**Soundness:** 4
**Presentation:** 3
**Contribution:** 3
**Rating:** 6
**Confidence:** 3

**Summary:**

This paper proposes the EpiRC-PGS algorithm for solving the robust constrained Markov Decision Process (RCMDP) problem and provides a theoretical proof for identifying an $\epsilon$-optimal policy. The paper first highlights the limitations of the traditional Lagrangian formulation in RCMDPs, introducing the epigraph form as a more effective approach for handling the policy optimization process. Finally, the authors present the EpiRC-PGS algorithm as a solution, which leverages the epigraph form to address RCMDPs effectively.

**Strengths:**

1. This paper identifies key limitations in traditional algorithms for robust constrained Markov Decision Processes (RCMDPs) and presents an innovative, streamlined approach to overcome these challenges.
2. The shift to an epigraph form represents a significant advancement, simplifying the solution method and providing an effective framework for robust optimization.
3. The empirical comparison highlights the practical effectiveness of EpiRC-PGS, showing its ability to achieve low-return, constraint-satisfying policies, making it an attractive solution in robust decision-making applications.

**Weaknesses:**

1. A key area for improvement is a more thorough analysis of the algorithm’s efficiency, especially in higher-dimensional state spaces.
2. While the $\epsilon$-optimality proof is solid, addressing any computational overhead in the inner loop for larger state spaces would enhance its applicability to real-world problems.

**Questions:**

Currently, experiments appear limited to low-dimensional cases. Extending tests to a more complex environment, like a two-dimensional space (e.g., velocity and distance in driving), could clarify if EpiRC-PGS maintains its time and space efficiency as complexity grows.

---

> ### Author Response · Authors · 2024-11-18
>
> **Thank you very much for the valuable comments. We are addressing the concerns raised in the review.**
>
> ## About Weakness 1
>
> > A key area for improvement is a more thorough analysis of the algorithm’s efficiency, especially in higher-dimensional state spaces.
>
> As you mentioned, studying high-dimensional problems is valuable, and our paper does not study high-dimensional settings where function approximation is required. Indeed, we only study the tabular setting and the computational complexity of our algorithm scales as $\widetilde{O}(S^3)$, where $S$ is the number of states as described in Remark 2 in the current draft. Note that S increases exponentially with the dimensionality of the states.
>
> **However, we remark that even in this tabular setting, no paper has solved this problem.** Additionally, since RMDP is known to become easily NP-hard without careful assumptions [1], extending tabular to high-dimensional settings may require non-trivial discussions on the structure of the uncertainty set and may demand significantly greater effort. Thus, we currently do not know what kind of theoretical extension to a high-dimensional setting is achievable.
>
> Additionally, we note that our EpiRC-PGS leverages a combination of binary search and policy gradient, the latter being inherently scalable and widely used in deep RL. Therefore, we believe it is likely that a practical deep RL algorithm could be built on top of our proposed epigraph framework, but such an empirical extension lies beyond the scope of this paper.
>
> [1] Robust Markov Decision Processes: https://pubsonline.informs.org/doi/abs/10.1287/moor.1120.0566?journalCode=moor
>
>
> ## About Weakness 2:
>
> > While the ϵ-optimality proof is solid, addressing any computational overhead in the inner loop for larger state spaces would enhance its applicability to real-world problems.
>
> As you suggest, the double-loop algorithm is often inefficient in practice. To address the potential inefficiency of the double-loop structure in practical computation, we present a possible single-loop solution and discuss the theoretical challenges in Appendix B.
> Specifically, it provides a Lagrangian extension of our epigraph form, and describes the hardness to show its strong duality.
>
> **Additionally, we remark that algorithmic development in MDPs often begins with double-loop algorithms [2] [3] before extending to a single-loop architecture [4] [5].** Thus, we believe our research, while currently based on a double-loop structure, represents an important stepping stone for future advancements in the RCMDP study.
>
> [2] A Dual Approach to Constrained Markov Decision Processes with Entropy Regularization: https://arxiv.org/abs/2110.08923
>
> [3] Policy Gradient for Rectangular Robust Markov Decision Processes: https://arxiv.org/abs/2212.10439
>
> [4] Last-Iterate Convergent Policy Gradient Primal-Dual Methods for Constrained MDPs: https://arxiv.org/abs/2306.11700
>
> [5] A Single-Loop Robust Policy Gradient Method for Robust Markov Decision Processes: https://arxiv.org/abs/2406.00274
>
> ## About Question
>
> > Currently, experiments appear limited to low-dimensional cases. Extending tests to a more complex environment, like a two-dimensional space (e.g., velocity and distance in driving), could clarify if EpiRC-PGS maintains its time and space efficiency as complexity grows.
>
> **Since our focus is on proposing the first theoretically guaranteed algorithm for tabular RCMDP and its theoretical analysis in the tabular setting, the aim of experiments is to confirm the theoretical guarantees of EpiRC-PGS (Corollary 1) and demonstrate the limitations of the Lagrangian method (Theorem 1) in identifying near-optimal policies.** Additionally, for low-dimensional experiments like driving, we think as long as we can discretize the state space into a finite tabular set, the dimensionality does not matter.
>
> Thus, we believe our experimental results (Figure 3) sufficiently achieve this aim, as the Lagrangian baselines failed to identify feasible policies, while EpiRC-PGS successfully identified them.
>
> =========
>
> **Finally, we want to thank the reviewer again for the detailed and valuable comments. If our response resolves your concerns satisfactorily, we want to kindly ask the reviewer to consider raising the score rating of our work.**
>
> **Please let us know if you have any further questions, and we will be happy to answer them. Additionally, if you have any further concerns, we will be happy to make changes and improve the paper.**

---

> > ### Comment · Reviewer_1d8S · 2024-11-27
> >
> > Thank you for your reply. I will maintain my score.

---

### Author Response · Authors · 2024-11-18
**General Response**

**First of all, we would like to sincerely thank all the reviewers for their valuable feedback and comments.**

The reviewers mainly pointed out the strengths of our theoretical contributions: our paper identifies key limitations in the traditional Lagrangian approach for robust constrained MDPs (RCMDPs), and overcomes the challenges by the simple epigraph approach.

On the other hand, the main criticisms were:
1. the lack of a high-dimensional algorithmic extension and experiment [1d8S, KWav],
2. the lack of complexity analysis [KWav, yT6U, FLm4],
3. and the inefficiency of the double-loop structure [YFQK, FLm4],
4. Additionally, the reviewer KWaV pointed out the readability issue of our paper.

For each review, we have provided detailed responses and used the feedback to refine our draft, improving its readability and addressing several questions. Below, we outline our key comments and summarize the key updates in the revised draft:

## Restatement of our contribution

We remark that our EpiRC-PGS is the first algorithm that is **theoretically** guaranteed to identify a near-optimal policy in RCMDPs. While several papers have tackled RCMDPs (Russel et al. (2020); Mankowitz et al. (2020), Sun et al. (2024), Ghosh (2024), Zhang et al. (2024) in the draft), **even in the tabular setting, none have succeeded in solving the RCMDP.**

Most of these existing RCMDP research has predominantly focused on large-scale empirical algorithms, overlooking the foundational theory of tabular MDPs. **However, as evidenced by the history of RL, a solid theoretical foundation in tabular MDPs is essential for the advancement of large-scale algorithms.** For example, DQN is based on Q-learning, TRPO stems from Safe Policy Iteration [1], and SAC originates from Soft Q-learning. Without theoretical guarantees in the tabular setting, it would be impossible to establish reliable RL algorithms that perform well in high-dimensional problems. This study contributes to addressing this gap by establishing foundational theories for RCMDPs.

[1]: Safe Policy Iteration: https://proceedings.mlr.press/v28/pirotta13.html

---

> ### Author Response · Authors · 2024-11-18
>
> ## Draft Updates
>
> The key updates of the draft are summarized as follows:
>
> 1. **We have restructured the draft to address readability concerns raised by reviewer KWaV.** For example, in the earlier draft, the algorithmic Idea and the algorithm implementations were discussed in separate sections, which complicated the motivation of each component of our EpiRC-PGS algorithm. To enhance readability, we have combined the motivation behind the algorithm with its implementation into a single section. As an example of the update, the revised Section 5.1 now describes the motivation, theories, and implementation of the binary search, while Section 5.2 does the same for the policy gradient subroutine.
> 2. **To enhance intuition about the algorithms, we have updated the notations, particularly those related to the evaluators of $\Delta_{b_0}(\pi)$.** In Assumption 1 of the new draft, we introduce the robust policy evaluator $\widehat{J}_n(\pi)$, which estimates the robust return value of $\pi$. Similarly, a new notation for the robust policy gradient evaluator is introduced in Assumption 3. These updates aim to clarify that our EpiRC-PGS algorithm can be implemented using existing techniques for robust policy gradient algorithms.
> 3. We provide more clear description in the current draft to address the clarity concern raised by reviewer KWaV. For example,
>     * **We updated the description of the problem of  Lagrangian methods in line 200.** The current draft explains
> “The Lagrangian approach aims to solve Equation (3) by expecting $\pi^\star \in \arg\min_{\pi \in \Pi} L_{\lambda^\star}(\pi)$. However, this expectation may not hold, as swapping the min-max is not necessarily equivalent to the original max-min problem. Therefore, to guarantee the performance of the Lagrange approach, the two questions must be addressed:
> $\mathrm{(i)}$ Can we ensure $\pi^\star \in \arg\min_{\pi \in \Pi} L_{\lambda^\star}(\pi)$? $\quad$  $\mathrm{(ii)}$ If so, is it tractable to solve $\min_{\pi \in \Pi} L_{\lambda}(\pi)$?”
>     * **The current Line 306 provides a detailed motivation why we do not consider a specific uncertainty set as follows:**
> “Note that without any assumptions about the uncertainty set $\mathcal{U}$, solving an RCMDP is NP-hard. However, imposing concrete structures on $\mathcal{U}$ can restrict the applicability of EpiRC-PGS. To enable our algorithm to handle a broader class of $\mathcal{U}$, we consider $\mathcal{U}$ where we can approximate the robust return value $J_{c_n, \mathcal{U}}(\pi)$ and its subgradient $\partial J_{c_n, \mathcal{U}}(\pi)$ as follows: … “
> 4. **In Remark 2 at line 470, we have added a computational complexity example of our proposed algorithm.** Specifically, our algorithm requires
> $$\widetilde{O} \left([\text{ Num of constraints } + 1] \times [\text{ inner loop iteration length }] \times [\text{robust policy gradient computation cost}] \right)$$
> operations. Since this computational complexity depends on the type of uncertainty set, in Remark 2, we only provide the analysis for the finite uncertainty set as a simple example. In this finite case, the complexity becomes $\widetilde{\mathcal{O}}(D^4 S^5 A^4 H^{14} |\mathcal{U}| \varepsilon^{-4} \times [\text{ Num of constraint + 1}])$. Similar analyses can be applied to other uncertainty sets.

---

### Meta-Review · Area_Chair_xjJc · 2024-12-21

**Metareview:**

This paper studies the problem of identifying near-optimal policies in Robust Constrained Markov Decision Processes (RCMDPs) using a novel approach based on the epigraph formulation. It achieves theoretical guarantees with convergence to an $\epsilon$-optimal policy, addressing key limitations in traditional Lagrangian methods through binary search and policy gradient subroutines. The reviewers highlighted the significant theoretical contributions, such as addressing gradient conflicts in robust optimization and demonstrating limitations of existing approaches, while also raising concerns about computational efficiency and the scope of experiments. During the rebuttal phase, the authors effectively clarified these points, emphasizing the foundational nature of the work and the relevance of their theoretical framework in advancing robust reinforcement learning. Although one rejecting reviewer criticized the lack of large-scale experiments, this critique appears misplaced, given the paper’s focus on theoretical advancements. Overall, the paper makes a substantial contribution to the theory of RCMDPs and is well-positioned for acceptance based on its foundational impact.

**Additional Comments On Reviewer Discussion:**

During the rebuttal period, several key points were raised by the reviewers. Reviewer 1d8S highlighted the need for a thorough analysis of the algorithm's efficiency and scalability, which the authors addressed by discussing the theoretical computational complexity in the revised draft and clarifying the potential for extending the work to high-dimensional settings. Reviewer KWav criticized the lack of large-scale experiments and the algorithm's high complexity but was reminded by the authors of the theoretical focus and foundational nature of the paper. Reviewer YFQK questioned the algorithm's reliance on initial state coverage and assumptions about robust policy evaluation, which the authors justified as necessary for policy gradient approaches and provided detailed clarification on their applicability. Reviewer FLm4 raised concerns about computational intensity and empirical evaluation, which were addressed by emphasizing the theoretical guarantees and the intent to focus on foundational contributions rather than practical benchmarks. Overall, the authors' responses demonstrated a clear understanding of the issues and provided sufficient justifications.

---

### Decision · Program_Chairs · 2025-01-22

Accept (Poster)